# Training A Foundation Model to Represent Graphs as Vectors

## Abstract

This paper aims to train a graph foundation model that is able to represent any graph as a vector preserving structural and semantic information useful for downstream graph-level tasks such as graph classification and graph clustering. To learn the features of graphs from diverse domains while maintaining strong generalization ability to new domains, we propose a multi-graph-based feature alignment method, which constructs weighted graphs using the attributes of all nodes in each dataset and then generates consistent node embeddings. To enhance the consistency of the features from different datasets, we propose a density maximization mean alignment algorithm with guaranteed convergence. The original graphs and generated node embeddings are fed into a graph neural network to achieve discriminative graph representations in contrastive learning. More importantly, to enhance the information preservation from node-level representations to the graph-level representation, we construct a reference distribution module without using any pooling operation. We also provide a theoretical generalization bound to support the effectiveness of the proposed model. The experimental results of few-shot/zero-shot graph classification and graph clustering show that our model outperforms the competitors.

## 1 Introduction

Graph data is a fundamental and widely prevalent form of structured data, representing entities as nodes and their relationships as edges. It plays a crucial role in diverse domains, including social networks, biological systems, citation networks, recommendation systems, and knowledge graphs. Given its ability to model complex relational patterns, graph data analysis has become a key focus in machine learning and data mining. In node-level tasks, the training set is usually a single but large graph, on which each node represents a sample. Node-level tasks include node embedding or representation (Grover & Leskovec, 2016; Cai et al., 2018), node classification (Kipf & Welling, 2017), node clustering (Wang et al., 2023), link prediction (Martínez et al., 2016), etc. For example, node classification might involve categorizing users in a social network, while link prediction could be used to recommend new connections.

On the other hand, graph-level tasks operate on entire graphs, where a dataset is composed of numerous graphs, and each graph is treated as a sample. Graph-level tasks address broader challenges such as graph comparison (Kobler et al., 2012), representation learning (Sun et al., 2020), classification (Xu et al., 2019), clustering (Cai et al., 2024), generation (Liao et al., 2019), etc. Graph comparison often relies on graph kernels (Gärtner et al., 2003; Vishwanathan et al., 2010; Shervashidze et al., 2011) or distances (Bunke, 1997; Zeng et al., 2009; Mémoli, 2011; Bento & Ioannidis, 2018) or deep learning methods (Sun & Fan, 2024) to measure similarity between different graphs, while graph representation learning aims to encode entire graphs into compact, informative embeddings for downstream tasks (You et al., 2020; 2021; Sun et al., 2023b). Graph classification, for example, is critical in chemistry for predicting molecular properties (Gilmer et al., 2017; Wang & Fan, 2024), whereas graph generation enables the creation of novel structures, such as drug-like molecules in computational biology (Hoogeboom et al., 2022).

The aforementioned methods of node-level learning and graph-level learning are dataset-specific. That means, for one dataset, we have to train a new model, e.g., a graph neural network, to solve the corresponding problem. This leads to the following two limitations. First, training a model

from scratch is very time-consuming, and it requires model selection and parameter tuning, which brings inconvenience to practical applications. Second, knowledge from historical data or tasks in the same domain or similar domains cannot be exploited well. To address the limitations, recently, graph foundation models (GFMs) have become an increasingly prominent area of research in graph data analysis due to their ability to harness diverse datasets to enhance performance across multiple tasks and domains. Emerging studies (Galkin et al., 2023; Zheng et al., 2023) indicate that GFMs exhibit strong generalization capabilities, even when applied to previously unseen graph structures. Mao et al. (2024) categorizes GFMs into three distinct types according to their adaptability: domain-specific, task-specific, and primitive models. Domain-specific GFMs (Xia et al., 2023; Zhang et al., 2023; Zheng et al., 2023) focus on extracting universal features within a specialized domain, enabling a single model to address multiple related tasks while often surpassing the performance of dedicated task-specific models. Task-specific GFMs (Galkin et al., 2024; Zhao et al., 2024b; Lachi et al., 2024), on the other hand, are trained on extensive datasets to excel at particular tasks, making them particularly valuable in domains where data is scarce. Primitive GFMs (Tang et al., 2024b) offer greater flexibility, but are limited in their applicability to specific datasets and tasks.

One key challenge in building GFMs is that graph patterns from different domains exhibit significant variation (Galkin et al., 2023), which is evident in both structural and feature representations. For example, in molecular graphs (Yang et al., 2016), the structure encodes 3D spatial arrangements and atomic bonds, while node features represent chemical properties. Conversely, in social networks (Dwivedi et al., 2023), the structure reflects user connections, and node features correspond to user profiles. These distribution differences make it difficult for a single model to learn domain-agnostic representations. A promising approach involves transforming both graph structures and node features into textual formats, then employing large language models (LLMs) to derive unified representations (Fatemi et al., 2023; Liu et al., 2023a; Tang et al., 2024a; Wang et al., 2024). Another approach is to improve existing graph learning paradigms (Liu et al., 2025) through innovations in the aspects of the backbone (Rong et al., 2020), pre-training (Qiu et al., 2020; You et al., 2020; Yu et al., 2025a), and adaptations (Fu et al., 2025; Yu et al., 2025c; Wang et al., 2025).

Despite these excellent works on GFMs, there are still a few limitations and much room for improvement. First, in LLM-based GFM, converting graph data into text may lose important information of topological structures and node features of graphs (Yu et al., 2025a), and the computation cost is often high due to the large sizes of LLMs. Second, most of the existing GFMs were proposed for node-level tasks (Zhao et al., 2024b; Wei et al., 2024; Wang et al., 2025) rather than graph-level tasks (Yu et al., 2025c; Fu et al., 2025), while graph-level tasks are often more challenging due to the difficulty in representing each graph as a vector or comparing two graphs (Mémoli, 2011; Sun & Fan, 2024). Lastly, training a single graph model across diverse domains while maintaining strong generalization ability to new domains or various downstream tasks remains an open problem.

This work proposes a GFM for graph-level tasks across diverse domains. Our contributions are:

- We present a novel GFM framework using a multi-graph construction based feature alignment strategy. The motivation is that the association of samples in a dataset is often sufficient to provide efficient solutions to many learning problems. Therefore, we extract the association information in each graph dataset by constructing multi-weight graphs of all nodes. The global graphs can generate relatively consistent features across domains.
- We provide a novel density-maximization algorithm to enhance the alignment of cross-domain features and prove its convergence theoretically. It is also useful in handling graphs without inherent node attributes.
- We develop a reference distribution module to improve the graph-level representation since simply performing pooling will result in significant information loss of node embeddings.
- We provide a generalization error bound to support the effectiveness of our model theoretically.

The experiments in the tasks of few/zero-shot graph classification and graph-level clustering demonstrate the effectiveness and superiority of our model in comparison to state-of-the-art competitors.

## 2 RELATED WORK

**Language Model-Free GFMs** Many studies have explored training GFMs using the "pre-train and adaptation" paradigm, leveraging message-passing or transformer-based GNN backbones.

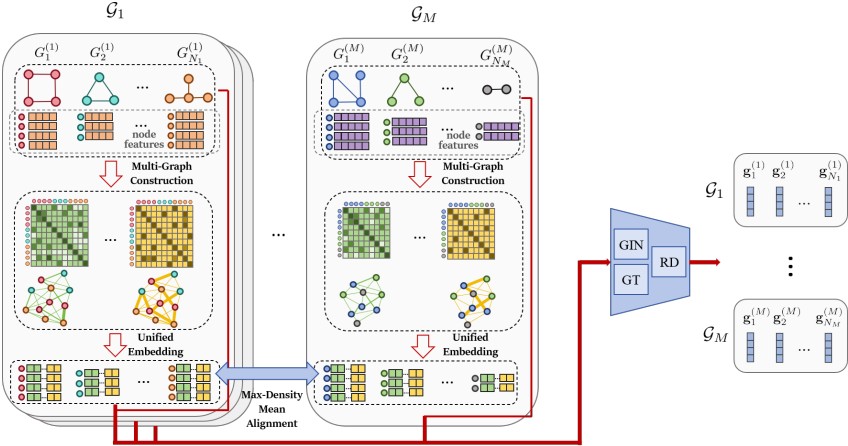

Figure 1: Flow-chart of the proposed method GraphVec-FM. $\mathcal{G}_1, \ldots, \mathcal{G}_M$ are $M$ datasets from different domains. The model represents each graph $G_i^{(j)}$ as a single vector $\mathbf{g}_i^{(j)}$ that can be used in graph-level downstream tasks.

These approaches typically employ contrastive or generative self-supervised learning for pretraining, followed by fine-tuning a subset of model parameters to adapt to downstream tasks or datasets (Liu et al., 2025). Contrastive methods including GCC (Qiu et al., 2020), InfoGraph (Sun et al., 2019), DGI (Veličković et al., 2019), SimGRACE (Xia et al., 2022), GCOPE (Zhao et al., 2024a) maximize agreement between augmented views to learn transferable representations, while generative methods (Hou et al., 2022; 2023) pre-train via graph reconstruction or property prediction. Recently, graph prompt tuning (Sun et al., 2022; Fang et al., 2023; Sun et al., 2023a; Liu et al., 2023b; Fu et al., 2025) has emerged to bridge the pretraining–downstream gap, and many recent GFMs adopt this paradigm (Yuan et al., 2025; Yu et al., 2025b). However, these works mainly emphasize adaptation, leaving the problem of learning unified graph representations underexplored. Due to space limitations, we defer more discussion of **Language Model-Free GFMs** and the related work on **LLM-based GFMs** and **GFM for Graph-Level Tasks** to Appendix B.

## 3 PROBLEM FORMULATION

First of all, the major notations used in this paper are shown in Table 5. Let $\mathcal{D} = \{\mathcal{G}_1, \mathcal{G}_2, \ldots, \mathcal{G}_M\}$ be a union of $M$ datasets of labeled graphs from $M$ different domains, where $\mathcal{G}_j = \{(G_1^{(j)}, y_1^{(j)}), (G_2^{(j)}, y_2^{(j)}), \ldots, (G_{N_j}^{(j)}, y_{N_j}^{(j)})\}$. Here, each graph $G_i^{(j)}$ is denoted as $G_i^{(j)} = (\mathbf{A}_i^{(j)}, \mathbf{X}_i^{(j)}, y_i^{(j)})$, where $\mathbf{A}_i^{(j)} \in \mathbb{R}^{n_i^{(j)} \times n_i^{(j)}}$ denotes the adjacency matrix, $\mathbf{X}_i^{(j)} \in \mathbb{R}^{n_i^{(j)} \times d_i^{(j)}}$ denotes the node attribute matrix, $n_i^{(j)}$ denotes the number of nodes, $d_i^{(j)}$ denotes the number of attributes, and $y_i^{(j)}$ denotes the graph label. Our goal is to use $\mathcal{D}$ to train a GFM, denoted as

$$F : \mathbb{G} \to \mathbb{R}^r \tag{1}$$

to represent any graph from the space $\mathbb{G}$ as an $r$-dimensional vector that is useful in downstream tasks such as graph classification, where $\mathbb{G}$ denotes the set of all graphs in the form of $(\mathbf{A}, \mathbf{X})$. Therefore, $F$ serves as a universal graph representation model.

To learn $F$ from $\mathcal{D}$, we need to address the following challenges:

- **Attributes inconsistency** The node attributes of graphs from different domains are different and not comparable at all. Thus the node attributes in $\mathcal{D}$ cannot be fed into $F$ directly.
- **Attributes absence** Many graph datasets do not contain node attributes, making them very different from graph datasets with node attributes. The heuristic method of constructing node attributes, such as using node degrees, does not comply with the semantic attributes of other datasets.
- **Information loss in pooling** Although there have been a few advanced graph pooling methods (Liu et al., 2022), converting nodes' embeddings into a single vector cannot fully utilize the information.

## 4 METHODOLOGY

### 4.1 MULTI-GRAPH BASED FEATURE ALIGNMENT

As mentioned, the node features of different domain graph datasets may vary significantly in semantics and dimensions. To capture domain-invariant features, we focus on the relationships among nodes across the entire dataset rather than the original features. The reason is that in many machine learning problems, using the relationships between samples or a graph constructed from the dataset can provide effective solutions. For instance, in spectral clustering (Ng et al., 2001), we use a similarity graph rather than the original features; in kernel support vector machine (Cortes & Vapnik, 1995), we can use a Gaussian kernel matrix, which is a similarity matrix of the data points. **Global Graph Construction** For each graph dataset $\mathcal{G}_j$, $j \in [M]$, we propose to construct a similarity graph over all nodes in the dataset using a Gaussian kernel function, i.e.,

$$\mathbf{K}_\lambda^{(j)} = \left[ \exp\left( -\frac{\|\mathbf{x}_u - \mathbf{x}_v\|_2^2}{2\lambda\mu^2} \right) \right]_{u,v=1}^{\bar{N}_j}, \quad \bar{N}_j = \sum_{i=1}^{N_j} n_i^{(j)} \tag{2}$$

where $\mathbf{x}_i$ is the $i$-th row of $\mathbf{X} := \|_{i=1}^{N_j} \mathbf{X}_i^{(j)}$ (vertical concatenation), $\mu$ is the mean of the pairwise distances between all nodes in the dataset, and $\lambda$ controls the bandwidth of the kernel. This setting ensures *translation, rotation, and scaling invariance*, which is important to extract comparable features across diverse datasets. $\mathbf{K}_\lambda^{(j)}$ is the adjacency matrix of this global graph of the nodes in $\mathcal{G}_j$. Note that a single $\mathbf{K}_\lambda^{(j)}$ exploits partial information of the node attributes of $\mathcal{G}_j$ and the optimal setting of $\lambda$ remains an open problem. Therefore, we use a number of different values for $\lambda$, e.g. $\lambda_1, \lambda_2, \ldots, \lambda_Q$, to construct **multiple global graphs** for the nodes in $\mathcal{G}_j$:

$$\mathcal{K}^{(j)} := \left\{ \mathbf{K}_{\lambda_1}^{(j)}, \mathbf{K}_{\lambda_2}^{(j)}, \ldots, \mathbf{K}_{\lambda_Q}^{(j)} \right\}, \quad j \in [M] \tag{3}$$

Note that the diversity of $\mathcal{K}^{(j)}$ can be further enhanced if more kernel families, e.g., $k(\mathbf{x}_u, \mathbf{x}_v) = \exp(-\alpha\|\mathbf{x}_u - \mathbf{x}_v\|_1)$, are considered.

**Consistent Embeddings from Global Graphs** For $\mathbf{K}_{\lambda_q}^{(j)}$, we compute $\bar{d}$-dimensional node embeddings using singular value decomposition (SVD):

$$\mathbf{Z}_{\lambda_q}^{(j)} = \mathbf{U}_{\bar{d}} \boldsymbol{\Sigma}_{\bar{d}}^{1/2}, \quad \mathbf{K}_{\lambda_q}^{(j)} = \mathbf{U}\boldsymbol{\Sigma}\mathbf{V}^\top \tag{4}$$

where $\mathbf{U}_{\bar{d}} \in \mathbb{R}^{\bar{N}_j \times \bar{d}}$ is composed of the first $\bar{d}$ columns of $\mathbf{U}$ and $\boldsymbol{\Sigma}_{\bar{d}}$ is a diagonal matrix consisting of the first (largest) $\bar{d}$ singular values. Then the final node feature matrix is obtained by concatenating embeddings from all scales, i.e.,

$$\mathbf{Z}^{(j)} = \left[ \mathbf{Z}_{\lambda_1}^{(j)}, \ldots, \mathbf{Z}_{\lambda_Q}^{(j)} \right] = \begin{bmatrix} \mathbf{Z}_1^{(j)} \\ \vdots \\ \mathbf{Z}_{N_j}^{(j)} \end{bmatrix} \in \mathbb{R}^{\sum_{i=1}^{N_j} n_i^{(j)} \times Q\bar{d}}, \quad j \in [M] \tag{5}$$

For datasets without node attributes, we generate node attributes using the truncated SVD of the self-looped adjacency matrix, i.e.,

$$\mathbf{X}_i^{(j)} = \mathrm{SVD}\left( \mathbf{A}_i^{(j)} + \mathbf{I}_{n_i^{(j)}} \right) \in \mathbb{R}^{n_i \times \bar{d}} \tag{6}$$

where $\mathrm{SVD}(\cdot)$ returns the singular vectors corresponding to the top-$\bar{d}$ singular values, similar to (4). Then we apply (2), (3), (4), and (5) to $\mathbf{X}_i^{(j)}$ to generate unified node embeddings. For large datasets, the Nyström approximation (Williams & Seeger, 2000) can be employed to accelerate the computation of the kernel matrix and SVD.

### 4.2 DENSITY MAXIMIZATION BASED MEAN ALIGNMENT

In SVD, individual singular vectors have arbitrary signs (Bro et al., 2008). This sign ambiguity may make the embeddings of two similar graphs very different, leading to significant difficulties in both the training and testing stages. Moreover, if two singular values are the same, the order of the corresponding singular vectors cannot be determined, which further increase the difficulty in learning. In

**Algorithm 1** Max-Density Mean Alignment

**Input:** $\boldsymbol{\mu}_1, \boldsymbol{\mu}_2, \ldots, \boldsymbol{\mu}_M; \gamma > 0; \eta > 0; T.$
1: Initialization: $\mathbf{R}_j^{(0)} = \mathbf{I}_{\bar{d}}, \forall j \in [M]$
2: $w_{ij} = \exp(-\gamma(\|\boldsymbol{\mu}_i\|^2 + \|\boldsymbol{\mu}_j\|^2)), (i,j) \in [M] \times [M]$
3: **for** $t = 1$ to $T$ **do**
4: $\quad k_{ij}^{(t-1)} = \exp(2\gamma\langle \mathbf{R}_i^{(t-1)}\boldsymbol{\mu}_i, \mathbf{R}_j^{(t-1)}\boldsymbol{\mu}_j\rangle),$ $\quad (i,j) \in [M] \times [M]$
5: $\quad$ **for** $i = 1$ to $M$ **do**
6: $\quad\quad \mathbf{H}_i^{(t)} = \sum_j w_{ij} k_{ij}^{(t-1)} \mathbf{R}_j^{(t-1)} \boldsymbol{\mu}_j \boldsymbol{\mu}_i^\top + \eta \mathbf{R}_i^{(t-1)}$
7: $\quad\quad$ SVD: $\mathbf{H}_i^{(t)} = \mathbf{U}_i \mathbf{S}_i \mathbf{V}_i^\top$
8: $\quad\quad \mathbf{R}_i^{(t)} = \mathbf{U}_i \mathbf{V}_i^\top$
9: $\quad$ **end for**
10: **end for**
**Output:** $\mathbf{R}_1, \mathbf{R}_2, \ldots, \mathbf{R}_M$

**Algorithm 2** Pre-training model

**Input:** Train datasets: $\mathcal{D} = \{\mathcal{G}_j\}_{j=1}^M$
1: Compute $\{\mathbf{Z}^{(j)}\}_{j=1}^M$ using (4) and (5)
2: $\{\mathbf{R}_j^{\lambda_q}\}_{j=1}^M \leftarrow$ Algorithm 1
3: $\mathbf{Z}_{\lambda_q}^{(j)} \leftarrow \mathbf{Z}_{\lambda_q}^{(j)} \mathbf{R}_j^{(\lambda_q)^\top}, \ q \in [Q], j \in [M]$
4: **repeat**
5: $\quad$ **for** $j = 1$ to $M$ **do**
6: $\quad\quad$ **for** $i = 1$ to $S$ **do**
7: $\quad\quad\quad$ Sample $\{\mathbf{G}_i^{(j)} = (\mathbf{Z}_i^{(j)}, \mathbf{A}_i^{(j)})\}_{i \in \mathcal{B}}$
8: $\quad\quad\quad \mathbf{g}_i^{(j)} \leftarrow F_{\mathcal{W},\mathcal{V},\gamma}(\mathbf{A}_i^{(j)}, \mathbf{Z}_i^{(j)}), i \in \mathcal{B}$
9: $\quad\quad\quad \mathcal{W} \leftarrow \mathcal{W} + \alpha_1 \nabla_\mathcal{W} \mathcal{L}(\mathcal{W}, \mathcal{V}, \gamma)$
$\quad\quad\quad\quad \mathcal{V} \leftarrow \mathcal{V} + \alpha_1 \nabla_\mathcal{V} \mathcal{L}(\mathcal{W}, \mathcal{V}, \gamma)$
$\quad\quad\quad\quad \gamma \leftarrow \gamma + \alpha_2 \nabla_\gamma \mathcal{L}(\mathcal{W}, \mathcal{V}, \gamma)$
10: $\quad\quad$ **end for**
11: $\quad$ **end for**
12: **until** Convergence conditions are met
**Output:** Pretrained model $F_{\mathcal{W},\mathcal{V},\gamma}$

machine learning, to ensure learnability and generalization, we require that the training samples and the testing samples are from the same distribution, or at least, their means are similar. Therefore, we proposed to align the mean embeddings of different graphs via maximizing the density.

Specifically, consider the SVD embeddings of $M \times Q$ graphs generated by the method in Section 4.1, for each $\lambda_q$, we compute the mean vectors of the embedding matrices $\mathbf{Z}_{\lambda_q}^{(j)}$ as $\boldsymbol{\mu}_j = \frac{1}{N_j} \mathbf{Z}_j^\top \mathbf{1}_{\bar{N}_j}$, where $j \in [M]$ and we have dropped the subscript $\lambda_q$ to simplify the notation for the following operations. For each graph $j$, we introduce an orthonormal matrix $\mathbf{R}_j \in \mathbb{R}^{\bar{d} \times \bar{d}}$, which will transform $\boldsymbol{\mu}_j$ as $\mathbf{R}_j \boldsymbol{\mu}_j, j \in [M]$, which means $\mathbf{R}_j \mathbf{Z}_{\lambda_q}^{(j)}$ is equivalent to $\mathbf{Z}_{\lambda_q}^{(j)}$ in preserving the information of $\mathbf{K}_{\lambda_q}^{(i)}$. We align all mean vectors using the corresponding orthonormal matrices by maximizing the density of the mean vectors. The density of each mean vector can be calculated by the kernel density estimation (Parzen, 1962):

$$\hat{p}(\boldsymbol{\mu}) = \frac{1}{M} \sum_{j=1}^M \frac{1}{(2\pi h)^{\bar{d}/2}} \exp\left(-\frac{\|\boldsymbol{\mu} - \boldsymbol{\mu}_j\|^2}{2h}\right) \tag{7}$$

where we use the Gaussian kernel with hyperparameter $h$. Let $\mathcal{R}$ be the set of all orthonormal matrices of size $\bar{d} \times \bar{d}$, i.e., $\mathcal{R} = \{\mathbf{R} \in \mathbb{R}^{\bar{d} \times \bar{d}} : \mathbf{R}^\top \mathbf{R} = \mathbf{I}_{\bar{d}}\}$. Then we maximize the total density of the $M$ mean vectors:

$$\underset{\mathbf{R}_j \in \mathcal{R}, j \in [M]}{\text{maximize}} \ \frac{1}{N} \sum_{i=1}^M \sum_{j=1}^M \frac{1}{(2\pi h)^{\bar{d}/2}} \exp\left(-\frac{\|\mathbf{R}_i \boldsymbol{\mu}_i - \mathbf{R}_j \boldsymbol{\mu}_j\|^2}{2h}\right) \tag{8}$$

Letting $\gamma = \frac{1}{2h}$, (8) is equivalent to the following problem

$$\underset{\mathbf{R}_j \in \mathcal{R}, j \in [M]}{\text{maximize}} \ \frac{1}{M} \sum_{i=1}^M \sum_{j=1}^M \exp\left(-\gamma \|\mathbf{R}_i \boldsymbol{\mu}_i - \mathbf{R}_j \boldsymbol{\mu}_j\|^2\right) \triangleq \mathcal{L}\left(\{\mathbf{R}_j\}_{j=1}^M\right) \tag{9}$$

The optimization is non-trivial due to the orthonormal constraints and the exponential functions. We propose an efficient algorithm in Algorithm 1. The detailed derivation for the algorithm is introduced in Appendix C. Theorem 4.1 provides a convergence guarantee for the optimization.

**Theorem 4.1.** *Let* $\left\{\mathcal{L}\left(\{\mathbf{R}_j^{(t)}\}_{j=1}^M\right)\right\}_t$ *and* $\left\{\{\mathbf{R}_j^{(t)}\}_{j=1}^M\right\}_t$ *be the sequences given by Algorithm 1. Then for any $\eta > 0$, it holds that:*

*(a)* $\left\{\mathcal{L}\left(\{\mathbf{R}_j^{(t)}\}_{j=1}^M\right)\right\}_t$ *is non-decreasing, i.e.,* $\mathcal{L}\left(\{\mathbf{R}_j^{(t)}\}_{j=1}^M\right) \geq \mathcal{L}\left(\{\mathbf{R}_j^{(t-1)}\}_{j=1}^M\right)$;

*(b)* $\left\{\{\mathbf{R}_j^{(t)}\}_{j=1}^M\right\}_t$ *is convergent, i.e.,* $\mathbf{R}_j^{(t)} - \mathbf{R}_j^{(t-1)} \to \mathbf{0}$ *when* $t \to \infty$.

Once $\mathbf{R}_1, \ldots, \mathbf{R}_M$ are optimized, we modify the embeddings of the $M \times Q$ global graphs as

$$\mathbf{Z}_{\lambda_q}^{(j)} \leftarrow \mathbf{Z}_{\lambda_q}^{(j)} \mathbf{R}_j^{(\lambda_q)^\top}, \quad q \in [Q], j \in [M], \tag{10}$$

where $\mathbf{R}_i^{(\lambda_q)}$ denotes the $\mathbf{R}_j$ we obtained for the kernel embeddings given by the $q$-th kernel function. Recalling (5) and using (10), we here obtain the modified embeddings $\mathbf{Z}_1^{(j)}, \ldots, \mathbf{Z}_{N_j}^{(j)}, j \in [M]$. It is worth noting that Algorithm 1 can also be applied to the generated attributes by (6) of graphs without inherent node attributes.

### 4.3 GIN AND GRAPH TRANSFORMER BASED MODEL

To design a universal graph representation model $F$, we incorporate two main components: a GIN module $f$ (Xu et al., 2019) and a graph transformer (GT) Rampášek et al. (2022) module $g$. We build GIN encoder on top of transformer encoder $g_\psi \circ f_\theta(\cdot)$, where $\theta$ and $\psi$ are the parameters. The GIN encoder specializes in learning local representations of the structure of a node's immediate neighborhood, while the transformer computes all pairwise node interactions, enabling global reasoning through attention mechanisms. The node representations of $G_i^{(j)}$ obtained from the model can be formulated as

$$\mathbf{H}_i^{(j)} = g_\psi \circ f_\theta\left(\mathbf{A}_i^{(j)}, \mathbf{Z}_i^{(j)}\right) \,\Big\|\, f_\theta\left(\mathbf{A}_i^{(j)}, \mathbf{Z}_i^{(j)}\right), \quad i \in [N_j], \quad j \in [M]. \tag{11}$$

For convenience, we let $\mathcal{W} = \{\psi, \theta\}$, which is the set of all parameters of the GIN and GT.

### 4.4 REFERENCE DISTRIBUTION BASED GLOBAL GRAPH REPRESENTATION

Current GFMs often use a single graph pooling method to obtain the final graph representation, which collects only first-order statistics and therefore leads to a loss of structural or semantic information. To obtain more informative graph embeddings from node embedding matrices, inspired by Wang & Fan (2024), we adopt a reference learning module.

The reference distribution (RD) module treats the nodes' latent representations of each graph as a discrete distribution and then measures the similarity between the latent graph's distribution and the learnable reference distribution to obtain the graph representation vectors. Hence, it can effectively preserve the information of node embeddings. Suppose we have $R$ reference discrete distributions $\{\mathbf{V}_1, \mathbf{V}_2 \ldots, \mathbf{V}_R\} \triangleq \mathcal{V}$, each $\mathbf{V}_b \in \mathbb{R}^{m \times d}$ can be understood as node embeddings of a virtual graph with $m$ nodes, drawn from one of $R$ different distributions with $m$ nodes. To get the graph representations from node embedding matrix $\mathbf{H}_i^{(j)}$, we measure the similarity between the graph $G_i^{(j)}$ and the reference distributions $\{\mathbf{V}_b\}_{b=1}^R$. Letting $\xi$ be a similarity measure between two distributions, the similarity between the graph $G_i^{(j)}$ and the reference distribution $\mathbf{V}_b$ can be represented as

$$s_{i,l}^{(j)} = \xi\left(\mathbf{H}_i^{(j)}, \mathbf{V}_b\right), \quad b \in [R] \tag{12}$$

We let $\xi$ be the negative kernelized Maximum Mean Discrepancy (MMD) (Gretton et al., 2012) to be the similarity measure $\xi$ and have

$$s_{i,b}^{(j)} = -\mathrm{MMD}\left(\mathbf{H}_i^{(j)}, \mathbf{V}_l\right) = -\Big\| \frac{1}{n_i} \sum_{p=1}^{n_i} \phi\left(\mathbf{h}_p^{(j)}\right) - \frac{1}{m} \sum_{q=1}^{m} \phi\left(\mathbf{v}_q^{(b)}\right) \Big\|_2$$

$$= -\Big[ \frac{1}{n_i^2} \sum_{p,p' \in [n_i]} k\left(\mathbf{h}_p^{(j)}, \mathbf{h}_{p'}^{(j)}\right) + \frac{1}{m^2} \sum_{q,q' \in [m]} k\left(\mathbf{v}_q^{(b)}, \mathbf{v}_{q'}^{(b)}\right) - \frac{2}{mn_i} \sum_{p \in [n_i], q \in [m]} k\left(\mathbf{h}_p^{(j)}, \mathbf{v}_q^{(b)}\right) \Big]^{\frac{1}{2}}$$

$$\tag{13}$$

where $\phi$ is the high-dimensional feature map induced by a kernel function, $\mathbf{h}_p^{(j)}$ is the $p$-th row of $\mathbf{H}_i^{(j)}$, $\mathbf{v}_q^{(b)}$ is the $q$-th row of $\mathbf{V}_b$, and $k(\mathbf{x}, \mathbf{x}') = \exp\left(-\gamma \|\mathbf{x} - \mathbf{x}'\|^2\right)$ is the Gaussian kernel with a learnable parameter $\gamma$. The final graph embedding $\mathbf{g}_i^{(j)} \in \mathbb{R}^r$ of graph $i$ in dataset $j$ combines the similarity vector $\mathbf{s}_i^{(j)}$ with a readout vector $\mathbf{p}_i^{(j)}$, i.e.,

$$\mathbf{g}_i^{(j)} = \mathbf{s}_i^{(j)} \big\| \mathbf{p}_i^{(j)}, \quad i \in [N_j], j \in [M] \tag{14}$$

where $\mathbf{p}_i^{(j)} = \mathrm{READOUT}(\mathbf{H}_i^{(j)})$ is obtained using a graph-level pooling operation.

### 4.5 MODEL PRE-TRAINING

Different datasets $\mathcal{G}_1, \mathcal{G}_2, \ldots, \mathcal{G}_M$ may contain varying numbers of classes. To unify the training framework across different datasets and avoid changing classifiers during training, we adopt the

supervised contrastive loss (SCL) (Oord et al., 2018). Therefore, we minimize the following loss

$$\mathcal{L}_{\text{SCL}}(\mathcal{W}, \mathcal{V}, \gamma) = -\sum_{j=1}^{M} \frac{1}{N_j} \sum_{i=1}^{N_j} \frac{1}{|C(i)|} \sum_{u \in C(i)} \log \frac{\exp\left(\zeta(\mathbf{g}_i^{(j)}, \mathbf{g}_u^{(j)})\right)}{\sum_{v \neq i}^{N_j} \exp\left(\zeta(\mathbf{g}_i^{(j)}, \mathbf{g}_v^{(j)})\right)} \quad (15)$$

where $\mathbf{g}_i^{(j)} = F_{\mathcal{W}, \mathcal{V}, \gamma}(\mathbf{A}_i^{(j)}, \mathbf{Z}_i^{(j)})$, $\zeta(\mathbf{u}, \mathbf{v}) = \frac{\mathbf{u}^\top \mathbf{v}}{\tau \|\mathbf{u}\| \cdot \|\mathbf{v}\|}$, $C(i)$ denotes the set of samples from the same class as $\mathbf{g}_i^{(j)}$, and $\tau$ is a temperature hyperparameter. This objective encourages graphs from the same class to be close in the embedding space while pushing apart samples from different classes. For convenience, we call our method Graph Vectorization Foundation Model (GraphVec-FM).

A key appeal of foundation models lies in their ability to leverage large-scale unlabeled data. GraphVec-FM can also be adapted to unsupervised pre-training scenario. Let $P(i)$ be the set of positive samples of $G_i$ obtained by augmentation, the unsupervised contrastive loss (USL) can be represented as

$$\mathcal{L}_{\text{UCL}}(\mathcal{W}, \mathcal{V}, \gamma) = -\sum_{j=1}^{M} \frac{1}{N_j} \sum_{i=1}^{N_j} \frac{1}{|P(i)|} \sum_{u \in P(i)} \log \frac{\exp\left(\zeta(\mathbf{g}_i^{(j)}, \mathbf{g}_u^{(j)})\right)}{\sum_{v \neq i}^{N_j} \exp\left(\zeta(\mathbf{g}_i^{(j)}, \mathbf{g}_v^{(j)})\right)} \quad (16)$$

### 4.6 Model Testing

When applying the pretrained model to graph-level downstream tasks, the output embeddings generated by the model can be directly utilized as input features for other machine learning models. Specifically, in few-shot graph classification, the mean alignment algorithm need to be performed on both train graphs and test graphs, the detailed formulation and algorithm are in Appendix G.2.

### 4.7 Extension to Zero-shot Graph Classification

Zero-shot learning (ZSL) has been regarded as an essential ability for foundation models. Following the prevalent zero-shot paradigm in graph learning (Li et al., 2024), we leverage language models to embed textual class descriptions into vector representations. The prediction is then performed by computing the similarity between graph embeddings generated by our model and these class embeddings. Concretely, for $K$ target classes, we first obtain class embedding vectors $\{\mathbf{y}_1, \ldots, \mathbf{y}_K\}$. For each graph $G$'s embedding $\mathbf{g}$, we solve the following regression problem for every class $k \in \{1, \ldots, K\}$:

$$\mathbf{W}_k^* = \arg\min_{\mathbf{W}} \|\mathbf{y} - \mathbf{g}\mathbf{W}\|_2^2 + \lambda \|\mathbf{W}\|_F^2 \quad (17)$$

Letting $e_k = \|\mathbf{y} - \mathbf{g}\mathbf{W}_k^*\|_2^2$, we obtain the predicted label for $G$ by $\hat{k} = \arg\min_k e_k$.

### 4.8 Generalization Error Bound

We provide a theoretical guarantee of generalization ability, i.e., the performance on unseen test datasets, of our model. Since the loss defined in (15) cannot intuitively reflect the model performance, we here consider a general metric learning loss $\ell \in [0, 1]$. An example is $\ell(G_u, G_v) = 1 - C_{uv} \cdot \zeta(\mathbf{g}_u, \mathbf{g}_v)$, where $C_{uv} = 1$ if $G_u$ and $G_u$ are in the same class and $C_{uv} = -1$ otherwise.

**Theorem 4.2.** *Denote $\vartheta_1$ the number of layers of each of the $Q$ GINs, $\vartheta_2$ the number of layers of the GT, $\kappa_1$ the MLP depth in each GIN, and $\kappa_2$ the MLP depth in the GT. Let $\mathbf{W}$ be the weight matrix in a layer of the networks. Let $\tilde{\mathbf{Z}}$ be the whole input data matrix of the GINs and denote $\beta = \|\tilde{\mathbf{Z}}\|_F$. Let $\varsigma = \max_{(i,j) \in [N] \times [M]} \|\mathbf{A}_i^{(j)}\|_2$. Suppose $\ell$ is $\tau$-Lipschitz continuous and the attention maps in GT are $\mu$-Lipschitz continuous. Denote $\mathcal{L}(F) = \mathbb{E}_{G, G' \sim \mathbb{G}}[\ell(G, G')]$. Then with probability $1 - \delta$ over the training dataset $\mathcal{D}$, the following inequality holds*

$$\mathcal{L}(F) \leq \frac{1}{MN(N-1)} \sum_{j=1}^{M} \sum_{u \neq v} \ell(G_u^{(j)}, G_v^{(j)}) + \frac{16 + 48\tau L_F \beta Q \bar{d} \sqrt{\ln(2Q\bar{d})} \ln(MN/2)}{MN} + \sqrt{\frac{\ln(1/\delta)}{2MN}}$$

*where $L_F = \left(4\sqrt{\frac{\gamma R}{n}} + \frac{1}{\sqrt{n}}\right) \left(\varsigma^{\vartheta_1} \max_{q \in [Q]} \prod_{j=1}^{\kappa_1} \|\mathbf{W}_j^{GIN_q}\|_2\right) \left(\mu^{\vartheta_2} \prod_j^{\kappa_2} \|\mathbf{W}_j^{GT}\|_2\right)$.*

The theorem shows the impacts of model architecture, input data size, and weight matrices on the generalization ability of our model:

- When the total number of training graphs $MN$ is larger, the bound is tighter, which is further verified by the experiments in Figure 4. Note that if we use the unsupervised contrastive loss to train the model, due to the data augmentation (though the samples are not independent), the generalization could be stronger. That's why in two cases of Table 2, the unsupervised learning outperformed the supervised learning.

- Although $\beta$ often scales with $\sqrt{n}$, we have a factor $\frac{1}{\sqrt{n}}$ in $L_F$. This means that the number of nodes in each graph does not have a significant impact on the generalization, provided that the spectral norms of $\mathbf{A}_i^{(j)}$ increase slowly with $n$. As a result, our model will generalize well to both small graphs (e.g., ENZYMES) and large graphs (e.g., REDDIT), as shown by Tables 1 and 2.

- Since $L_F$ scales with $\mathcal{O}(\sqrt{\gamma R})$, we could use a relatively large $R$ to enrich the final vector representation for each graph, thereby improving the expressiveness. Moreover, $L_F$ is not very sensitive to $\gamma$, which is learned adaptively.

## 5 EXPERIMENTS

### 5.1 FEW-SHOT GRAPH CLASSIFICATION

**Datasets and Baselines** Following (Fu et al., 2025), we use five datasets from TUDataset (Morris et al., 2020), including ENZYMES, DD, NCI1, NCI109, and Mutagencity, to conduct few-shot graph classification experiments. Among the five datasets, we use each of them as the downstream dataset for testing while leveraging the remaining four datasets for pre-training. We evaluate our methods against 4 different pre-training methods, including GraphCL (You et al., 2020), SimGRACE (Xia et al., 2022), GPPT (Sun et al., 2022), and GraphPrompt (Liu et al., 2023b). Each of these methods adopts 7 different tuning mechanisms, including prompt-tuning methods GraphPrompt, ALL-in-one (Sun et al., 2023a), GPF (Fang et al., 2023), GPF-plus (Fang et al., 2023), and solely training classifiers. To further evaluate the transferability of GraphVec-FM in different domains, we conduct more experiments on 4 social network datasets, including COLLAB, REDDIT-BINARY, IMDB-BINARY, IMDB-MULTI and 3 computer vision datasets including Letter-med, COIL-RAG and Cuneiform using GraphVec-FM pre-trained on 5 bio-chemical datasets mentioned above. The results are compared with BRIDGE (Yuan et al., 2025), ProNoG (Yu et al., 2025b) and EdgePrompt+ (Fu et al., 2025). More details about the settings and datasets can be found in Appendix G.3 and G.1.

**Results** The results are reported in Table 1 and Table 2. Our GraphVec-FM achieves the best or most competitive performance among other pre-trained models with different pre-training and tuning strategies. In social network and computer vision datasets that have a significant gap between pre-training datasets in both semantics and structure, our GraphVec-FM consistently outperforms other baselines. This performance demonstrates that our pre-trained model effectively learns generalizable graph embeddings across different domains without relying on delicately designed tuning methods. It also highlights the model's capability to capture features from diverse domains while maintaining a strong generalization ability to new domains. Compared to the supervised pre-trained model, the unsupervised pre-trained model exhibits only a slight decrease in accuracy. The unsupervised GraphVec-FM still outperforms all four baselines on 4 of the 7 datasets, and even surpasses the supervised pretrained GraphVec-FM on IMDB-B and Cuneiform.

In addition, the performance of GraphVec-FM with the mean alignment is better than GraphVec-FM without the mean alignment. This observation validates the effectiveness of our proposed density maximization-based mean alignment algorithm. Meanwhile, we see that the GraphVec-FM without the mean alignment still remains competitive compared to other baseline methods and achieves second second-best performance on average accuracy.

### 5.2 ZERO-SHOT GRAPH CLASSIFICATION

We conduct zero-shot graph classification experiments on 2 social network datasets and 2 computer vision datasets using GraphVec-FM pretrained on 5 biochemical datasets. The results are shown in Table 3. Our methods outperform the other two pretrained GFMs on all 4

Table 1: 50-shot graph classification performance comparison with different pre-train models. We color the **best** and second best models. The compared numbers are from EdgePrompt.

| Pre-training | Tuning Methods | ENZYMES | DD | NCI1 | NCI109 | Mutagenicity | Average |
|---|---|---|---|---|---|---|---|
| GraphCL | Classifier Only | $30.50_{\pm1.16}$ | $62.89_{\pm2.19}$ | $62.49_{\pm1.95}$ | $61.68_{\pm0.93}$ | $66.62_{\pm1.87}$ | 56.84 |
| | GraphPrompt (Liu et al., 2023b) | $27.83_{\pm1.61}$ | $64.33_{\pm1.79}$ | $63.19_{\pm1.71}$ | $62.18_{\pm0.48}$ | $67.62_{\pm0.65}$ | 57.03 |
| | ALL-in-one (Sun et al., 2023a) | $25.92_{\pm0.55}$ | $66.54_{\pm1.82}$ | $57.52_{\pm2.61}$ | $62.74_{\pm0.78}$ | $63.43_{\pm2.53}$ | 55.23 |
| | GPF (Fang et al., 2023) | $30.08_{\pm1.25}$ | $64.54_{\pm2.22}$ | $62.66_{\pm1.83}$ | $62.29_{\pm0.90}$ | $66.54_{\pm1.85}$ | 57.22 |
| | GPF-plus (Fang et al., 2023) | $31.00_{\pm1.50}$ | $67.26_{\pm2.29}$ | $64.56_{\pm1.10}$ | $62.84_{\pm0.22}$ | $66.82_{\pm1.63}$ | 58.50 |
| | EdgePrompt (Fu et al., 2025) | $29.50_{\pm1.57}$ | $64.16_{\pm2.13}$ | $63.05_{\pm2.11}$ | $62.59_{\pm0.93}$ | $66.87_{\pm1.88}$ | 57.23 |
| | EdgePrompt+ (Fu et al., 2025) | $34.00_{\pm1.25}$ | $67.98_{\pm2.05}$ | $66.30_{\pm2.54}$ | $66.52_{\pm0.91}$ | $67.47_{\pm2.37}$ | 60.45 |
| SimGRACE | Classifier Only | $27.07_{\pm1.04}$ | $61.77_{\pm2.40}$ | $61.27_{\pm3.64}$ | $62.12_{\pm1.10}$ | $67.36_{\pm0.71}$ | 55.92 |
| | GraphPrompt (Liu et al., 2023b) | $26.87_{\pm1.47}$ | $62.58_{\pm1.84}$ | $62.45_{\pm1.52}$ | $62.41_{\pm0.69}$ | $68.03_{\pm0.78}$ | 56.47 |
| | ALL-in-one | $25.73_{\pm1.18}$ | $65.16_{\pm1.47}$ | $58.52_{\pm1.59}$ | $62.01_{\pm0.66}$ | $64.43_{\pm1.00}$ | 55.17 |
| | GPF (Fang et al., 2023) | $28.53_{\pm1.76}$ | $65.64_{\pm0.70}$ | $61.45_{\pm3.13}$ | $61.90_{\pm1.26}$ | $67.19_{\pm0.74}$ | 56.94 |
| | GPF-plus (Fang et al., 2023) | $27.33_{\pm2.01}$ | $67.20_{\pm1.56}$ | $61.61_{\pm2.89}$ | $62.84_{\pm0.23}$ | $67.69_{\pm0.64}$ | 57.33 |
| | EdgePrompt (Fu et al., 2025) | $29.33_{\pm2.30}$ | $63.97_{\pm2.14}$ | $62.02_{\pm3.02}$ | $62.02_{\pm1.03}$ | $67.55_{\pm0.85}$ | 56.98 |
| | EdgePrompt+ (Fu et al., 2025) | $32.67_{\pm2.53}$ | $67.72_{\pm1.62}$ | $67.07_{\pm1.96}$ | $66.53_{\pm1.30}$ | $68.31_{\pm1.36}$ | 60.46 |
| EP-GPPT | Classifier Only | $29.08_{\pm1.35}$ | $62.12_{\pm2.82}$ | $56.85_{\pm4.35}$ | $62.27_{\pm0.78}$ | $66.30_{\pm1.78}$ | 55.32 |
| | GraphPrompt (Liu et al., 2023b) | $26.67_{\pm1.60}$ | $61.61_{\pm1.91}$ | $58.77_{\pm0.97}$ | $62.16_{\pm0.89}$ | $66.37_{\pm1.17}$ | 55.12 |
| | ALL-in-one (Fang et al., 2023) | $24.92_{\pm1.33}$ | $63.61_{\pm2.12}$ | $59.14_{\pm2.12}$ | $59.70_{\pm1.37}$ | $64.86_{\pm1.60}$ | 54.45 |
| | GPF (Fang et al., 2023) | $28.33_{\pm1.73}$ | $63.48_{\pm2.08}$ | $58.14_{\pm4.16}$ | $62.52_{\pm1.39}$ | $66.10_{\pm0.96}$ | 55.71 |
| | GPF-plus (Fang et al., 2023) | $29.25_{\pm1.30}$ | $66.92_{\pm2.34}$ | $62.93_{\pm3.23}$ | $64.13_{\pm1.42}$ | $67.57_{\pm1.45}$ | 58.16 |
| | EdgePrompt (Fu et al., 2025) | $28.33_{\pm3.41}$ | $64.03_{\pm2.26}$ | $59.85_{\pm3.15}$ | $62.98_{\pm1.44}$ | $66.36_{\pm1.22}$ | 56.31 |
| | EdgePrompt+ (Fu et al., 2025) | $32.75_{\pm2.26}$ | $66.16_{\pm1.60}$ | $63.58_{\pm2.07}$ | $65.15_{\pm1.60}$ | $68.35_{\pm1.57}$ | 59.20 |
| EP-GraphPrompt | Classifier Only | $31.33_{\pm3.22}$ | $62.58_{\pm2.40}$ | $62.09_{\pm2.31}$ | $60.19_{\pm1.71}$ | $65.13_{\pm0.81}$ | 55.32 |
| | GraphPrompt (Liu et al., 2023b) | $30.20_{\pm1.93}$ | $64.72_{\pm1.98}$ | $62.57_{\pm1.45}$ | $62.32_{\pm0.95}$ | $65.85_{\pm0.65}$ | 57.13 |
| | ALL-in-one (Fang et al., 2023) | $29.07_{\pm1.16}$ | $65.60_{\pm2.38}$ | $58.67_{\pm2.42}$ | $57.69_{\pm1.08}$ | $64.66_{\pm0.76}$ | 55.14 |
| | GPF (Fang et al., 2023) | $30.93_{\pm1.76}$ | $66.21_{\pm1.66}$ | $61.80_{\pm2.78}$ | $62.27_{\pm1.18}$ | $65.61_{\pm0.59}$ | 57.36 |
| | GPF-plus (Fang et al., 2023) | $30.67_{\pm3.06}$ | $67.50_{\pm2.45}$ | $62.59_{\pm2.09}$ | $61.98_{\pm1.60}$ | $65.51_{\pm1.10}$ | 57.65 |
| | EdgePrompt (Fu et al., 2025) | $30.80_{\pm2.09}$ | $65.87_{\pm1.35}$ | $61.75_{\pm2.49}$ | $62.33_{\pm1.65}$ | $65.77_{\pm0.90}$ | 57.30 |
| | EdgePrompt+ (Fu et al., 2025) | $33.27_{\pm2.71}$ | $67.47_{\pm2.14}$ | $65.06_{\pm1.84}$ | $64.64_{\pm1.57}$ | $66.42_{\pm1.31}$ | 59.37 |
| BRIDGE (Yuan et al., 2025) | | $36.67_{\pm5.96}$ | $64.95_{\pm3.38}$ | $63.50_{\pm2.27}$ | $61.78_{\pm1.63}$ | $65.12_{\pm2.83}$ | 58.40 |
| **GraphVec-FM** | | $\mathbf{51.60}_{\pm2.75}$ | $\mathbf{74.58}_{\pm1.43}$ | $\mathbf{67.06}_{\pm0.58}$ | $\mathbf{66.78}_{\pm2.43}$ | $\mathbf{69.38}_{\pm1.86}$ | **65.88** |
| **GraphVec-FM** w/o mean alignment | | $49.33_{\pm1.48}$ | $73.14_{\pm1.16}$ | $65.80_{\pm1.40}$ | $65.21_{\pm2.05}$ | $67.00_{\pm1.31}$ | 64.09 |

Table 2: Few-shot graph classification results on social network and computer vision datasets. 5-shot and 1-shot settings are adopted on COIL-RAG and Cuneiform, respectively due to a lack of enough samples in some classes.

| Dataset | COLLAB 50-shot | REDDIT-B 50-shot | IMDB-B 50-shot | IMDB-M 50-shot | Letter-med 50-shot | COIL-RAG 5-shot | Cuneiform 1-shot |
|---|---|---|---|---|---|---|---|
| BRIDGE (Yuan et al., 2025) | $54.52_{\pm3.73}$ | $57.80_{\pm8.79}$ | $50.20_{\pm6.27}$ | $36.6_{\pm5.48}$ | $35.40_{\pm2.09}$ | $27.02_{\pm6.90}$ | $23.07_{\pm9.42}$ |
| ProNoG (Yu et al., 2025b) | $46.88_{\pm3.14}$ | $74.33_{\pm2.05}$ | $60.8_{\pm5.19}$ | $40.53_{\pm0.66}$ | $56.98_{\pm5.83}$ | $34.97_{\pm7.62}$ | $10.00_{\pm7.92}$ |
| EdgePrompt+(Fu et al., 2025) | $68.76_{\pm1.60}$ | $74.60_{\pm1.60}$ | $\mathbf{71.17}_{\pm1.07}$ | $46.60_{\pm0.50}$ | $74.66_{\pm1.69}$ | $5.60_{\pm0.21}$ | $18.22_{\pm0.72}$ |
| GraphVec-FM | $\mathbf{69.79}_{\pm2.99}$ | $\mathbf{77.79}_{\pm2.19}$ | $\underline{68.87}_{\pm4.06}$ | $\underline{46.76}_{\pm0.99}$ | $\mathbf{84.27}_{\pm1.10}$ | $\mathbf{73.69}_{\pm1.43}$ | $\underline{45.32}_{\pm3.63}$ |
| Unsupervised GraphVec-FM | $68.72_{\pm1.87}$ | $\underline{77.47}_{\pm4.36}$ | $65.28_{\pm2.28}$ | $\mathbf{47.7}_{\pm1.00}$ | $68.40_{\pm4.49}$ | $\underline{59.38}_{\pm1.14}$ | $\mathbf{50.09}_{\pm6.11}$ |

datasets, especially on two datasets from computer vision. Without any fine-tuning, the results validate the generalization ability of GraphVec-FM and transferability of the graph embeddings obtained from the pretrained model on out-of-distribution domains and unseen structures.

## 5.3 GRAPH CLUSTERING

To further validate the superiority of our proposed model on graph-level tasks, we conducted experiments on graph clustering: applying spectral clustering (Ng et al., 2001) to the graph representations produced by the pretrained model. As shown in Table 4, our methods perform best.

Table 3: Zero-shot graph classification results on social network datasets and computer vision datasets.

| Dataset | COLLAB | IMDB-B | Letter-med | COIL-RAG |
|---|---|---|---|---|
| ProNoG(Yu et al., 2025b) | 62.20 | 65.00 | 41.33 | 25.02 |
| EdgePrompt+(Fu et al., 2025) | 56.74 | 54.70 | 46.80 | 33.87 |
| GraphVec-FM | **66.24** | **69.19** | **78.08** | **58.35** |

**More Results** The comparison of the transductive and inductive settings of global graph construction, 1/5/10/20-shot learning, the impact of the number of pre-training datasets, more graph clustering results, an ablation study on the reference layer, robustness evaluation on noisy input graphs and time and memory consumption are in Appendices H.1, H.2, H.3, H.4, H.6, H.7, and H.8 respectively.

## 6 CONCLUSIONS

This paper presented a graph foundation model trained on graph datasets from diverse domains. The model aims to represent graphs as vectors that are effective in graph-level downstream tasks. We in-

Table 4: Graph clustering performance on ENZYMES, NCI1, COLLAB, REDDIT-BINARY, REDDIT-MULTI. The comparison numbers are from AMGC (Yang et al., 2025).

| Method | ENZYMES | | NCI1 | | COLLAB | | REDDIT-BINARY | | REDDIT-MULTI | |
|---|---|---|---|---|---|---|---|---|---|---|
| | ACC | NMI | ACC | NMI | ACC | NMI | ACC | NMI | ACC | NMI |
| RW +SC | $17.0_{\pm 0.0}$ | $0.7_{\pm 0.0}$ | N/A | N/A | N/A | N/A | N/A | N/A | N/A | N/A |
| WL +SC | $21.0_{\pm 0.0}$ | $3.1_{\pm 0.0}$ | $50.1_{\pm 0.0}$ | $0.0_{\pm 0.0}$ | $53.2_{\pm 0.0}$ | $2.0_{\pm 0.0}$ | $57.6_{\pm 0.0}$ | $9.0_{\pm 0.0}$ | $18.7_{\pm 0.0}$ | $9.0_{\pm 0.0}$ |
| WL-OA +SC | $20.0_{\pm 0.0}$ | $1.4_{\pm 0.0}$ | $53.2_{\pm 0.0}$ | $0.9_{\pm 0.0}$ | $54.2_{\pm 0.0}$ | $0.2_{\pm 0.0}$ | $53.8_{\pm 0.0}$ | $5.6_{\pm 0.0}$ | $20.9_{\pm 0.0}$ | $9.6_{\pm 0.0}$ |
| SP +SC | $22.0_{\pm 0.0}$ | $2.6_{\pm 0.0}$ | $50.1_{\pm 0.0}$ | $0.1_{\pm 0.0}$ | $48.7_{\pm 0.0}$ | $17.9_{\pm 0.0}$ | $57.8_{\pm 0.0}$ | $2.2_{\pm 0.0}$ | $20.3_{\pm 0.0}$ | $6.1_{\pm 0.0}$ |
| LT +SC | $17.0_{\pm 0.0}$ | $0.4_{\pm 0.0}$ | N/A | N/A | N/A | N/A | N/A | N/A | N/A | N/A |
| GK +SC | $17.1_{\pm 0.1}$ | $0.8_{\pm 0.3}$ | $52.9_{\pm 0.9}$ | $0.7_{\pm 1.4}$ | $56.8_{\pm 1.4}$ | $15.5_{\pm 1.9}$ | $50.3_{\pm 0.3}$ | $0.2_{\pm 0.1}$ | $18.7_{\pm 0.9}$ | $7.2_{\pm 0.3}$ |
| InfoGraph +KM | $22.1_{\pm 1.0}$ | $2.4_{\pm 0.5}$ | $54.1_{\pm 2.2}$ | $1.3_{\pm 1.1}$ | $59.6_{\pm 1.8}$ | $14.4_{\pm 3.0}$ | $51.3_{\pm 2.1}$ | $2.3_{\pm 0.4}$ | $20.3_{\pm 0.9}$ | $0.5_{\pm 0.2}$ |
| InfoGraph +SC | $23.8_{\pm 0.5}$ | $4.6_{\pm 0.7}$ | $54.9_{\pm 1.7}$ | $0.9_{\pm 0.6}$ | $60.9_{\pm 2.5}$ | $15.4_{\pm 3.3}$ | $50.8_{\pm 1.3}$ | $1.6_{\pm 0.6}$ | $24.7_{\pm 1.3}$ | $4.8_{\pm 0.6}$ |
| GraphCL +KM | $21.5_{\pm 0.2}$ | $1.6_{\pm 0.1}$ | $55.4_{\pm 1.7}$ | $0.5_{\pm 0.3}$ | $58.0_{\pm 1.2}$ | $17.8_{\pm 2.0}$ | $51.9_{\pm 3.3}$ | $3.4_{\pm 1.2}$ | $25.3_{\pm 0.9}$ | $5.3_{\pm 0.3}$ |
| GraphCL +SC | $25.3_{\pm 0.3}$ | $4.8_{\pm 0.4}$ | $50.8_{\pm 1.6}$ | $0.6_{\pm 0.6}$ | $57.8_{\pm 0.6}$ | $17.0_{\pm 1.3}$ | $55.9_{\pm 2.1}$ | $3.2_{\pm 1.0}$ | $27.3_{\pm 1.3}$ | $5.4_{\pm 0.8}$ |
| JOAO + KM | $21.7_{\pm 0.4}$ | $4.9_{\pm 0.4}$ | $51.1_{\pm 0.4}$ | $0.4_{\pm 0.2}$ | $58.3_{\pm 1.5}$ | $18.7_{\pm 2.6}$ | $54.3_{\pm 2.9}$ | $4.2_{\pm 1.8}$ | $26.6_{\pm 0.6}$ | $3.6_{\pm 1.2}$ |
| JOAO + SC | $24.4_{\pm 1.4}$ | $3.2_{\pm 0.7}$ | $51.5_{\pm 3.0}$ | $0.9_{\pm 1.2}$ | $58.2_{\pm 0.9}$ | $17.1_{\pm 2.1}$ | $55.9_{\pm 1.2}$ | $6.7_{\pm 2.0}$ | $25.6_{\pm 0.6}$ | $2.5_{\pm 0.2}$ |
| GLCC(Ju et al., 2023) | $24.4_{\pm 1.4}$ | $3.2_{\pm 0.7}$ | $60.9_{\pm 2.3}$ | $5.3_{\pm 1.9}$ | $60.3_{\pm 0.6}$ | $18.2_{\pm 1.3}$ | $67.6_{\pm 3.4}$ | $9.2_{\pm 2.6}$ | $32.4_{\pm 2.1}$ | $11.8_{\pm 1.3}$ |
| AMGC(Yang et al., 2025) | $26.7_{\pm 2.0}$ | $5.2_{\pm 1.3}$ | $62.7_{\pm 3.0}$ | $6.4_{\pm 1.9}$ | $61.2_{\pm 1.0}$ | $20.5_{\pm 1.6}$ | $64.3_{\pm 1.9}$ | $12.1_{\pm 3.3}$ | $35.5_{\pm 2.3}$ | $16.1_{\pm 0.9}$ |
| GraphVec-FM | $29.1_{\pm 0.4}$ | $7.7_{\pm 0.3}$ | $64.8_{\pm 0.0}$ | $6.5_{\pm 0.0}$ | $61.8_{\pm 0.0}$ | $21.2_{\pm 0.0}$ | $71.6_{\pm 0.0}$ | $20.7_{\pm 0.0}$ | $40.0_{\pm 0.2}$ | $17.3_{\pm 0.1}$ |

troduced a multi-graph construction method to generate consistent node embeddings across different datasets and a reference distribution module to effectively utilize the information of node embeddings. The experiments of few-shot/zero-shot graph classification and graph clustering demonstrated the superiority of our method over the competitors.

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

# A NOTATIONS

| Symbol | Meaning | Symbol | Meaning |
|---|---|---|---|
| $x$ | a real number | $\mathbf{x}$ | a vector |
| $\mathbf{X}$ | a matrix | $\mathbf{I}_n$ | identity matrix of size $n \times n$ |
| $G$ | a graph | $\mathbf{g}$ | vector representation of $G$ |
| $\mathcal{G}$ | a set of graphs | $\mathcal{D}$ | a dataset |
| $\|\mathbf{x}\|$ | the Euclidean norm of $\mathbf{x}$ | $\|\mathbf{x}\|_1$ | the $\ell_1$ norm of $\mathbf{x}$ |
| $[M]$ | the set $\{1, 2, \dots, M\}$ | $\mathbf{X}\|\mathbf{Y}$ or $[\mathbf{X}, \mathbf{Y}]$ | vertical concatenation |
| $\|\mathbf{X}\|_F$ | Frobenius norm of matrix | $\|\mathbf{X}\|_2$ | spectral norm of matrix |

Table 5: Notations

# B MORE ABOUT RELATED WORK

**Language Model-Free GFMs** Many studies have explored training GFMs using the "pre-train and adaptation" paradigm, leveraging message-passing-based or transformer-based GNNs as backbones. These approaches typically employ contrastive or generative self-supervised learning for pretraining, followed by fine-tuning a subset of model parameters to adapt to downstream tasks or datasets (Liu et al., 2025). Contrastive methods (Qiu et al., 2020; Sun et al., 2019; Veličković et al., 2019; Xia et al., 2022) typically aim to produce generalized graph representations through maximizing the agreement between different augmentations of the same instance. For example, GraphCL (You et al., 2020) designs four types of graph data augmentations to learn invariant representations under specialized perturbations. GCOPE (Zhao et al., 2024a) employs a graph contrastive learning framework and introduces coordinators which are some virtual nodes that function as dynamic bridges between disparate graph datasets. Focused on node-level tasks, it effectively mitigates negative transfer effects when pretraining graph models on cross-domain datasets. In the meantime, generative methods pre-train GNNs through graph reconstruction or property prediction. For instance, GraphMAEs (Hou et al., 2022; 2023) employed the reconstruction of features with masking strategies. Recently, graph prompt tuning methods (Sun et al., 2022; Fang et al., 2023; Sun et al., 2023a; Liu et al., 2023b) have been proposed as an adaptation mechanism to bridge the gap between pretraining tasks and downstream tasks. GraphPrompt (Liu et al., 2023b) converts the pre-training task and downstream tasks to follow the same template based on subgraph similarity and uses learnable prompt vectors to implement different aggregation schemes for readout in different downstream tasks. EdgePrompt (Fu et al., 2025) manipulates input graphs by learning prompt vectors for edges and incorporates the edge prompts through message passing in the pre-trained GNN models.

**LLM-based GFMs** These models utilize the strong capacity of large language models to conduct graph analysis. For instance, GraphQA (Fatemi et al., 2023) converts graph connectivity into textual descriptions and uses LLMs to answer graph reasoning questions. By enriching these prompts with domain-specific context, GraphQA can effectively learn cross-domain structural representations, essentially serving as a structural GFM. Similar approaches include NLGraph (Wang et al., 2024), which tackles tasks like shortest path finding by translating graphs into text, demonstrating another viable pathway for unified structure learning. For unifying node feature representations, the One For All (OFA) framework (Liu et al., 2023a) offers an innovative solution. It aggregates diverse graph datasets into a unified text-attributed graph (TAG) format, then leverages LLMs to jointly learn feature representations that transcend domain boundaries. This approach effectively bridges the gap between heterogeneous graph data sources.

**GFM for Graph-Level Tasks** As mentioned before, most of the existing GFMs are designed for node-level tasks. There are a few studies that focus on graph-level tasks across domains. For instance, Chauhan et al. tries to pretrain GNNs on certain classes of a dataset and conduct few-shot classification on the remaining classes within the same dataset. Hassani (2022) adopts a meta-learning approach to learn model initialization for few-shot graph classification. These graph-level models are usually small and not general. Some GFMs can be adapted to graph-level tasks. For instance, GraphPrompt (Liu et al., 2023b), GraphPrompt+ (Yu et al., 2024a), and EdgePrompt (Fu

et al., 2025) use learnable prompts to adjust graph-level pooling for obtaining domain-adaptive graph embeddings. Other works such as SAMGPT (Yu et al., 2025a), ProNoG (Yu et al., 2025b) and MultiGPrompt (Yu et al., 2024b) are also designed to effectively perform graph classification, but they mainly build on well-designed node embedding and use simple global pooling to apply the model to graph-level tasks.

## C    DERIVATION OF ALGORITHM 1

Recall that we want to solve

$$\underset{\mathbf{R}_j \in \mathcal{R}, j \in [M]}{\text{maximize}} \ \frac{1}{M} \sum_{i=1}^{M} \sum_{j=1}^{M} \exp\left(-\gamma \|\mathbf{R}_i \boldsymbol{\mu}_i - \mathbf{R}_j \boldsymbol{\mu}_j\|^2\right) \triangleq \mathcal{L}\left(\{\mathbf{R}_j\}_{j=1}^{M}\right) \tag{18}$$

For convenience, we let $R := \{\mathbf{R}_j \in \mathcal{R}\}_{i \in [M]}$. It follows that

$$\underset{R}{\text{maximize}} \ \sum_i \sum_j \exp\left(-\gamma \|\mathbf{R}_i \boldsymbol{\mu}_i - \mathbf{R}_j \boldsymbol{\mu}_j\|^2\right)$$

$$= \underset{R}{\text{maximize}} \ \sum_i \sum_j \exp\left(-\gamma \left(\|\boldsymbol{\mu}_i\|^2 + \|\boldsymbol{\mu}_j\|^2\right)\right) \exp(2\gamma \langle \mathbf{R}_i \boldsymbol{\mu}_i, \mathbf{R}_j \boldsymbol{\mu}_j \rangle) \tag{19}$$

$$= \underset{R}{\text{maximize}} \ \sum_i \sum_j w_{ij} \exp(2\gamma \langle \mathbf{R}_i \boldsymbol{\mu}_i, \mathbf{R}_j \boldsymbol{\mu}_j \rangle) \triangleq L(R)$$

where $w_{ij} := \exp\left(-\gamma \left(\|\boldsymbol{\mu}_i\|^2 + \|\boldsymbol{\mu}_j\|^2\right)\right)$. Let $R^{(t-1)}$ be the decision variables at iteration $t-1$. We introduce the first-order approximation of $L(R)$ at $R^{(t-1)}$ as follows:

$$\hat{L}(R) := \sum_i \sum_j w_{ij} \exp\left(2\gamma \langle \mathbf{R}_i^{(t-1)} \boldsymbol{\mu}_i, \mathbf{R}_j^{(t-1)} \boldsymbol{\mu}_j \rangle\right)$$

$$+ \sum_i \left\langle 4\gamma \sum_j w_{ij} \exp\left(2\gamma \langle \mathbf{R}_i^{(t-1)} \boldsymbol{\mu}_i, \mathbf{R}_j^{(t-1)} \boldsymbol{\mu}_j \rangle\right) \mathbf{R}_j^{(t-1)} \boldsymbol{\mu}_j \boldsymbol{\mu}_i^\top, \mathbf{R}_i - \mathbf{R}_i^{(t-1)} \right\rangle \tag{20}$$

$$= \sum_i \sum_j w_{ij} k_{ij}^{(t-1)} + 4\gamma \sum_i \left\langle \sum_j w_{ij} k_{ij}^{(t-1)} \mathbf{R}_j^{(t-1)} \boldsymbol{\mu}_j \boldsymbol{\mu}_i^\top, \mathbf{R}_i - \mathbf{R}_i^{(t-1)} \right\rangle$$

where $k_{ij}^{(t-1)} := \exp\left(2\gamma \langle \mathbf{R}_i^{(t-1)} \boldsymbol{\mu}_i, \mathbf{R}_j^{(t-1)} \boldsymbol{\mu}_j \rangle\right)$. Since $L(R)$ is a convex function, it follows that

$$L(R) \geq \hat{L}(R) \tag{21}$$

It follows that

$$L(R) \geq \hat{L}(R) - 2\gamma\eta \sum_i \|\mathbf{R}_i - \mathbf{R}_i^{(t-1)}\|_F^2 \triangleq \bar{L}(R) \tag{22}$$

It means $\bar{L}(R)$ is a lower bound of $L(R)$. Therefore, instead of $L(R)$, we propose to maximize $\bar{L}(R)$:

$$\underset{R}{\text{maximize}} \ \sum_i \sum_j w_{ij} k_{ij}^{(t-1)} + 4\gamma \sum_i \left\langle \sum_j w_{ij} k_{ij}^{(t-1)} \mathbf{R}_j^{(t-1)} \boldsymbol{\mu}_j \boldsymbol{\mu}_i^\top, \mathbf{R}_i - \mathbf{R}_i^{(t-1)} \right\rangle$$

$$- 2\gamma\eta \sum_i \|\mathbf{R}_i - \mathbf{R}_i^{(t-1)}\|_F^2$$

$$= \underset{R}{\text{maximize}} \ \sum_i \left\langle 4\gamma \sum_j w_{ij} k_{ij}^{(t-1)} \mathbf{R}_j^{(t-1)} \boldsymbol{\mu}_j \boldsymbol{\mu}_i^\top, \mathbf{R}_i - \mathbf{R}_i^{(t-1)} \right\rangle + 4\gamma\eta \sum_i \left\langle \mathbf{R}_i^{(t-1)}, \mathbf{R}_i \right\rangle \tag{23}$$

$$= \underset{R}{\text{maximize}} \ \sum_i \left\langle \sum_j w_{ij} k_{ij}^{(t-1)} \mathbf{R}_j^{(t-1)} \boldsymbol{\mu}_j \boldsymbol{\mu}_i^\top + \eta \mathbf{R}_i^{(t-1)}, \mathbf{R}_i - \mathbf{R}_i^{(t-1)} \right\rangle$$

$$= \underset{R}{\text{maximize}} \ \sum_i \left\langle \mathbf{H}_i^t, \mathbf{R}_i \right\rangle$$

where $\mathbf{H}_i^{(t)} := \sum_j w_{ij} k_{ij}^{(t-1)} \mathbf{R}_j^{(t-1)} \boldsymbol{\mu}_j \boldsymbol{\mu}_i^\top + \eta \mathbf{R}_i^{(t-1)}$. This is the well-known orthogonal Procrustes problem Schönemann (1966). The optimal solution for each $\mathbf{R}_i$ is

$$\mathbf{R}_i^t = \mathbf{U}_i \mathbf{V}_i^\top \tag{24}$$

where $\mathbf{U}_i$ and $\mathbf{V}_i$ are from the SVD of $\mathbf{H}_i^t$, i.e., $\mathbf{H}_i^t = \mathbf{U}_i \mathbf{S}_i \mathbf{V}_i^\top$.

## D  PROOF FOR THEOREM 4.1

*Proof.* According to the definition of $\bar{L}(R)$, we have

$$\bar{L}(R^{(t-1)}) = L(R^{(t-1)}) \tag{25}$$

At iteration $t$, since (24) provides the optimal solution to maximize$_R$ $\bar{L}(R)$, we have

$$\bar{L}(R^{(t)}) \geq \bar{L}(R) \tag{26}$$

which holds for any $R$. Letting $R = R^{(t-1)}$ and using (25), we have

$$\hat{L}(R^{(t)}) - 2\gamma\eta \sum_i \|\mathbf{R}_i - \mathbf{R}_i^{(t-1)}\|_F^2 \geq \bar{L}(R^{(t-1)}) = L(R^{(t-1)}) \tag{27}$$

which means

$$\hat{L}(R^{(t)}) \geq L(R^{(t-1)}) + 2\gamma\eta \sum_i \|\mathbf{R}_i - \mathbf{R}_i^{(t-1)}\|_F^2 \tag{28}$$

According to (21), we have

$$\hat{L}(R^{(t)}) \leq L(R^{(t)}) \tag{29}$$

Now combing (28) and (29), we arrive at

$$L(R^{(t)}) \geq L(R^{(t-1)}) + 2\gamma\eta \sum_i \|\mathbf{R}_i - \mathbf{R}_i^{(t-1)}\|_F^2 \tag{30}$$

This means the objective function is non-increasing. Summing up both sides of (30) from 1 to $t$, we obtain

$$L(R^{(t)}) \geq L(R^{(0)}) + 2\gamma\eta \sum_i \|\mathbf{R}_i - \mathbf{R}_i^{(t-1)}\|_F^2 \tag{31}$$

Since $L(R^{(t)}) < \infty$ and $L(R^{(0)}) > -\infty$, we obtain $2\gamma\eta \sum_i \|\mathbf{R}_i - \mathbf{R}_i^{(t-1)}\|_F^2 < \infty$. As $\gamma < 0$ and $\rho > 0$, we conclude that $\sum_i \|\mathbf{R}_i - \mathbf{R}_i^{(t-1)}\|_F^2 < \infty$. That means, when $t \to \infty$, $\mathbf{R}_i - \mathbf{R}_i^{(t-1)} \to \mathbf{0}$. The $\mathbf{R}_i^{(t)}$ is convergent, $\forall i \in [M]$. $\qquad\square$

## E  PROOF FOR THEOREM 4.2

Since (15) does not explicitly show the error related to classification or metric learning, here we consider the following pair-wise loss function $\ell$ instead. An example is as

$$\ell(G_u, G_v) = 1 - C_{uv} \cdot \zeta(\mathbf{g}_u, \mathbf{g}_v) \tag{32}$$

where $\zeta(\mathbf{g}_u, \mathbf{g}_v) = \frac{\mathbf{g}_u^\top \mathbf{g}_v}{\|\mathbf{g}_u\|\|\mathbf{g}_v\|}$ and $C_{uv} = 1$ if $G_u$ and $G_u$ are in the same class and $C_{uv} = -1$ if they are in different classes. Note that $\mathbf{g} = F(G)$, where $F \in \mathcal{F}$. The empirical risk is

$$\hat{\mathcal{L}}_\mathcal{D}(F) = \frac{1}{M} \sum_{j=1}^M \frac{1}{N(N-1)} \sum_{u \neq v} \ell(G_u^{(j)}, G_v^{(j)}) \triangleq \frac{1}{M} \sum_{j=1}^M \bar{\mathcal{L}}_{G_j}(F) \tag{33}$$

where we have assumed $N_1 = N_2 = \cdots = N_M = N$ for convenience and $\bar{\mathcal{L}}_{G_j}(F) = \frac{1}{N(N-1)} \sum_{u \neq v} \ell(G_u^{(j)}, G_v^{(j)})$. The true risk is

$$\mathcal{L}(F) = \mathbb{E}_{G,G \sim \mathbb{G}}[\ell(G, G')] \tag{34}$$

We would like to bound

$$\sup_{F\in\mathcal{F}}\{\hat{\mathcal{L}}_{\mathcal{D}}(F)-\mathcal{L}(F)\} \tag{35}$$

For any $\mathcal{D} = \{\mathcal{G}_1,\dots,\mathcal{G}_j,\dots,\mathcal{G}_M\}$ and $\tilde{\mathcal{D}} = \{\mathcal{G}_1,\dots,\tilde{\mathcal{G}}_j,\dots,\mathcal{G}_M\}$, where $\mathcal{G}_j = \{G_1^{(j)},\dots,G_i^{(j)},\dots,G_N^{(j)}\}$ and $\tilde{\mathcal{G}}_j = \{G_1^{(j)},\dots,\tilde{G}_i^{(j)},\dots,G_N^{(j)}\}$, we have

$$
\begin{aligned}
&\left| \sup_{F\in\mathcal{F}}\{\hat{\mathcal{L}}_{\mathcal{D}}(F)-\mathcal{L}(F)\} - \sup_{F\in\mathcal{F}}\{\hat{\mathcal{L}}_{\tilde{\mathcal{D}}}(F)-\mathcal{L}(F)\} \right|\\
&\leq \sup_{F\in\mathcal{F}}\left| \hat{\mathcal{L}}_{\mathcal{D}}(F)-\hat{\mathcal{L}}_{\tilde{\mathcal{D}}}(F) \right|\\
&= \sup_{F\in\mathcal{F}}\left| \frac{1}{MN(N-1)}\left( \sum_{\mathcal{G}_j:u\neq v}\ell(G_u^{(j)},G_v^{(j)}) - \sum_{\tilde{\mathcal{G}}_j:u\neq v}\ell(G_u^{(j)},G_v^{(j)}) \right) \right|\\
&\leq \sup_{F\in\mathcal{F}}\left| \frac{1}{MN(N-1)}\left( \sum_{v\neq i}\left( \ell(G_i^{(j)},G_v^{(j)}) - \ell(\tilde{G}_i^{(j)},G_v^{(j)}) \right) \right) \right|\\
&\leq \sup_{F\in\mathcal{F}}\left| \frac{1}{MN(N-1)}\left( \sum_{v\neq i}\left| \ell(G_i^{(j)},G_v^{(j)}) - \ell(\tilde{G}_i^{(j)},G_v^{(j)}) \right| \right) \right|\\
&\leq \frac{1}{MN}
\end{aligned}
\tag{36}
$$

where the last inequality holds due to that $0\leq\ell\leq 1$. Applying the McDiarmid's inequality (Lemma E.1) to $\sup_{F\in\mathcal{F}}\{\hat{\mathcal{L}}_{\mathcal{D}}(F)-\mathcal{L}(F)\}$, with probability at least $1-\delta$, we have

$$\sup_{F\in\mathcal{F}}\{\hat{\mathcal{L}}_{\mathcal{D}}(F)-\mathcal{L}(F)\} \leq \mathbb{E}_{\mathcal{D}}\left( \sup_{F\in\mathcal{F}}\{\hat{\mathcal{L}}_{\mathcal{D}}(F)-\mathcal{L}(F)\} \right) + \sqrt{\frac{\ln(1/\delta)}{2MN}} \tag{37}$$

For convenience, we let $\bar{\ell}(G_u,G_v) = \ell(G_u,G_v) - \mathcal{L}(F).$, we have the following derivation

$$
\begin{aligned}
&\mathbb{E}_{\mathcal{D}}\left( \sup_{F\in\mathcal{F}}\frac{1}{M}\sum_{j=1}^{M}\frac{1}{N(N-1)}\sum_{u\neq v}\bar{\ell}(G_u^{(j)},G_v(j)) \right)\\
&=\mathbb{E}_{\mathcal{D}}\left( \sup_{F\in\mathcal{F}}\frac{1}{M}\sum_{j=1}^{M}\frac{1}{N!}\sum_{\pi}\frac{1}{\lfloor N/2\rfloor}\sum_{i=1}^{\lfloor N/2\rfloor}\bar{\ell}\left( G_{\pi(i)}^{(j)},G_{\pi(\lfloor N/2\rfloor+i)}^{(j)} \right) \right)\\
&\leq\mathbb{E}_{\mathcal{D}}\left( \frac{1}{N!}\sum_{\pi}\sup_{F\in\mathcal{F}}\frac{1}{M}\sum_{j=1}^{M}\frac{1}{\lfloor N/2\rfloor}\sum_{i=1}^{\lfloor N/2\rfloor}\bar{\ell}\left( G_{\pi(i)}^{(j)},G_{\pi(\lfloor N/2\rfloor+i)}^{(j)} \right) \right)\\
&\leq\frac{1}{N!}\sum_{\pi}\mathbb{E}_{\mathcal{D}}\left( \sup_{F\in\mathcal{F}}\frac{1}{M}\sum_{j=1}^{M}\frac{1}{\lfloor N/2\rfloor}\sum_{i=1}^{\lfloor N/2\rfloor}\bar{\ell}\left( G_{\pi(i)}^{(j)},G_{\pi(\lfloor N/2\rfloor+i)}^{(j)} \right) \right)\\
&=\mathbb{E}_{\mathcal{D}}\left( \sup_{F\in\mathcal{F}}\frac{1}{M}\sum_{j=1}^{M}\frac{1}{\lfloor N/2\rfloor}\sum_{i=1}^{\lfloor N/2\rfloor}\bar{\ell}\left( G_{\pi(i)}^{(j)},G_{\pi(\lfloor N/2\rfloor+i)}^{(j)} \right) \right)\\
&=\mathbb{E}_{\mathcal{D}}\left( \sup_{F\in\mathcal{F}}\left\{ \tilde{\mathcal{L}}_{\mathcal{D}}(F)-\mathcal{L}(F) \right\} \right)
\end{aligned}
\tag{38}
$$

where $\tilde{\mathcal{L}}_{\mathcal{D}}(F) = \frac{1}{M}\sum_{j=1}^{M}\frac{1}{\lfloor N/2\rfloor}\sum_{i=1}^{\lfloor N/2\rfloor}\ell\left( G_{\pi(i)}^{(j)},G_{\pi(\lfloor N/2\rfloor+i)}^{(j)} \right).$

For convenience, we let $S = M\lfloor N/2 \rfloor$ and rename the graph-pair as $(G_s, \bar{G}_s)$. So we have $S$ independent samples. By introducing a virtual dataset $\mathcal{D}' \subset \mathbb{G}$ with size $S$, we obtain

$$
\begin{aligned}
&\mathbb{E}_{\mathcal{D}}\left(\sup_{F \in \mathcal{F}}\left\{\tilde{\mathcal{L}}_{\mathcal{D}}(F) - \mathcal{L}(F)\right\}\right) \\
=&\mathbb{E}_{\mathcal{D}}\left(\sup_{F \in \mathcal{F}}\left\{\frac{1}{S}\sum_{s=1}^{S}\ell\left(G_s, \bar{G}_s\right) - \mathcal{L}(F)\right\}\right) \\
=&\mathbb{E}_{\mathcal{D}}\left(\sup_{F \in \mathcal{F}}\left\{\frac{1}{S}\sum_{s=1}^{S}\ell\left(G_s, \bar{G}_s\right) - \mathbb{E}_{\mathcal{D}'}\left(\frac{1}{S}\sum_{s=1}^{S}\ell\left(G'_s, \bar{G}'_s\right)\right)\right\}\right) \\
\leq&\mathbb{E}_{\mathcal{D},\mathcal{D}'}\left(\sup_{F \in \mathcal{F}}\frac{1}{S}\sum_{s=1}^{S}\left[\ell\left(G_s, \bar{G}_s\right) - \ell\left(G'_s, \bar{G}'_s\right)\right]\right)
\end{aligned}
\tag{39}
$$

where the inequality holds due to Jensen's inequality. By introducing the Rademacher variable $\epsilon_s \in \{-1, 1\}$, we have

$$
\begin{aligned}
&\mathbb{E}_{\mathcal{D}}\left(\sup_{F \in \mathcal{F}}\left\{\tilde{\mathcal{L}}_{\mathcal{D}}(F) - \mathcal{L}(F)\right\}\right) \\
\leq&\mathbb{E}_{\mathcal{D},\mathcal{D}'}\mathbb{E}_{\epsilon}\left(\sup_{F \in \mathcal{F}}\frac{1}{S}\sum_{s=1}^{S}\epsilon_s\left[\ell\left(G_s, \bar{G}_s\right) - \ell\left(G'_s, \bar{G}'_s\right)\right]\right) \\
\leq&\mathbb{E}_{\mathcal{D},\mathcal{D}'}\mathbb{E}_{\epsilon}\left(\sup_{F \in \mathcal{F}}\left\{\frac{1}{S}\sum_{s=1}^{S}\epsilon_s\ell\left(G_s, \bar{G}_s\right)\right\} + \sup_{F \in \mathcal{F}}\left\{\frac{1}{S}\sum_{s=1}^{S}(-\epsilon_s)\left(G'_s, \bar{G}'_s\right)\right\}\right) \\
\leq&2\mathbb{E}_{\mathcal{D},\epsilon}\left(\sup_{F \in \mathcal{F}}\frac{1}{S}\sum_{s=1}^{S}\epsilon_s\ell\left(G_s, \bar{G}_s\right)\right) \\
=&2\mathbb{E}_S(\hat{\mathcal{R}}_S(\mathcal{F}))
\end{aligned}
\tag{40}
$$

where $\mathcal{R}_S(\mathcal{F}) := \mathbb{E}_S(\hat{\mathcal{R}}_S(\mathcal{F}))$ is the Rademacher complexity.

Combining (37), we arrive at

$$
\sup_{F \in \mathcal{F}}\{\hat{\mathcal{L}}_{\mathcal{D}}(F) - \mathcal{L}(F)\} \leq 2\mathcal{R}_S(\mathcal{F}) + \sqrt{\frac{\ln(1/\delta)}{2MN}}
\tag{41}
$$

According to Lemma E.4, Lemma E.5, and Lemma E.6, the Lipschitz constants of the GIN, GT, and reference layer are

$$
\begin{aligned}
L_{\text{GIN}} =& \max_{(i,p) \in [N] \times [M]}\|\mathbf{A}_i^{(p)}\|_2^{\vartheta}\prod_{j=1}^{\vartheta'}\|\mathbf{W}_j\|_2 \\
L_{\text{Ref}} =&4\sqrt{\frac{\gamma R}{n}} \\
L_{\text{GT}} =&\mu^{\vartheta}\prod_{j=1}^{\vartheta\vartheta'}\|\mathbf{W}_j\|_2
\end{aligned}
\tag{42}
$$

Since there are $Q$ parallel GINs, according to Lemma E.8, the Lipschitz constant of their combinations is

$$
L_{\text{QGIN}} = \max_q L_{\text{GIN}}^{(q)}
\tag{43}
$$

where $L_{\text{GIN}}^{(q)} = \max_{(i,p) \in [N] \times [M]}\|\mathbf{A}_i^{(p)}\|_2^{\vartheta}\prod_{j=1}^{\vartheta'}\|\mathbf{W}_j^{(q)}\|_2$. Based on the composition of these network components and their specific configurations, the Lipschitz constant of $F$ is

$$
L_F = \left(4\sqrt{\frac{\gamma R}{n}} + \frac{1}{\sqrt{n}}\right)\left(\max_{(i,p) \in [N] \times [M]}\|\mathbf{A}_i^{(p)}\|_2^{\vartheta_1}\max_{q \in [Q]}\prod_{j=1}^{\kappa_1}\|\mathbf{W}_j^{\text{GIN}_q}\|_2\right)\left(\mu^{\vartheta_2}\prod_j^{\kappa_2}\|\mathbf{W}_j^{\text{GT}}\|_2\right)
\tag{44}
$$

where $\kappa_1$ is the maximum number of MLP layers in each GIN and $\kappa_2$ is the total number of weight matrices excluding those in the attention maps of the transformer. Suppose the loss function $\ell$ is $\tau$-Lipschitz, then the Lipschitz constant of $\ell \circ \mathcal{F}$ is $L_{\ell \circ F} = \tau L_F$.

Let $\tilde{\mathbf{Z}}^{(j)} = [\bar{\mathbf{A}}^{(j)} \mathbf{Z}_1^{(j)}, \ldots, \bar{\mathbf{A}}^{(j)} \mathbf{Z}_Q^{(j)}]$, where $\bar{\mathbf{A}}^{(j)} = \text{diag}(\mathbf{A}_1^{(j)}, \ldots, \mathbf{A}_N^{(j)}) \in \mathbb{R}^{Nn \times Nn}$. We further form $\hat{\mathbf{Z}} = [\tilde{\mathbf{Z}}^{(1)}; \tilde{\mathbf{Z}}^{(2)}; \ldots; \tilde{\mathbf{Z}}^{(M)}] \in \mathbb{R}^{MNn \times Q\bar{d}}$. According to Lemma E.7, the covering number of $\mathcal{Z} = \{\hat{\mathbf{Z}} \in \mathbb{R}^{MNn \times Q\bar{d}} : \|\hat{\mathbf{Z}}\|_F \leq \beta\}$ is bounded as

$$\ln \mathcal{N}\left(\mathcal{Z}, \epsilon, \|\cdot\|_F\right) \leq \frac{\beta^2 Q^2 \bar{d}^2 \ln\left(2Q\bar{d}\right)}{\epsilon^2} \tag{45}$$

Therefore, using Lemma E.9, the covering number of $\ell \circ \mathcal{F} \times \mathcal{Z}$ is bounded as

$$\ln \mathcal{N}\left(\ell \circ \mathcal{F}, \epsilon, \|\cdot\|_F\right) \leq \frac{\tau^2 L_F^2 \beta^2 Q^2 \bar{d}^2 \ln\left(2Q\bar{d}\right)}{\epsilon^2} \triangleq \frac{\varphi}{\epsilon^2} \tag{46}$$

Using Lemma E.2, we can bound the Rademacher complexity of our model class as

$$\begin{aligned}
\mathcal{R}_S(\ell \circ \mathcal{F}) &\leq \inf_{\alpha > 0} \left( \frac{4\alpha}{\sqrt{S}} + \frac{12}{S} \int_\alpha^{\sqrt{S}} \frac{\sqrt{\varphi}}{\epsilon} \, d\epsilon \right) \\
&\leq \inf_{\alpha > 0} \left( \frac{4\alpha}{\sqrt{S}} + \frac{12\sqrt{\varphi}}{S} \ln\left(\frac{\sqrt{S}}{\alpha}\right) \right) \\
&\leq \frac{4 + 12\sqrt{\varphi}\ln(S)}{S}
\end{aligned} \tag{47}$$

where in the last inequality we have let $\alpha = 1/\sqrt{S}$.

Now combining (47), (46), and (41), we arrive at

$$\hat{\mathcal{L}}_\mathcal{D}(F) \leq \mathcal{L}(F) + \frac{16 + 48\tau L_F \beta Q\bar{d}\sqrt{\ln\left(2Q\bar{d}\right)} \ln(MN/2)}{MN} + \sqrt{\frac{\ln(1/\delta)}{2MN}} \tag{48}$$

This completes the proof.

### E.1 SUPPORTING LEMMAS AND THEIR PROOFS

**Lemma E.1** (McDiarmid's inequality (McDiarmid et al., 1989)). *Suppose $f : \prod_{k=1}^m \Omega_k \to \mathbb{R}$ with bounded differences $\{c_k\}_{k=1}^m$ then, for all $\epsilon > 0$, there holds*

$$\Pr_{\mathbf{z}} \{f(\mathbf{z}) - \mathbb{E}_{\mathbf{z}} f(\mathbf{z}) \geq \epsilon\} \leq e^{-\frac{2\epsilon^2}{\Sigma_{k=1}^m c_k^2}}$$

**Lemma E.2.** *Suppose the Lipschitz constant of $\ell \circ F$ is $L$, then the Rademacher complexity of $\ell \circ \mathcal{F}$ is bound as*

$$\mathcal{R}_S(\ell \circ \mathcal{F}) \leq xxx \tag{49}$$

*Proof.* We show the Dudley entropy integral bound Bartlett et al. (2017) below.

**Lemma E.3.** *Let $\mathcal{F}$ be a real-valued function class taking values in $[0, 1]$, and assume that $\mathbf{0} \in \mathcal{F}$. Then*

$$\mathcal{R}_S(\mathcal{F}) \leq \inf_{\alpha > 0} \left( \frac{4\alpha}{\sqrt{S}} + \frac{12}{S} \int_\alpha^{\sqrt{S}} \sqrt{\ln \mathcal{N}\left(\epsilon, \mathcal{F}, \rho\right)} \, d\epsilon \right).$$

$\square$

**Lemma E.4.** *The Lipschitz constant of the reference layer is $L_{ref} = 4\sqrt{\frac{\theta R}{n}}$.*

*Proof.* According to the definition of MMD, we have

$$
\left| \text{MMD}^2\left(\mathbf{H}, \mathbf{V}\right) - \text{MMD}^2\left(\mathbf{H}', \mathbf{V}\right) \right| \leq \left| \frac{1}{n^2} \sum_{i,j=1}^{n} \left[ \exp\left(-\theta \|\mathbf{h}_i - \mathbf{h}_j\|_2^2\right) - \exp\left(-\theta \|\mathbf{h}'_i - \mathbf{h}'_j\|_2^2\right) \right] \right|
$$

$$
+ \left| \frac{2}{mn} \sum_{i=1}^{n} \sum_{j=1}^{m} \left[ \exp\left(-\theta \|\mathbf{h}_i - \mathbf{v}_j\|_2^2\right) - \exp\left(-\theta \|\mathbf{h}'_i - \mathbf{v}_j\|_2^2\right) \right] \right|
$$

$$
\overset{(a)}{\leq} \frac{\sqrt{\theta}}{n^2} \sum_{i,j=1}^{n} \left| \|\mathbf{h}_i - \mathbf{h}_j\|_2 - \|\mathbf{h}'_i - \mathbf{h}'_j\|_2 \right| + \frac{2\sqrt{\theta}}{mn} \sum_{i=1}^{n} \sum_{j=1}^{m} \left| \|\mathbf{h}_i - \mathbf{v}_j\|_2 - \|\mathbf{h}'_i - \mathbf{v}_j\|_2 \right|
$$

$$
\overset{(b)}{\leq} \frac{\sqrt{\theta}}{n^2} \sum_{i,j=1}^{n} \left\| \left(\mathbf{h}_i - \mathbf{h}'_i\right) - \left(\mathbf{h}_j - \mathbf{h}'_j\right) \right\|_2 + \frac{2\sqrt{\theta}}{mn} \sum_{i=1}^{n} \sum_{j=1}^{m} \left\| \left(\mathbf{h}_i - \mathbf{h}'_i\right) - \left(\mathbf{v}_j - \mathbf{v}_j\right) \right\|_2
$$

$$
\leq \frac{4\sqrt{\theta}}{n} \sum_{i=1}^{n} \|\mathbf{h}_i - \mathbf{h}'_i\|_2
$$

$$
\overset{(c)}{\leq} 4\sqrt{\frac{\theta}{n}} \|\mathbf{H} - \mathbf{H}'\|_F
$$

In the above derivation, (a) holds due to $\left| \exp\left(-x^2\right) - \exp\left(-y^2\right) \right| \leq |x - y|$ for any $x, y \geq 0$, (b) holds due to the triangle inequality, and (c) holds by the Cauchy–Schwarz inequality.

The output of the layer is $\mathbf{S}$, for which we have

$$
\|\mathbf{S} - \mathbf{S}'\|_2 = \sqrt{\sum_{i=1}^{N} \sum_{j=1}^{R} |s_{ij} - s'_{ij}|^2}
$$

$$
\leq 4\sqrt{\frac{\theta}{n}} \sqrt{\sum_{i=1}^{N} \sum_{j=1}^{R} \|\mathbf{H}_i - \mathbf{H}'_i\|_F^2}
$$

$$
= 4\sqrt{\frac{\theta R}{n}} \|\mathbf{H} - \mathbf{H}'\|_F
$$

This finished the proof.

$\square$

**Lemma E.5.** *Suppose the GIN $f$ has $Q$ layers and each layer has an MLP of $Q'$ layers. Then the Lipschitz constant of $f$ is $L_{GIN} = \max_{i \in [N]} \|\mathbf{A}_i\|_2^Q \prod_{j=1}^{QQ'} \|\mathbf{W}_j\|_2$.*

*Proof.* Recall that the $l$-th layer of the GIN can be formulated as

$$
f^{(l)}\left(\mathbf{A}, \mathbf{Z}^{(l-1)}\right) = \text{MLP}^{(l)}\left(\left(\mathbf{A} + \epsilon \mathbf{I}\right) \cdot \mathbf{Z}^{(l-1)}\right) \tag{50}
$$

where $\mathbf{Z}^{(0)} = \mathbf{X}$. For convenience, let $\epsilon = 0$. We put all adjacency matrices together to form a big block diagonal matrix $\bar{\mathbf{A}}$ of size $Nn \times Nn$. Then the spectral norm of $\bar{\mathbf{A}}$ is $\max_{i \in [N]} \|\mathbf{A}_i\|_2$. Similarly, we form a big matrix $\bar{\mathbf{Z}}$ of size $Nn \times d$. Then we have

$$
\bar{Z}^{(l)} = f^{(l)}\left(\bar{\mathbf{A}}, \bar{\mathbf{Z}}^{(l-1)}\right) = \text{MLP}^{(l)}\left(\bar{\mathbf{A}}\bar{\mathbf{Z}}^{(l-1)}\right) \tag{51}
$$

Then the Lipschitz constant of $f^{(l)}$ is $\max_{i \in [N]} \|\mathbf{A}_i\|_2 \prod_{j=1}^{Q} \rho_j \|\mathbf{W}_j\|_2$, where $W_j$ is the weight matrix and $\rho_j$ is the Lipschitz constant of the layer. Since most activation functions such as ReLu and Sigmoid are 1-Lipschitz, we let $\rho_i = 1 \ \forall i$. Given that $f$ has $Q$ layers, we conclude that the Lipschitz constant is $\max_{i \in [N]} \|\mathbf{A}_i\|_2^Q \prod_{j=1}^{QQ'} \|\mathbf{W}_j\|_2$. $\square$

**Lemma E.6.** *Suppose the graph transformer $g$ is composed of $Q$ blocks and each block has an MLP of $Q'$ layers. Suppose the attention map is $\mu$-Lipschitz. Then the Lipschitz constant of $g$ is $L_{GT} = \mu^{QQ'} \prod_{j=1}^{QQ'} \|\mathbf{W}_j\|_2$.*

*Proof.* Recall that the self-attention is

$$\text{attn}\left(\mathbf{\Gamma}_i^{(j)}\right) = \text{softmax}\left(\frac{(\mathbf{\Gamma}_i^{(j)}\mathbf{W}_Q)(\mathbf{\Gamma}_i^{(j)}\mathbf{W}_K)^\top}{\sqrt{d'}}\right)(\mathbf{\Gamma}_i^{(j)}\mathbf{W}_V) \tag{52}$$

Assume that the softmax operation is $\mu$-Lipschitz with respect to the input $\mathbf{\Gamma}_i^{(j)}$. The Lipschitz constant of the self-attention mechanism is $\mu\|\mathbf{W}_V\|_2$. The self-attention is then followed by a residual connection, layer normalization, and MLP of $Q$-layers. We omit the residual connection and the layer normalization since they have a tiny impact on the analysis. For the MLP, the Lipschitz constant is $\prod_{j=1}^{Q'}\|\mathbf{W}_j\|_2$, where $\mathbf{W}_j$ is the weight matrix of layer $j$ and the activation functions are assumed to be 1-Lipschitz. Since $g$ has $Q$ sequential blocks, the total Lipschitz constant is $\mu^{QQ'}\prod_{j=1}^{QQ'}\|\mathbf{W}_j\|_2$. $\qquad\square$

**Lemma E.7** (Lemma 3.2 in (Bartlett et al., 2017))**.** *Let conjugate exponents $(p, q)$ and $(r, s)$ be given with $p \leq 2$, as well as positive reals $(a, b, \epsilon)$ and positive integer m. Let matrix $\mathbf{X} \in \mathbb{R}^{n \times d}$ be given with $\|\mathbf{X}\|_p \leq b$. Then*

$$\ln\mathcal{N}\left(\left\{\mathbf{X}\mathbf{A} : \mathbf{A} \in \mathbb{R}^{d \times m}, \|\mathbf{A}\|_{q,s} \leq a\right\}, \epsilon, \|\cdot\|_F\right) \leq \left\lceil\frac{a^2 b^2 m^{2/r}}{\epsilon^2}\right\rceil \ln(2dm)$$

**Lemma E.8.** *Suppose $\mathbf{Z}_i \in \mathbb{R}^{n \times d}$ and $f_i(\mathbf{Z}_i)$ is $L_i$-Lipschitz continuous with respect to $\mathbf{Z}_i$, where $i = 1, \ldots, Q$. Let $\bar{\mathbf{Z}} = [\mathbf{Z}_i; \ldots; \mathbf{Z}_Q] \in \mathbb{R}^{n \times dQ}$. Let $F = [f_1, f_2, \ldots, f_Q]$. Then the Lipschitz constant of $F(\bar{\mathbf{Z}})$ with respect to $\bar{\mathbf{Z}}$ is $L_F = \max_i \alpha_i$.*

*Proof.* Based on the settings, we have

$$\begin{aligned}
&\|F(\bar{\mathbf{Z}}) - F(\bar{\mathbf{Z}}')\|_F \\
&= \left\|f_1(\mathbf{Z}_1) - f_1(\mathbf{Z}_1') \quad \ldots \quad f_Q(\mathbf{Z}_Q) - f_Q(\mathbf{Z}_Q')\right\|_F \\
&= \sqrt{\sum_{i=1}^{Q}\|f_i(\mathbf{Z}_i) - f_i(\mathbf{Z}_i')\|_F^2} \\
&\leq \sqrt{\sum_{i=1}^{Q}\alpha_i^2\|\mathbf{Z}_i - \mathbf{Z}_i'\|_F^2} \\
&\leq \max_i \alpha_i\sqrt{\sum_{i=1}^{Q}\|\mathbf{Z}_i - \mathbf{Z}_i'\|_F^2} \\
&= \max_i \alpha_i\|\bar{\mathbf{Z}} - \bar{\mathbf{Z}}'\|_F
\end{aligned} \tag{53}$$

$\qquad\square$

**Lemma E.9.** *Suppose $\phi$ is an $\alpha$-Lipschitz continuous function, then $\ln\mathcal{N}(\epsilon, \phi \circ \mathcal{F}, \rho) \leq \ln\mathcal{N}(\epsilon/\alpha, \mathcal{F}, \rho)$.*

*Proof.* This is a well-known result, and we will not repeat the proof. $\qquad\square$

## F  GIN AND GRAPH TRANSFORMER BASED MODEL

To design a universal graph representation model $F$, we incorporate two main components: a GNN module $f$ and a graph transformer module $g$. We build GNN encoder on top of transformer encoder $g_\psi \circ f_\theta(\cdot)$. The GNN encoder specializes in learning local representations of the structure of a node's immediate neighborhood, while the transformer computes all pairwise node interactions, enabling global reasoning through attention mechanisms. Specifically, we adopt the Graph Isomorphism Network (GIN) (Xu et al., 2019) as the GNN encoder, and its $l$-th layer can be formulated as

$$f^{(l)}\left(\mathbf{A}_i^{(j)}, \mathbf{Z}_i^{(j)}\right) = \text{MLP}^{(l)}\left(\left(\tilde{\mathbf{A}}_i^{(j)} + \epsilon\mathbf{I}\right) \cdot \mathbf{Z}_i^{(j)}\right) \tag{54}$$

Table 6: Dataset Statistics.

| Dataset | Domain | #Graphs | #Avg.Nodes | #Features | #Classes | Task |
|---|---|---|---|---|---|---|
| ENZYMES | Bioinformatics | 600 | 32.63 | 21 | 6 | Graph Classification/Graph Clustering |
| NCI1 | Small molecules | 4110 | 29.87 | 37 | 2 | Graph Classification/Graph Clustering |
| NCI109 | Small molecules | 4127 | 29.68 | 38 | 2 | Graph Classification |
| DD | Bioinformatics | 1178 | 284.32 | 89 | 2 | Graph Classification |
| Mutagenicity | Small molecules | 4337 | 30.32 | 14 | 2 | Graph Classification |
| COLLAB | Social networks | 5000 | 74.49 | 0 | 2 | Graph Classification/Graph Clustering |
| REDDIT-BINARY | Social networks | 2000 | 429.63 | 0 | 2 | Graph Classification/Graph Clustering |
| REDDIT-MULTI | Social networks | 4999 | 508.52 | 0 | 5 | Graph Clustering |
| IMDB-BINARY | Social networks | 1000 | 19.77 | 0 | 2 | Graph Classification |
| IMDB-MULTI | Social networks | 1500 | 13.00 | 0 | 3 | Graph Classification |
| Letter-med | Computer vision | 2250 | 4.67 | 2 | 15 | Graph Classification |
| COIL-RAG | Computer vision | 3900 | 3.01 | 64 | 100 | Graph Classification |
| Cuneiform | Computer vision | 267 | 21.27 | 10 | 30 | Graph Classification |

where $\tilde{\mathbf{A}}_i^{(j)}$ is the adjacency matrix of $G_i^{(j)}$ with self-loops, $\epsilon$ is a hyperparameter, $\mathrm{MLP}^{(l)}$ is a multilayer perceptron (MLP) in layer $l$, and the parameters to optimize are denoted as $\theta$.

The graph transformer (GT) module consists of a self-attention mechanism and a feed-forward network, which is usually an MLP. Let $\boldsymbol{\Gamma}_i^{(j)} \in \mathbb{R}^{n_i \times d}$ represent the matrix of hidden states, and $\mathbf{W}_Q$, $\mathbf{W}_K$, and $\mathbf{W}_V$ of size $d \times d'$ be projection matrices, the self-attention mechanism is

$$\mathrm{attn}\left(\boldsymbol{\Gamma}_i^{(j)}\right) = \mathrm{softmax}\left(\frac{(\boldsymbol{\Gamma}_i^{(j)}\mathbf{W}_Q)(\boldsymbol{\Gamma}_i^{(j)}\mathbf{W}_K)^\top}{\sqrt{d'}}\right)(\boldsymbol{\Gamma}_i^{(j)}\mathbf{W}_V) \tag{55}$$

which is further transformed to $\hat{\boldsymbol{\Gamma}}_i^{(j)} = \mathrm{Norm}\left(\boldsymbol{\Gamma}_i^{(j)} + \mathrm{attn}\left(\boldsymbol{\Gamma}_i^{(j)}\right)\right)$. Then the $l$-th transformer block can be formulated as

$$g^{(l)}\left(\boldsymbol{\Gamma}_i^{(j)}\right) = \mathrm{Norm}\left(\hat{\boldsymbol{\Gamma}}_i^{(j)} + \mathrm{FFN}\left(\hat{\boldsymbol{\Gamma}}_i^{(j)}\right)\right) \tag{56}$$

We denote the parameters of the transformer module as $\psi$. Finally, we concatenate the outputs of the GIN and GT, leading to the following node representations of $G_i^{(j)}$:

$$\mathbf{H}_i^{(j)} = g_\psi \circ f_\theta\left(\mathbf{A}_i^{(j)}, \mathbf{Z}_i^{(j)}\right) \Big\| f_\theta\left(\mathbf{A}_i^{(j)}, \mathbf{Z}_i^{(j)}\right), \quad i \in [N_j], \quad j \in [M]. \tag{57}$$

For convenience, we let $\mathcal{W} = \{\psi, \theta\}$, which is the set of all parameters of the GIN and GT.

# G DETAILS ABOUT EXPERIMENTAL SETTINGS

## G.1 DATASETS

The basic information and statistics of the graph datasets we used in the experiments are shown in Table 6. In our experiments, the concatenation of the original node attributes and node labels in the datasets is used as initial input node features.

## G.2 DETAILS OF MODEL TESTING IN FEW-SHOT GRAPH CLASSIFICATION

Specifically, let the dataset in the downstream task be $\mathcal{G}^{\mathrm{Down}} = \{\mathcal{G}^{\mathrm{train}}, \mathcal{G}^{\mathrm{test}}\}$, where $\mathcal{G}^{\mathrm{train}} = \{(\mathbf{A}_i^{\mathrm{train}}, \mathbf{X}_i^{\mathrm{train}})\}_{i=1}^{N_{\mathrm{train}}}$ and $\mathcal{G}^{\mathrm{test}} = \{(\mathbf{A}_i^{\mathrm{test}}, \mathbf{X}_i^{\mathrm{test}})\}_{i=1}^{N_{\mathrm{test}}}$. For $\mathcal{G}^{\mathrm{train}}$, applying (2), (3), (4), and (5) sequentially, we obtain $\mathbf{Z}^{\mathrm{train}}$, the aligned node feature matrix of the training set, which is further modified by using Algorithm 1. Now we apply the pretrained model to $\mathbf{Z}^{\mathrm{train}}$ to obtain the embedding vector of each training graph, i.e., $\mathbf{g}_i^{\mathrm{train}} = F_{\mathcal{W},\mathcal{V},\gamma}(\mathbf{A}_i^{\mathrm{train}}, \mathbf{Z}_i^{\mathrm{train}})$, $i \in N_{\mathrm{train}}$.

Let the kernel matrix of the training set be $\mathbf{K}_{\lambda_q}^{\mathrm{train}} = \mathbf{U}\boldsymbol{\Sigma}\mathbf{V}^\top$, and the cross-kernel matrix between the test and training sets be $\mathbf{K}_{\lambda_q}^{\mathrm{test}}$. $\mathbf{Z}_{\lambda_q}^{\mathrm{test}} = \mathbf{K}_{\lambda_q}^{\mathrm{test}}\mathbf{V}_{\bar{d}}\boldsymbol{\Sigma}_{\bar{d}}^{-1/2}$, $q \in [Q]$. Then we obtain $\mathbf{Z}^{\mathrm{test}} = \left[\mathbf{Z}_{\lambda_1}^{\mathrm{test}}, \mathbf{Z}_{\lambda_2}^{\mathrm{test}}, \ldots, \mathbf{Z}_{\lambda_Q}^{\mathrm{test}}\right]$, the aligned node feature matrix of the testing set, which is further modified by using Algorithm 1. Now, similar to the training data, we have $\mathbf{g}_i^{\mathrm{test}} = F_{\mathcal{W},\mathcal{V},\gamma}(\mathbf{A}_i^{\mathrm{test}}, \mathbf{Z}_i^{\mathrm{test}})$,

$i \in N_{\text{test}}$. These steps are summarized in Algorithm 3, where the underlined values are frozen in Algorithm 1.

---

**Algorithm 3** Few-shot graph classification

---

**Input:** $\mathcal{G}^{\text{Down}} = \left\{ \mathcal{G}^{\text{train}}, \mathcal{G}^{\text{test}} \right\}, \left\{ \mathbf{R}_{\text{pre}}^{(j)} \right\}_{j=1}^{M}, \left\{ \boldsymbol{\mu}_{\text{pre}}^{(j)} \right\}_{j=1}^{M}$
1: Compute $\mathbf{Z}^{\text{train}}, \mathbf{Z}^{\text{test}}$ using (4) and (5).
2: $\mathbf{R}^{\text{train}} \leftarrow$ Algorithm 1$\left( \boldsymbol{\mu}^{\text{train}}, \left\{ \underline{\boldsymbol{\mu}}_{\text{pre}}^{(j)} \right\}_{j=1}^{M}, \left\{ \underline{\mathbf{R}}_{\text{pre}}^{(j)} \right\}_{j=1}^{M} \right)$
   $\mathbf{R}^{\text{test}} \leftarrow$ Algorithm 1$\left( \boldsymbol{\mu}^{\text{test}}, \boldsymbol{\mu}^{\text{train}}, \underline{\mathbf{R}}^{\text{train}}, \left\{ \underline{\boldsymbol{\mu}}_{\text{pre}}^{(j)} \right\}_{j=1}^{M}, \left\{ \underline{\mathbf{R}}_{\text{pre}}^{(j)} \right\}_{j=1}^{M} \right)$
3: Mean alignment: $\mathbf{Z}^{\text{train}} \leftarrow \mathbf{Z}^{\text{train}} \mathbf{R}^{\text{train}\top}, \mathbf{Z}^{\text{test}} \leftarrow \mathbf{Z}^{\text{test}} \mathbf{R}^{\text{test}\top}$.
4: Representation: $\mathbf{g}_i^{\text{train}} \leftarrow F_{\mathcal{W}, \mathcal{V}, \gamma} \left( \mathbf{A}_i^{\text{train}}, \mathbf{Z}_i^{\text{train}} \right), i \in [|\mathcal{G}^{\text{train}}|]$
5: Train the softmax classifier $f_c$ on $\{ \mathbf{g}_i^{\text{train}} \}$.
6: $\hat{\mathbf{y}}_i^{\text{test}} = f_c \circ F_{\mathcal{W}, \mathcal{V}, \gamma} \left( \mathbf{A}_i^{\text{test}}, \mathbf{Z}_i^{\text{test}} \right), i \in [|\mathcal{G}^{\text{test}}|]$.
**Output:** Predicted graph labels $\{ \hat{\mathbf{y}}_i^{\text{test}} \}$

---

### G.3 IMPLEMENTATION DETAILS

In our experiments, we use 6 Gaussian kernels with different $\lambda_q \in \{0.25, 0.5, 1, 2, 5, 10\}$. For all kernel matrices and the adjacency matrix, the truncated dimension of SVD $\bar{d}$ is set as 32. For each global graph obtained by Gaussian kernels, we use 6-GIN encoder to encode node features from different global graphs respectively. We implement each GIN encoder with 3 graph convolutional layers. The size of each hidden layer in GIN is set to 128. The graph transformer module consists of 3 equally wide layers, each containing 4 attention heads, with the dimension of each attention head set as 48. In the pre-training stage, all modules are optimized using Adam optimizer (Kinga et al., 2015) with fixed learning rate $\alpha_1 = 0.0005$ and a weight decay factor of $10^{-5}$, trained for 50 epochs. The Gaussian kernel parameter $\gamma$ in the reference layer employs a separate learning rate $\alpha_2 = 0.1$. The batch size for all datasets is fixed to 64.

**Few-shot learning settings** In the downstream tasks of few-shot graph classification, the classifier is a softmax classifier, which follows the setting in EdgePrompt (Fu et al., 2025). Regarding data splitting, we randomly choose 50 graphs in each class for training, and the remaining samples are used for testing. The number of epochs is set to 500, and the learning rate of the classifier is set to 0.001 for graph few-shot training. As the k-shot tasks are balanced classification, we employ accuracy as the evaluation metric following EdgePrompt.

For ProNoG, we used the provided checkpoint from the official open-source repository as the pretrained model. For BRIDGE, we followed the recommended settings in their paper and code to pretrain on the Cora dataset. In the downstream adaptation stage, we adopted the recommended hyperparameters for both methods. Experiments on COLLAB, REDDIT-B, IMDB-B, IMDB-M, and Letter-med are conducted under 50-shot setting following experiments in our paper. For COIL-RAG and Cuneiform, due to a lack of enough samples per class, we adopt 5-shot and 1-shot settings, respectively. Since the three baselines do not handle datasets without node attributes, to ensure fair comparison, we handle social network datasets without node attributes (COLLAB, REDDIT-B, IMDB-B, IMDB-M) uniformly across all methods. Following our proposed approach, we generate node attributes using truncated SVD on A+I (adjacency matrix with self-loops) as input for all baseline models. All the results of our method are obtained from models trained on 5 bio-chemical datasets (ENZYMES, DD, NCI1, NCI109, Mutagenicity) mentioned in the main part of the paper, which differ significantly from social networks and computer vision data in both semantics and structure.

**Zero-shot learning settings** In a zero-shot learning scenario, we use language model e5-base (Wang et al., 2022) to embed textual class descriptions into vector representations for all methods shown in Table 3.

**Unsupervised pre-training settings** Following You et al. (2020), we construct 3 augmentations for each graph before the global multi-graph construction and max-density mean alignment.

We conduct all experiments on a 14 vCPU Intel(R) Xeon(R) Gold 6348 CPU with one Nvidia A800-80G GPU, CUDA 11.8. We repeat five times with different random seeds and report the average results with standard deviation calculated by the numpy library function.

# H  MORE RESULTS

## H.1  COMPARISON OF INDUCTIVE AND TRANSDUCTIVE SETTINGS OF GLOBAL GRAPH CONSTRUCTION

We also conduct our experiments under a transductive learning setting, where the downstream dataset is treated as a whole for feature alignment and mean alignment, followed by train-set splitting. In the transductive setting, due to the combination of training and testing graphs, the features given by the global graph is better, which makes the performance better than that in the inductive learning, shown by Figure 2.

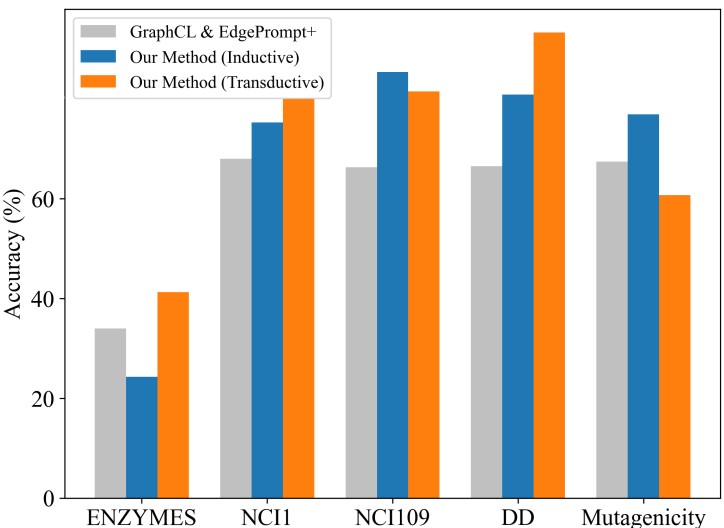

Figure 2: Comparison of inductive and transductive settings of global graph construction of our method. The best baseline GraphCL with EdgePrompt+ is also compared.

## H.2  FEW-SHOT LEARNING WITH FEWER LABELED SAMPLES

As shown in Figure 3, the classification accuracy of our method GraphVec-FM increases as the number of labeled samples increases. Our GraphVec-FM with 20-shot even outperforms the competitors with 50-shot in Table 1 of the main paper.

## H.3  IMPACT OF NUMBER OF PRE-TRAINING DATASETS

Figure 4 shows the change of classification accuracy when the number of datasets used in pre-training increases from 1 to 4. We can see that with more datasets used in pre-training, the performance in downstream tasks becomes better. This result indicates that the generalization ability of graph embeddings generated by our GraphVec-FM can benefit from the increase in the number of training datasets, which is an important capability for GFM. It can also be observed that even using model pre-trained on only 1 dataset, GraphVec-FM still outperforms other baselines shown in Table 1.

## H.4  MORE GRAPH CLUSTERING RESULTS

To further evaluate the performance of GraphVec-FM in graph clustering task, we conduct experiments on 4 more datasets. The results are shown in Table 7. We also provide the full version of Table 4 with NMI in Table 8

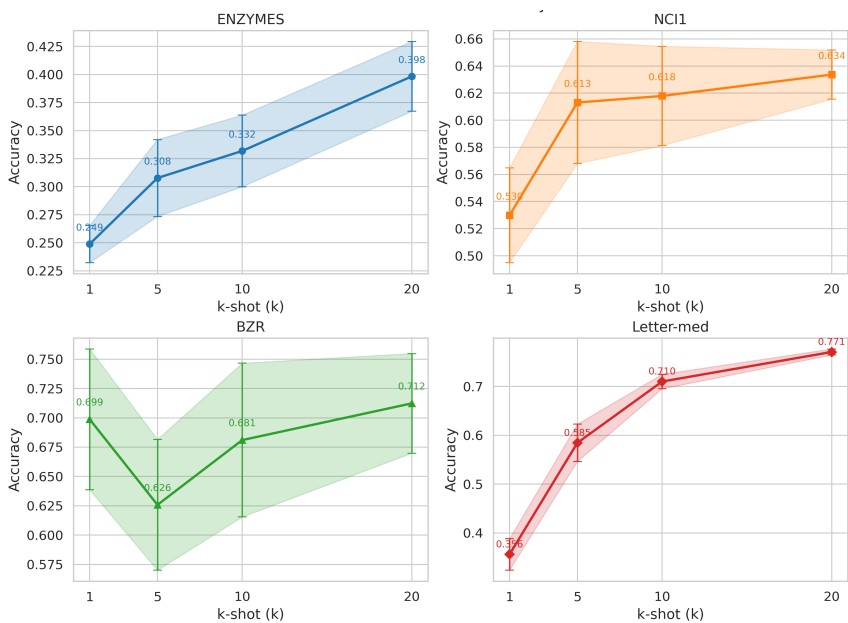

Figure 3: Classification accuracy trends of our method GraphVec-FM with varying $k$ values in few-shot learning across four datasets (PROTEINS, NCI109, DD, and Mutagenicity), shaded area represents standard deviation

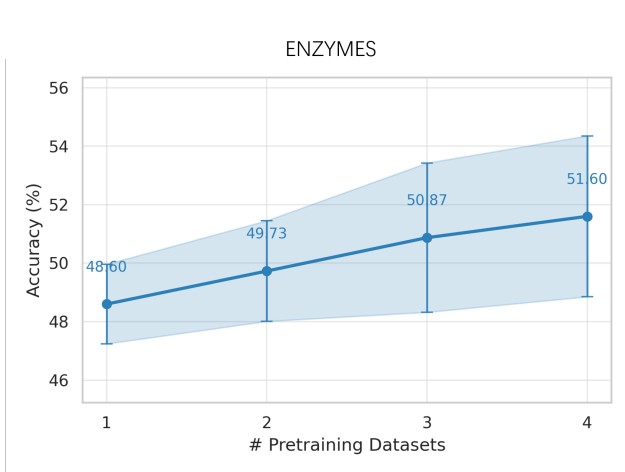

Figure 4: The change of classification accuracy in ENZYMES when the number of datasets used in pre-training increases from $1$ to $4$.

Table 7: Graph clustering results on PTC-MM, MUTAG, COX2 and BZR

| Dataset | PTC-MM | | | MUTAG | | | COX2 | | | BZR | | |
|---|---|---|---|---|---|---|---|---|---|---|---|---|
| | ACC | NMI | ARI | ACC | NMI | ARI | ACC | NMI | ARI | ACC | NMI | ARI |
| GraphCL+SC | $62.09 \pm 0.56$ | $2.14 \pm 0.43$ | $3.36 \pm 0.87$ | $73.22 \pm 2.66$ | $\mathbf{32.19} \pm 2.05$ | $23.44 \pm 2.45$ | $75.01 \pm 2.12$ | $1.24 \pm 0.37$ | $\mathbf{2.39} \pm 2.28$ | $72.88 \pm 1.66$ | $\mathbf{1.90} \pm 0.38$ | $\mathbf{3.47} \pm 0.59$ |
| GWF (Xu et al., 2022)+SC | $53.02 \pm 1.66$ | $0.36 \pm 0.28$ | $0.21 \pm 0.09$ | $73.92 \pm 4.30$ | $18.35 \pm 3.85$ | $24.48 \pm 4.69$ | $58.83 \pm 4.46$ | $1.16 \pm 0.41$ | $1.45 \pm 1.21$ | $52.76 \pm 0.80$ | $3.47 \pm 1.16$ | $-0.71 \pm 0.32$ |
| GLCC (Ju et al., 2023) | $61.61 \pm 0.24$ | $0.63 \pm 0.41$ | $1.24 \pm 1.38$ | $71.99 \pm 3.08$ | $13.18 \pm 6.93$ | $16.89 \pm 8.28$ | $77.37 \pm 1.11$ | $0.02 \pm 0.03$ | $-0.30 \pm 0.42$ | $63.62 \pm 9.79$ | $1.18 \pm 0.60$ | $1.12 \pm 0.97$ |
| Our Method | $\mathbf{65.74} \pm 0.00$ | $\mathbf{4.35} \pm 0.00$ | $\mathbf{6.31} \pm 0.00$ | $\mathbf{81.38} \pm 0.00$ | $31.00 \pm 0.00$ | $\mathbf{38.96} \pm 0.00$ | $\mathbf{78.58} \pm 0.00$ | $\mathbf{2.37} \pm 0.00$ | $2.19 \pm 0.00$ | $\mathbf{77.28} \pm 0.00$ | $0.16 \pm 0.00$ | $1.50 \pm 0.00$ |

Table 8: Graph clustering performance on ENZYMES, NCI1, COLLAB, REDDIT-BINARY, REDDIT-MULTI. The comparison numbers are from AMGC (Yang et al., 2025).

| Method | ENZYMES | | | NCI1 | | | COLLAB | | | REDDIT-BINARY | | | REDDIT-MULTI | | |
|---|---|---|---|---|---|---|---|---|---|---|---|---|---|---|---|
| | ACC | NMI | ARI | ACC | NMI | ARI | ACC | NMI | ARI | ACC | NMI | ARI | ACC | NMI | ARI |
| RW +SC | $17.0_{\pm0.0}$ | $0.7_{\pm0.0}$ | $0.3_{\pm0.0}$ | N/A | N/A | N/A | N/A | N/A | N/A | N/A | N/A | N/A | N/A | N/A | N/A |
| WL +SC | $21.0_{\pm0.0}$ | $3.1_{\pm0.0}$ | $1.5_{\pm0.0}$ | $50.1_{\pm0.0}$ | $0.0_{\pm0.0}$ | $0.0_{\pm0.0}$ | $53.2_{\pm0.0}$ | $2.0_{\pm0.0}$ | $0.5_{\pm0.0}$ | $57.6_{\pm0.0}$ | $9.0_{\pm0.0}$ | $2.2_{\pm0.0}$ | $18.7_{\pm0.0}$ | $9.0_{\pm0.0}$ | $4.0_{\pm0.0}$ |
| WL-OA +SC | $20.0_{\pm0.0}$ | $1.4_{\pm0.0}$ | $0.3_{\pm0.0}$ | $53.2_{\pm0.0}$ | $0.9_{\pm0.0}$ | $0.8_{\pm0.0}$ | $54.2_{\pm0.0}$ | $0.2_{\pm0.0}$ | $2.6_{\pm0.0}$ | $53.8_{\pm0.0}$ | $5.6_{\pm0.0}$ | $3.8_{\pm0.0}$ | $20.9_{\pm0.0}$ | $9.6_{\pm0.0}$ | $3.2_{\pm0.0}$ |
| SP +SC | $22.0_{\pm0.0}$ | $2.6_{\pm0.0}$ | $1.7_{\pm0.0}$ | $50.1_{\pm0.0}$ | $0.1_{\pm0.0}$ | $0.0_{\pm0.0}$ | $48.7_{\pm0.0}$ | $17.9_{\pm0.0}$ | $13.9_{\pm0.0}$ | $57.8_{\pm0.0}$ | $2.2_{\pm0.0}$ | $2.2_{\pm0.0}$ | $20.3_{\pm0.0}$ | $6.1_{\pm0.0}$ | $0.1_{\pm0.0}$ |
| LT +SC | $17.0_{\pm0.0}$ | $0.4_{\pm0.0}$ | $0.0_{\pm0.0}$ | N/A | N/A | N/A | N/A | N/A | N/A | N/A | N/A | N/A | N/A | N/A | N/A |
| GK +SC | $17.1_{\pm0.1}$ | $0.8_{\pm0.3}$ | $0.0_{\pm0.0}$ | $52.9_{\pm0.9}$ | $0.7_{\pm1.4}$ | $0.3_{\pm0.6}$ | $56.8_{\pm1.4}$ | $15.5_{\pm1.9}$ | $9.3_{\pm2.1}$ | $50.3_{\pm0.3}$ | $0.2_{\pm0.1}$ | $0.0_{\pm0.0}$ | $18.7_{\pm0.9}$ | $7.2_{\pm0.3}$ | $0.3_{\pm0.1}$ |
| InfoGraph +KM | $22.1_{\pm1.0}$ | $2.4_{\pm0.5}$ | $1.3_{\pm0.5}$ | $54.1_{\pm2.2}$ | $1.3_{\pm1.1}$ | $0.9_{\pm0.9}$ | $59.6_{\pm1.8}$ | $14.4_{\pm3.0}$ | $6.6_{\pm2.3}$ | $51.3_{\pm2.1}$ | $2.3_{\pm0.4}$ | $0.6_{\pm0.2}$ | $20.3_{\pm0.9}$ | $0.5_{\pm0.2}$ | $0.0_{\pm0.0}$ |
| InfoGraph +SC | $23.8_{\pm0.5}$ | $4.6_{\pm0.7}$ | $2.2_{\pm0.4}$ | $54.9_{\pm1.7}$ | $0.9_{\pm0.6}$ | $1.0_{\pm0.8}$ | $60.9_{\pm2.5}$ | $15.4_{\pm3.3}$ | $9.3_{\pm3.5}$ | $50.8_{\pm1.3}$ | $1.6_{\pm0.6}$ | $0.6_{\pm0.0}$ | $24.7_{\pm1.3}$ | $4.8_{\pm0.6}$ | $3.2_{\pm0.6}$ |
| GraphCL +KM | $21.5_{\pm0.2}$ | $1.6_{\pm0.1}$ | $0.9_{\pm0.1}$ | $55.4_{\pm1.7}$ | $0.5_{\pm0.3}$ | $1.0_{\pm0.9}$ | $58.0_{\pm1.2}$ | $17.8_{\pm2.0}$ | $11.3_{\pm0.6}$ | $51.9_{\pm3.3}$ | $3.4_{\pm1.2}$ | $0.2_{\pm0.0}$ | $25.3_{\pm0.9}$ | $5.3_{\pm0.3}$ | $4.3_{\pm0.6}$ |
| GraphCL +SC | $25.3_{\pm0.3}$ | $4.8_{\pm0.4}$ | $2.0_{\pm0.3}$ | $50.8_{\pm1.6}$ | $0.6_{\pm0.6}$ | $1.1_{\pm0.8}$ | $57.8_{\pm0.6}$ | $17.0_{\pm1.3}$ | $10.1_{\pm0.7}$ | $55.9_{\pm2.1}$ | $3.2_{\pm1.0}$ | $0.3_{\pm0.2}$ | $27.3_{\pm1.3}$ | $5.4_{\pm0.8}$ | $4.2_{\pm1.1}$ |
| JOAO + KM | $21.7_{\pm0.4}$ | $4.9_{\pm0.4}$ | $2.1_{\pm0.2}$ | $51.1_{\pm0.4}$ | $0.4_{\pm0.2}$ | $0.1_{\pm0.0}$ | $58.3_{\pm1.5}$ | $18.7_{\pm2.6}$ | $11.1_{\pm1.8}$ | $54.3_{\pm2.9}$ | $4.2_{\pm1.8}$ | $0.8_{\pm0.3}$ | $26.6_{\pm0.6}$ | $3.6_{\pm1.2}$ | $2.5_{\pm0.2}$ |
| JOAO + SC | $24.4_{\pm1.4}$ | $3.2_{\pm0.7}$ | $1.7_{\pm0.8}$ | $51.5_{\pm3.0}$ | $0.9_{\pm1.2}$ | $0.4_{\pm1.2}$ | $58.2_{\pm0.9}$ | $17.1_{\pm2.1}$ | $10.6_{\pm0.8}$ | $55.9_{\pm1.2}$ | $6.7_{\pm2.0}$ | $1.4_{\pm0.6}$ | $25.6_{\pm0.6}$ | $2.5_{\pm0.2}$ | $3.4_{\pm0.3}$ |
| GLCC | $24.4_{\pm1.4}$ | $3.2_{\pm0.7}$ | $1.7_{\pm0.8}$ | $60.9_{\pm2.3}$ | $5.3_{\pm1.9}$ | $3.6_{\pm2.6}$ | $60.3_{\pm0.6}$ | $18.2_{\pm1.3}$ | $12.1_{\pm0.9}$ | $67.6_{\pm3.4}$ | $9.2_{\pm2.6}$ | $8.7_{\pm1.7}$ | $32.4_{\pm2.1}$ | $11.8_{\pm1.3}$ | $8.2_{\pm1.6}$ |
| AMGC | $26.7_{\pm2.0}$ | $5.2_{\pm1.3}$ | $2.8_{\pm0.7}$ | $62.7_{\pm3.0}$ | $6.4_{\pm1.9}$ | $6.4_{\pm3.6}$ | $61.2_{\pm1.0}$ | $20.5_{\pm1.6}$ | $12.9_{\pm0.9}$ | $64.3_{\pm1.9}$ | $12.1_{\pm3.3}$ | $10.5_{\pm2.7}$ | $35.5_{\pm2.3}$ | $16.1_{\pm0.9}$ | $12.0_{\pm0.7}$ |
| GraphVec-FM | $29.1_{\pm0.4}$ | $7.7_{\pm0.3}$ | $3.5_{\pm0.2}$ | $64.8_{\pm0.0}$ | $6.5_{\pm0.0}$ | $8.7_{\pm0.0}$ | $61.8_{\pm0.0}$ | $21.2_{\pm0.0}$ | $18.9_{\pm0.0}$ | $71.6_{\pm0.0}$ | $20.7_{\pm0.0}$ | $18.6_{\pm0.0}$ | $40.0_{\pm0.2}$ | $17.3_{\pm0.1}$ | $12.1_{\pm0.2}$ |

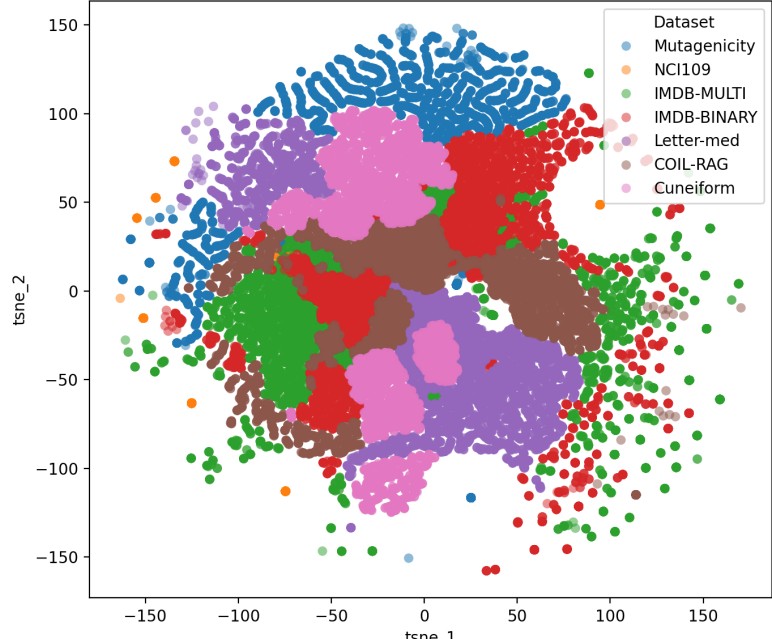

Figure 5: T-SNE visualization of aligned node embeddings of datasets from different domains.

## H.5 VISUALIZATION OF ALIGNED NODE EMBEDDINGS

## H.6 ABLATION STUDY

To verify the effectiveness of our proposed methods and modules, we conduct ablation study on global multi-graph construction, mean alignment algorithm, and reference layer. For the global multi-graph, we vary the number of multi-graphs from 1-6. Figure 6 and Figure 7 demonstrate the impact of the number of kernel parameters on downstream few-shot graph classification accuracy. These results were obtained by incrementally increasing the set of Gaussian kernel bandwidths from [0.25] to the full set [0.25, 0.5, 1, 2, 5, 10] used in our main experiments. It can be observed that classification accuracy improves with a greater number of global multi-graphs, particularly for datasets with original continuous node attributes. This observation further illustrates that the global multi-graphs constructed by using different kernel parameters help capture patterns from original features.

We conducted experiments using simple pooling without reference layers, and the results are presented in Table 9. After removing the reference layer module, the performance on all datasets shows degradation, especially on COLLAB(-4.7%), IMDB-BINARY(-7.1%), and Cuneiform(-4.2%). By removing the mean alignment, the performance on COLLAB, Letter-Med, and Cuneiform shows an

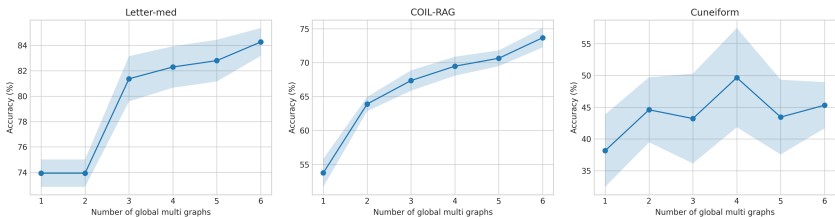

Figure 6: The few-shot graph classification accuracy in datasets with node attributes when the number of global multi-graphs increases from 1 to 6.

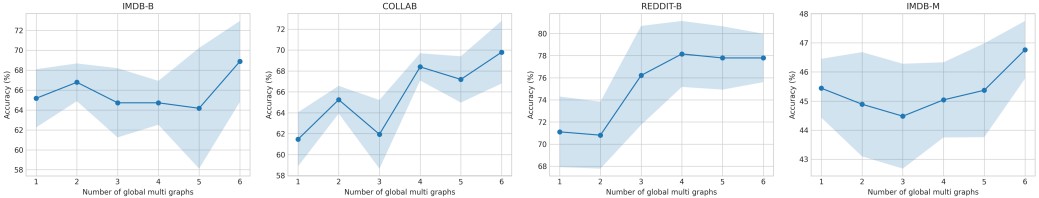

Figure 7: The few-shot graph classification accuracy in datasets without node attributes when the number of global multi-graphs increases from 1 to 6.

evident decrease (-4.9%, -2.8%, and -2.4%, respectively). Similarly, we also conduct experiments that remove the alignment module and both the alignment module and the reference layer. The overall impact of the two modules is shown in Table H.6.

Table 9: Ablation study of reference layer and mean alignment module.

| Model Variant | Dataset | | | | | | |
|---|---|---|---|---|---|---|---|
| | COLLAB (50-shot) | REDDIT-B (50-shot) | IMDB-B (50-shot) | IMDB-M (50-shot) | Letter-med (50-shot) | COIL-RAG (5-shot) | Cuneiform (1-shot) |
| Original Model | **69.79** ± 2.99 | **77.79** ± 2.19 | **68.87** ± 4.06 | **46.76** ± 0.99 | **84.27** ± 1.10 | **73.69** ± 1.43 | **45.32** ± 3.63 |
| Mean Readout Only | 65.04 ± 2.75 | 77.79 ± 3.11 | 61.78 ± 2.59 | 46.07 ± 2.60 | 83.17 ± 0.99 | 72.87 ± 1.26 | 41.04 ± 3.64 |
| w/o alignment | 64.90 ± 1.81 | 76.74 ± 5.43 | 66.06 ± 4.44 | 45.78 ± 1.46 | 81.87 ± 1.31 | 72.14 ± 0.35 | 42.87 ± 5.18 |

Table 10: The individual effect of alignment algorithm and reference layer on downstream cross-domain graph classification. The reported performance is averaged on 7 datasets.

| Alignment | Reference layer | Avg. Acc. (%) |
|---|---|---|
| ✓ | ✓ | 66.60 |
| ✓ | ✗ | 63.95 |
| ✗ | ✓ | 64.37 |
| ✗ | ✗ | 63.16 |

## H.7 ROBUSTNESS EVALUATION ON NOISY INPUT GRAPHS

To validate our model's performance on noisy graph data, we randomly added/deleted 10% edges to 50% of the test graphs during the few-shot test phase, and the results are shown in Table 11. It can be observed that there is only a slight decrease in terms of accuracy when the input graphs are perturbed or noisy, which demonstrates the robustness of our model.

## H.8 TIME AND MEMORY CONSUMPTION

To evaluate the computational cost and runtime of model pre-training, we conducted experiments on two datasets of different scales: the larger deezer_ego_net dataset (9,629 graphs) and the smaller ENZYMES dataset (600 graphs), each for 10 epochs. The wall-clock time and peak GPU/RAM

Table 11: Few-shot graph classification performance comparison between original input and perturbed input

| Dataset | COLLAB 50-shot | REDDIT-B 50-shot | IMDB-B 50-shot | IMDB-M 50-shot | Letter-med 50-shot | COIL-RAG 5-shot | Cuneiform 1-shot |
|---|---|---|---|---|---|---|---|
| Original Graphs | $\mathbf{69.79} \pm 2.99$ | $\mathbf{77.79} \pm 2.19$ | $\mathbf{68.87} \pm 4.06$ | $\mathbf{46.76} \pm 0.99$ | $\mathbf{84.27} \pm 1.10$ | $\mathbf{73.69} \pm 1.43$ | $\mathbf{45.32} \pm 3.63$ |
| 50% Perturbed Graphs | $69.21 \pm 1.20$ | $77.68 \pm 2.30$ | $65.22 \pm 1.07$ | $46.07 \pm 1.55$ | $83.47 \pm 3.22$ | $72.13 \pm 1.93$ | $44.14 \pm 3.82$ |

memory usage are presented in the Table 12. We also compared the wall-clock time and memory cost of pre-training with ProNoG (Yu et al., 2025b) and BRIDGE (Yuan et al., 2025) on the same dataset. It can be observed that our GraphVec-FM requires less training time, especially on relatively large datasets. GraphVec-FM demands more memory consumption, which is primarily due to the computation and storage of the global graph.

Table 12: Performance and resource utilization comparison

| Method | deezer_ego_net | | | ENZYMES | | |
|---|---|---|---|---|---|---|
| | wall-clock time (s) | GPU Peak Memory (GB) | RAM Peak Memory (GB) | wall-clock time (s) | GPU Peak Memory (GB) | RAM Peak Memory (GB) |
| ProNoG | 1194.61 | 0.12 | 1.1 | 69.54 | 0.05 | 0.96 |
| BRIDGE | 1600.28 | 0.31 | 1.27 | 116.99 | 0.18 | 1.17 |
| Our Method | 899.22 | 1.04 | 46.42 | 76.15 | 0.26 | 8.35 |

To ensure fair comparison, since ProNoG and BRIDGE process 4 graphs at once, we set our model's batch size to 4. All experiments are conducted on 14 vCPU Intel(R) Xeon(R) Gold 6348 CPU with one Nvidia A800-80G GPU, CUDA 11.8.

