# OpenReview forum: "Training A Foundation Model to Represent Graphs as Vectors"
_ICLR.cc/2026/Conference — Submitted to ICLR 2026_

### Official Review · Reviewer_auzx · 2025-10-18

**Soundness:** 3
**Presentation:** 2
**Contribution:** 2
**Rating:** 4
**Confidence:** 4

**Summary:**

The authors a Graph Foundation Model (GFM) designed to learn universal graph-level representations to preserve structural and semantic information. The key objective is to train a model that can embed any graph into a vector representation suitable for downstream tasks, e.g., graph classification, clustering, few-shot, and zero-shot tasks.

This consists of three main components: (1) Multi-graph Feature Alignment (Weighted graphs are introduced based on node attributes across datasets to generate consistent node embeddings that align features from diverse graph sources. (2) Density Maximization Mean Alignment (DMMA) to enhance feature consistency across datasets, with convergence guarantees. (3) Reference Distribution Module to preserve information flow from node to graph-level representations by aligning to a reference distribution.
A theoretical generalization bound is provided to justify the approach’s transferability.

**Strengths:**

I like the idea of building a domain-general graph representation model, aligning with the recent foundation models in non-Euclidean domains. The multi-graph-based feature alignment and DMMA algorithm offer an interesting approach to unifying representations across heterogeneous datasets.

- A theoretical depth is introduced

- It is practical in few-shot and zero-shot settings

**Weaknesses:**

- The overall pipeline (alignment + contrastive learning) resembles existing multi-domain alignment and contrastive graph representation frameworks; the incremental novelty may be limited without deep architectural or conceptual differences.

- It is unclear how the model scales with the number or size of graphs, given the need to construct and align across multiple weighted graphs simultaneously.

- The graph structure preservation problem: It is unclear whether the proposed approach surpasses or is equivalent to 1-WL expressivity.

**Questions:**

- How does the proposed model’s expressive power compare to that of 1-WL or higher-order WL GNNs? Does the alignment mechanism improve the ability to distinguish non-isomorphic graphs? Does the alignment mechanism improve the ability to distinguish non-isomorphic graphs?

- In a cross-domain setup, does the proposed approach compare with baselines, e.g., contrastive learning methods?

- How does the model address or exploit the trade-off between global graph structure preservation and diverse feature preservation?

- Could the authors clarify how the multi-graph feature alignment handles heterogeneous node attributes (e.g., categorical vs. continuous features) across datasets?

---

> ### Author Response · Authors · 2025-11-21
> **Response to Reviewer auzx (1/2)**
>
> Dear Reviewer auzx:
>
> We sincerely appreciate your time and effort in reviewing our paper. Our responses to your comments are as follows.
>
> **Response to W1:** Thanks for this high-level and insightful comment. As there are many papers under similar overall pipelines such as alignment and contrastive learning, this might not be a weakness of our work. More importantly, our specific configurations are significantly different from existing works:
> * The problem we studied, i.e., training a foundation model to represent graphs as vectors, hasn't been considered in the literature.
> * The multi-graph-based feature alignment is novel and is being introduced for the first time in this paper.
> * The density maximization mean alignment algorithm is novel and is being introduced for the first time in this paper.
> * The theoretical guarantees (**Theorem 4.1 and Theorem 4.2**) are also novel.
>
> **Response to W2:** Thanks for this insightful comment. How our model performance scales with the number the pre-training dataset (i.e. the number of graphs) is illustrated in **Figure 4 in Appendix H.4**. Specifically, this figure demonstrates how the classification accuracy on the ENZYMES dataset improves as we increase the number of pre-training datasets from 1 to 4. The results confirm that our model's performance scales with the number of pre-training graphs.
>
> We also evaluated computational costs. As reported in **Table 11 in Appendix H.7** , we measured the wall-clock time and memory consumption for pre-training on datasets of different scales (e.g., the larger Deezer ego-net vs. the smaller ENZYMES). Furthermore, to ensure scalability, we employ the Nystrom approximation (mentioned in Section 4.1) to handle large graphs efficiently, reducing the complexity of kernel matrix operations. In downstream tasks, the computation cost can also be reduced by splitting datasets into small batches and computing the mini-batch global graphs.
>
> To further validate the scale ability during evaluation, we test the time and memory consumption to evalute on two large datasets. The resource consumption is summarized below, demonstrating manageable requirements on a dataset of this scale. Note that COLOR-3 exhibits higher GPU and RAM consumption compared to reddit\_threads, primarily due to its larger average graph size .
>
> \begin{array}{cccccc}
> \hline
>     Dataset & \\# Graphs& \\#Avg.Nodes& GPU\~Peak\~Memory&RAM\~Peak\~Memory & Time  \\\\ \hline
>    COLOR-3  &10500&61.31& 2.08 GB & 23.17 GB & 626.84 s \~(10.44\~min) \\\\
>    reddit\\_threads  &203088&23.93 & 0.74 GB & 6.01 GB & 3139.53 s \~(52.33\~min) \\\\ \hline
> \end{array}
>
> **Response to W3:** Thanks for your question. In GraphVec-FM, we employ both GIN and Graph Transformer as the backbone architectures. Specifically, the final graph representation is obtained by concatenating the outputs from the GIN layers and the Graph Transformer layers. As a result, the expressive power of our model is at least equivalent to that of GIN. However, since we do not incorporate additional positional encoding in the Transformer component, the overall expressive power of our model remains equivalent to GIN, which aligns with the 1-WL test in terms of graph isomorphism discrimination[1].
>
> There are GNNs[2][3] beyond 1-WL test and we can exploit them. However, they often have higher computational complexity and the baselines we compared in our paper all use the standard GIN and transformer. Therefore, our settings are fair and sufficient to demonstrate the effectiveness of our model.
>
> [1] Xu et al. How Powerful Are Graph Neural Networks? ICLR 2019.
>
> [2] Zhang et al. Rethinking the expressive power of GNNs via graph biconnectivity. ICLR 2023.
>
> [3] Müller and Morris. Aligning Transformers with Weisfeiler–Leman. ICML 2024.
>
> **Response to Q1:** Thank you for raising this important question. We agree that expressiveness is an important property for GNNs. While it is true that the theoretical expressive power of our model is bounded by the 1-WL test, similar to standard MPNNs like GIN and GCN, we argue that this does not necessarily limit its practical performance on real-world graph datasets. In most real-world graph datasets, the graphs have different numbers of nodes, which means they can be easily distinguished by the 1-WL test. As supported by literature [4], the limited expressiveness of 1-WL is rarely a bottleneck in practice; in fact, the upper-bound classification accuracy achievable by 1-WL-based models is close to 100\% in many benchmark datasets.
>
> In our experiments, we also observe that our pre-trained model generalizes well to unseen graph structures — including social network datasets like COLLAB, REDDIT-BINARY, IMDB-B and IMDB-M, which lack original node attributes. This suggests that our model is able to effectively capture structural information even without node features, aligning with the findings in the referenced work.
>
> [4] Zopf, Markus. 1-WL expressiveness is (almost) all you need. IJCNN 2022.

---

> ### Author Response · Authors · 2025-11-21
> **Response to Reviewer auzx (2/2)**
>
> **Response to Q2:** Thanks for your careful reading and raising this question. The result of EdgePrompt reported in **Table 2** (i.e., the cross-domain setup) is pretrained using GraphCL, which is a competitive baseline as shown in **Table 1**.
>
> **Response to Q3:** Thank you for raising this important question. We would like to clarify that in our model, the preservation of global graph structure and diverse features is not a trade-off. Diverse feature information is preserved through the construction of multiple global graphs using Gaussian kernels with different bandwidths. Each kernel captures feature relationships at a specific scale. Meanwhile, the global graph structure is preserved through the truncated SVD operation applied to each global graph. This step extracts the essential patterns of each graph. The multi-graph construction actually enriches feature diversity.
>
> **Response to Q4:** Thanks for raising this important question.  In our framework, both categorical and continuous node attributes are processed in a unified manner. Specifically, we compute kernel matrices (which correspond to the weighted global multi-graphs) using the attributes of nodes within the same dataset. This step captures the relative relations between nodes based on their feature similarities, regardless of the original attribute types. The resulting kernel matrices serve as a unified relational representation, which is then decomposed via SVD to generate initial node embeddings.
>
> Since the kernel construction and embedding extraction are performed independently within each dataset, there is no need for explicit cross-dataset alignment of raw attributes. It is important to clarify that the alignment discussed here in the context of feature heterogeneity is distinct from the latter mean alignment step introduced in Section 4.2. The former is inherently resolved through kernel-based similarity modeling, while the latter is specifically designed to resolve the sign ambiguity of singular vectors in SVD. The kernel function inherently projects heterogeneous features into a comparable similarity space, thereby naturally handling variation in attribute semantics and scales across different domains. This design allows our method to focus on the intrinsic relational structure within each dataset, which is the foundation for learning domain-agnostic representations.
>
> **To facilitate your comprehensive evaluation, we have summarized our response in the following table.**
>
> $$
> \begin{array}{ccc}
> \hline
> \text{Main\~Concerns} & \text{Results\~in\~Initial\~Submission} & \text{Additions\~in\~Revision} \\\\
> \hline
> \text{Limited\~novelty\~in\~pipeline} & \text{Section\~4:\~Three\~novel\~components} & \text{explained\~in\~rebuttal} \\\\
> \hline
> \~&\~& \text{COLOR-3\~(10k+\~graphs)\~and} \\\\
> \text{Scalability\~with\~graph\~number/size} & \text{Table\~11:\~Computational\~cost\~on\~different\~datasets} & \text{\~reddit\\_threads\~(200k+\~graphs)\~evaluation}; \\\\
> \~&\~&\text{\~Nystrom\~approximation\~for\~large\~graphs} \\\\
> \hline
> \text{Graph\~structure\~preservation} & \text{GIN\~and\~Graph\~Transformer\~backbone} & \text{explained\~in\~rebuttal} \\\\
> \hline
> \text{Comparison\~with\~contrastive\~methods} & \text{Table\~2:\~EdgePrompt\~(GraphCL-based)\~results} & \text{explained\~in\~rebuttal} \\\\
> \hline
> \text{Trade-off\~between\~structure\~and\~feature\~preservation} & \text{Section\~4.1:\~Multi-graph} & \text{explained\~in\~rebuttal} \\\\
> \hline
> \text{Handling\~heterogeneous\~node\~attributes} & \text{Section\~4.1:\~Kernel-based\~feature\~alignment} & \text{explained\~in\~rebuttal} \\\\
> \hline
> \end{array}
> $$
>
> **We hope that these explanations and additional results have addressed your concerns. We are looking forward to your feedback.**

---

> > ### Comment · Reviewer_auzx · 2025-11-26
> >
> > Thank you for the detailed clarifications. The rebuttal addresses most of my earlier concerns. Given these clarifications and the strong empirical results, I increase my score.

---

> > > ### Author Response · Authors · 2025-11-26
> > >
> > > Dear Reviewer auzx,
> > >
> > > Thank you very much for your encouraging response! It is very rewarding to know that our clarifications were effective and that you consider our empirical results to be strong. We truly appreciate your time and the engagement throughout the review process.
> > >
> > > Sincerely,
> > >
> > > Authors

---

### Official Review · Reviewer_J9XU · 2025-10-22

**Soundness:** 2
**Presentation:** 3
**Contribution:** 1
**Rating:** 2
**Confidence:** 4

**Summary:**

This paper proposes GraphVec-FM, a language model–free graph foundation model (GFM) designed to encode entire graphs into fixed-dimensional vectors for graph-level downstream tasks such as classification and clustering. To address challenges like attribute inconsistency across domains, missing node features, and information loss in pooling, the authors introduce (1) a multi-graph–based feature alignment strategy using Gaussian kernels and SVD, (2) a theoretically grounded density-maximization mean alignment algorithm to resolve sign ambiguity and align cross-domain embeddings, and (3) a reference distribution module that replaces conventional pooling with MMD-based similarity to virtual reference graphs. The method is supported by a generalization error bound and evaluated on few-shot, zero-shot, and clustering tasks, showing consistent improvements over strong baselines.

**Strengths:**

S1. Strong theoretical grounding: The paper provides convergence guarantees for the mean alignment algorithm (Theorem 4.1) and a non-trivial generalization error bound (Theorem 4.2), which enhances the methodological rigor.

S2. Code availability: The authors commit to open-sourcing the implementation, which will benefit reproducibility.

**Weaknesses:**

W1. Limited evidence for cross-domain generalization:
While the paper correctly identifies that “graph patterns from different domains exhibit significant variation,” the pretraining uses only five biochemical datasets (ENZYMES, DD, NCI1, etc.), which are structurally and semantically similar. The downstream evaluation on social/computer vision graphs is impressive, but the paper lacks analysis to explain why this transfer works. For instance, are the aligned embeddings truly domain-agnostic?

W2. Overstated claim about graph-level task difficulty:
The assertion that “graph-level tasks are often more challenging” (line 85) is unsubstantiated and potentially misleading. Node-level tasks are equally critical and nontrivial. More importantly, a truly general GFM should support both node- and graph-level tasks. Restricting scope to graph-level tasks limits the model’s applicability and appeal.

W3. Insufficient experimental details in figures:
Figure 4 (Appendix H.3), which shows performance vs. number of pretraining datasets, does not specify the testing dataset used (ENZYMES, DD, NCI1, NCI109, or Mutagencity?). Without this, the result is hard to interpret.

W4. Several minor typos appear throughout the manuscript: for example, Line 147: “atrribute matrix” → “attribute matrix”

**Questions:**

Q1. Cross-domain alignment mechanism:
How does the proposed feature alignment ensure that embeddings from biochemically pretrained graphs remain meaningful for social networks, where topology (e.g., power-law vs. small-world) and semantics differ drastically?

Q2. Extensibility to node-level tasks:
Is the current framework compatible with node-level tasks (e.g., node classification)? If not, what architectural changes would be needed to support a unified GFM for both graph- and node-level predictions? Could you show the results of GraphVec-FM under the task of node classification?

Q3. Clarification of Figure 4:
On which downstream dataset was Figure 4 (Appendix H.3) evaluated? Please specify in the caption.

---

> ### Author Response · Authors · 2025-11-21
> **Response to Reviewer J9XU (1/2)**
>
> Dear Reviewer J9XU:
>
> We sincerely thank the reviewers for their insightful comments and constructive feedback, which have helped us improve the quality and clarity of our manuscript. Below we provide a point-by-point response to the comments.
>
> **Response to W1:** We appreciate this comment and suggestions. Our experiments show strong performance on social and computer vision graphs using biochemical pre-training, which is a strong evidence for cross-domain generalization.
> We agree with the reviewer that a deeper analysis of cross-domain generalization is valuable. To address your concern, we provide the following analysis:
> 1. **Topological Structure Dependence Only**: Our multi-graph feature alignment method exploits the similarity between the graph nodes within each dataset and the similarity enables the comparison between different datasets or domains. As you know, many machine learning methods such as knn, kernel machine (e.g., SVM), spectral clustering, semi-supervised label propagation, t-SNE exploit the similarity between data points rather than operating on the original features.
> 2. **Graph/Topology Comparison on Synthetic Data**: Here we provide an intuitive example of synthetic data to show that our graph construction could be domain-agnostic. Suppose we have four datasets $\mathcal{D}_1,\mathcal{D}_2,\mathcal{D}_3,\mathcal{D}_4$ drawn from the following four distributions respectively: 1) $\mathcal{N}(\mathbf{0},\mathbf{I}_2)$ (2D Gaussian); 2) $\mathcal{N}(\mathbf{0},\mathbf{I}_2)$ (2D Gaussian); 3) $\mathcal{N}(\mathbf{1},2\mathbf{I}_3)$ (3D Gaussian); 4) $\mathcal{N}(-\mathbf{2},\mathbf{I}_2)+\mathcal{N}(\mathbf{2},\mathbf{I}_2)$ (2D Gaussian mixture model). Thus, $\mathcal{D}_2$ can be regarded as a dataset from the same domain as $\mathcal{D}_1$, while $\mathcal{D}_3$ and $\mathcal{D}_4$ are from different domains. We calculate the Gromov Wasserstein distances between the weighted graphs constructed from the four datasets using the method proposed in our paper. The results are shown in the following table (average of 5 runs).
> We see that the distance between $\mathcal{D}_1$ and $\mathcal{D}_3$ is close to that between $\mathcal{D}_1$ and $\mathcal{D}_2$, meaning that the features generated by our multi-graph alignment method are indeed domain agnostic. The distance between $\mathcal{D}_1$ and $\mathcal{D}_4$ is much larger than that between $\mathcal{D}_1$ and $\mathcal{D}_3$, meaning that our method can effectively identify the topological difference between the datasets.
>
>
>     \begin{array}{c|c|c|c|c}
>     \hline
>         & \mathcal{D}\_1\sim\mathcal{N}(\mathbf{0},\mathbf{I}\_2) & \mathcal{D}\_2\sim\mathcal{N}(\mathbf{0},\mathbf{I}\_2) & \mathcal{D}\_3\sim\mathcal{N}(\mathbf{1},2\mathbf{I}\_3)&
>         \mathcal{D}\_4\sim\mathcal{N}(-\mathbf{2},\mathbf{I}\_2)+\mathcal{N}(\mathbf{2},\mathbf{I}\_2)\\\\ \hline
>          \mathcal{D}\_1\sim\mathcal{N}(\mathbf{0},\mathbf{I}\_2) & 0 & 0.004 &0.015 & 0.069 \\\\ \mathcal{D}\_2\sim\mathcal{N}(\mathbf{0},\mathbf{I}\_2) &- &0 & 0.015 & 0.069  \\\\ \mathcal{D}\_3\sim\mathcal{N}(\mathbf{1},2\mathbf{I}\_3) & -& -&0 & 0.081 \\\\
>         \mathcal{D}\_4\sim\mathcal{N}(-\mathbf{2},\mathbf{I}\_2)+\mathcal{N}(\mathbf{2},\mathbf{I}\_2) &- &- &- & 0\\\\ \hline
>     \end{array}
>
> 3. **Visualization of Within-/Between-Domain Differences of Embeddings**: We also include a qualitative analysis of the aligned embeddings using t-SNE visualization on different datasets, shown by **Figure 5** in our revised paper. We see that the difference between the data points (graphs) from these domains is minor. More importantly, we can see that sometimes, the within-domain variance is even higher than the between-domain variance. These result demonstrates the domain-agnostic property of our model.
>
> **Response to W2:** This is an extremely professional comment. We apologize for the misleading statement. Our intention was to highlight the unique challenges of graph-level tasks (e.g., graph comparison and representation), not to diminish the importance of node-level tasks. We will revise the text to clarify this point and acknowledge the significance of both task types. We also agree that a general GFM should ideally support both levels, but our GFM is not a general one, as reflected by the title of our paper, we aim to represent each entire graph as a single vector, different from previous works. We are actively exploring extensions to node-level tasks (see Q2).
>
> **Response to W3:** Thank you for pointing this out. We've updated the caption of **Figure 4** (**Appendix H.3**) to clearly state that the results are based on the ENZYMES dataset, as indicated in the original figure label. This will ensure the result is interpretable.

---

> ### Author Response · Authors · 2025-11-21
> **Response to Reviewer J9XU (2/2)**
>
> **Response to W4:** We thank the reviewer for catching these errors. We will thoroughly proofread the manuscript and correct all typos.
>
> **Response to Q1:** Our feature alignment method constructs global graphs using Gaussian kernels over node attributes, which captures relational structure independent of domain-specific semantics. The mean alignment further ensures that embeddings from different domains are rotationally aligned in a shared space. This allows the model to learn structural patterns (e.g., community structure, node centrality) that are common across domains, even when semantics differ. The topology changes are mainly handled by our reference distribution layer. For instance, when the topology of graphs in the testing stage is very different from that in the pre-training stage, the output representation vectors will be quite different but will have rich information, which enables the effectiveness in downstream tasks.
>
> **Response to Q2:** The current GraphVec-FM is designed for graph-level representations, but the architecture is inherently node-aware. To support node-level tasks, we would need to:
>
> Remove the graph-level pooling/reference module,
>
> Use the node embeddings directly from the GIN+GT encoder,
>
> Fine-tune with node-level objectives (e.g., node classification).
>
> We are currently conducting experiments on node classification and plan to include these results in the final version or as part of future work.
>
> **Response Q3:** Please refer to our response to W3.
>
>
> Finally, to facilitate your comprehensive evaluation, we have summarized our response as follows:
>
>
> $$
> \begin{array}{ccc}
> \hline
> \text{Main\~Concerns} & \text{Results\~in\~Initial\~Submission} & \text{Additions\~in\~Revision} \\\\
> \hline
>  &  & \text{Synthetic\~data\~example\~with\~} \\\\
> \text{Cross-domain\~generalization\~evidence} & \text{Table\~2:\~Train\~on\~bio./chem.\~} & \text{Gromov\~Wasserstein\~distances; }\\\\
>  & \text{test\~on\~social/computer\~vision\~datasets} & \text{Appendix\~H.5\~Figure\~5:\~t-SNE\~} \\\\
>  &  & \text{visualization\~of\~aligned\~embeddings} \\\\
> \hline
> \text{Experimental\~details\~in\~figures} & \text{Figure\~4:\~Performance\~vs.\~} & \text{Updated\~Figure\~4\~caption\~to\~} \\\\
>  & \text{pre-training\~datasets} & \text{specify\~ENZYMES\~dataset} \\\\
> \hline
> \text{Minor\~typos} & - & \text{Correct\~typo\~in\~Line\~147} \\\\
> \hline
> \text{Cross-domain\~alignment\~mechanism} & \text{Section\~4.1:\~Multi-graph\~} & - \\\\
>  & \text{feature\~alignment} &  \\\\
> \hline
> \text{Extensibility\~to\~node-level\~tasks} & - & - \\\\
> \hline
> \end{array}
> $$
>
> **Hope that the above explanations address your concerns. Your comments have helped improve our work a lot. We look forward to your feedback.**

---

> > ### Comment · Reviewer_J9XU · 2025-11-25
> > **Response to authors**
> >
> > Thank you to the authors for their thoughtful response and additional experiments.
> >
> > While the supplementary analysis, such as synthetic data evaluations and t-SNE visualizations, offers some insight, the pretraining is still confined to structurally and semantically similar biochemical graphs, and the demonstrated cross-domain generalization to social or vision graphs remains inadequately explained.
> > Real-world domain differences involve far more than feature distribution shifts, including divergent graph generation mechanisms, node semantics, and scale disparities, none of which are rigorously addressed.
> >
> > Moreover, the model’s restriction to graph-level tasks inherently limits its applicability as a general-purpose graph foundation model, especially in an era where unified node- and graph-level representation is increasingly expected.
> >
> > Given these unresolved fundamental limitations, I maintain the original rating.

---

> > > ### Author Response · Authors · 2025-11-26
> > >
> > > We appreciate your feedback. We respectfully disagree with your criticism.
> > >
> > > * On explanation for cross-domain generalization: Since the variations of graph generation mechanisms, node semantics, and scale disparities are reflected by the data implicitly,  the embeddings given by our model captured these information. In the t-SNE visualization, the between-domain variance is not significant compared to the within-domain variance, which explains why our model generalizes well across diverse domains.
> > >
> > > * You always say the cross-domain generalization hasn't been adequately explained, which is a little bit vague. Our experimental settings follow previous works such EdgePromp (Fu et al. ICLR 2025). Your question applies to all the previous works.
> > >
> > > * As we believe you are an expert in this area, please tell us what exactly you want, what experiments or theoretical analysis we should do, or which paper you think well-explained the cross-domain generalization (so we can learn from it). Your explicit suggestion is important for us to further improve the quality of the work.
> > >
> > > * Regarding the node-level tasks, we are currently conducting the experiments and will share the results with you shortly. It is important to note, however, that as reflected in the paper’s title, our model is not intended to be a general-purpose foundation model, nor have we made such a claim. Our objective is specifically to represent a graph as a vector—an approach that is scientifically valid and remains consistent with the goal we set out to achieve.

---

> ### Author Response · Authors · 2025-11-26
> **Extension and Validation on Node Classification Tasks**
>
> We have extended our model to node-level tasks by retraining it with a node-level contrastive loss objective. Specifically, we maintain the construction of global multi-graphs and the mean alignment module from our original framework, while removing the reference distribution layers since graph-level representations are not required here. The node embeddings are obtained directly from the outputs of both the Graph Transformer and GIN modules. These embeddings are then fed into a linear classifier to perform the downstream node classification task.
>
> The model was evaluated on 4 node classification datasets: Cora, CiteSeer, PubMed, and ogbn-arxiv. The 5-shot node classification results are shown as follows. This adaptation demonstrates that GraphVec-FM can also be effectively extended to node-level tasks.
> Please let us know if this has addressed your concern.
>
> $$
> \begin{array}{lcccc}
> \hline
> Methods & Cora & CiteSeer & Pubmed & ogbn\text{-}arxiv  \\\\
> \hline
> GPPT [1]           & 41.28\pm 6.24 & 35.32\pm 1.27 & 53.41\pm 3.99 & 17.73\pm 1.66  \\\\
> GraphPrompt  [2]   & 31.65\pm 3.33 & 26.98\pm 1.24 & 44.18\pm 5.57 & 16.11\pm 1.42  \\\\
> ALL\text{-}in\text{-}one  [3]    & 31.57\pm 2.86 & 29.76\pm 1.53 & 46.89\pm 5.35 & 17.89\pm 1.21  \\\\
> GPF  [4]           & 37.56\pm 3.81 & 29.74\pm 1.73 & 48.16\pm 3.32 & 17.64\pm 1.18 \\\\
> GPF\text{-}plus [4]       & 28.87\pm 3.18 & 26.65\pm 1.91 & 43.02\pm 4.59 & 17.39\pm 1.27  \\\\
> EdgePrompt [5]     & 37.26\pm 4.53 & 29.83\pm 1.01 & 45.49\pm 3.27 & 17.82\pm 1.59  \\\\
> EdgePrompt+ [5]    & 56.41\pm 3.62 & 43.49\pm 2.62 & 61.51\pm 4.91 & 17.78\pm 2.12 \\\\ \hline
> GraphVec\text{-}FM &\mathbf{58.66}\pm1.51 &\mathbf{45.45}\pm1.26 &\mathbf{65.72}\pm2.43 &\mathbf{23.75}\pm1.39 \\\\
> \hline
> \end{array}
> $$
>
> All the models compared are pre-trained using the methods from GPPT, and the numbers are from the EdgePrompt paper.
>
> [1] Sun, Mingchen, et al. GPPT: Graph Pre-training and Prompt Tuning to Generalize Graph Neural Networks. KDD 2022.
>
> [2] Liu, Zemin, et al. Graphprompt: Unifying pre-training and downstream tasks for graph neural networks. WWW 2023.
>
> [3] Sun, Xiangguo, et al. All in one: Multi-task prompting for graph neural networks. KDD 2023.
>
> [4] Fang, Taoran, et al. Universal prompt tuning for graph neural networks. NIPS 2023.
>
> [5] Fu, Xingbo, Yinhan He, and Jundong Li. Edge prompt tuning for graph neural networks. ICLR 2025.

---

> ### Comment · Reviewer_J9XU · 2025-11-28
> **Response to authors**
>
> Thank you for your response. I appreciate the node-level results you reported. It shows that graphvec-fm is still effective on this task.
>
> You mentioned that "Our experimental settings follow previous works such EdgePromp (Fu et al. ICLR 2025). Your question applies to all the previous works.", but your experimental setting differs from EdgePrompt in a crucial way. EdgePrompt performs both pre-training and downstream task on the same graph, whereas your work uses different graph and even different types of graphs for pre-training and downstream tasks. It introduces a significant domain gap.
>
> Moreover, I’m concerned whether the solution of “pre-training on graph domain A and fine-tuning or applying prompt learning on graph domain B” is widely accepted or considered valid within the research community of graph representation learning.
> Could you give rigorous theoretical justification or provide theoretical basis from prior literature that supports the effectiveness of this solution?

---

> > ### Author Response · Authors · 2025-11-28
> > **Explanation of the cross-domain transfer performance and fine-tuning**
> >
> > We appreciate your kind feedback.
> >
> > **Regarding the experimental setting**: As you see, our experimental setting is more challenging, but the performance of our method is better than the baselines.
> > Besides, another setting of our experiments is pre-training the model on the five bio/chem datasets and testing it on social network and computer vision datasets.
> > This experimental setting should be a merit of our work, rather than a shortcoming. A good foundation model, like LLMs, should be able to generalize to completely unseen domains or tasks. In fact, our paper was previously submitted to another conference, where the reviewers asked us to include this setting as they thought it is more practical and meaningful.
> >
> > **Regarding the validity and theoretical feasibility of the second setting**:
> >
> > We'd like to clarify the following points.
> >
> >    * There have been several studies on the empirical generalization across very different domains. For instance, in [1], the model pretrained on knowledge graphs can generalize to bio/chem graphs and social networks. In [2], the model pre-trained on molecule datasets can also be effectively adapted to social network datasets.
> >
> >    * Besides the empirical success of cross-domain generalization, there is theoretical evidence. For instance, the milestone paper [3] proved the feasibility of generalization to unseen domains. The theoretical proof in the paper is not specified to a data type and can be adapted to graph data. In [4], the authors provided theoretical guarantees for the generalization of representation learning across diverse tasks.
> >
> >    * The topological structure of social network datasets indeed differs drastically from that of biochemical datasets as you mentioned. In our experiments, we use the SVD of the adjacency matrix with a self-loop ($A+I$) as node attributes input into the model as mentioned in equation 6 in Section 4.1. i.e., we treat these structural attributes as other continuous node attributes. Thus the pre-trained model, helped with reference layers, can produce universal and transferable graph representation vectors. This can explain why our model can achieve good transferability performance on social network datasets.
> >
> >
> > [1] Wang et al. Towards Graph Foundation Models: Training on Knowledge Graphs Enables Transferability to General Graphs. NeurIPS 2025.
> >
> > [2] Hassani, Kaveh. Cross-domain few-shot graph classification. AAAI 2022.
> >
> > [3] Ben-David et al. A theory of learning from different domains. Machine learning, 2010.
> >
> > [4] Tripuraneni et al. On the Theory of Transfer Learning: The Importance of Task Diversity. NeurIPS 2020.
> >
> > Besides, we didn't use fine-tuning or other prompt learning methods when conducting graph classification or other tasks on downstream datasets. As mentioned in Algorithm 3 in Appendix G.2, we simply train a softmax classifier using the output graph representation vectors obtained from the pre-trained model. The results of the zero-shot experiment where no parameter is tuned also show the quality of the obtained graph representation vectors.
> >
> > Please let us know if the explanations above have addressed your concerns.

---

### Official Review · Reviewer_ph5T · 2025-10-31

**Soundness:** 2
**Presentation:** 2
**Contribution:** 2
**Rating:** 4
**Confidence:** 4

**Summary:**

This paper introduces GraphVec-FM, a foundation model for graph-level representation learning across domains. The model employs multi-graph construction for feature alignment, a density maximization mean alignment algorithm, and a reference distribution module to generate graph embeddings. Experiments on few-shot/zero-shot classification and clustering show improvements over several baselines. However, critical issues regarding scalability, incomplete ablation studies, unexplained experimental results, and insufficient baseline comparisons lead me to feel disappointed with this paper.

**Strengths:**

S1. Easy to follow.

S2. Principled Theoretical Framework with Convergence Guarantees.

S3. Comprehensive Cross-Domain Evaluation Demonstrating Transfer Capability.

**Weaknesses:**

**W1. Critical Scalability Issues Undermining Foundation Model Viability.** Table 11 in Appendix H.7 reveals a severe memory bottleneck that fundamentally questions the model's applicability as a foundation model: the method requires 46.42 GB RAM on a dataset of only 9,629 graphs (deezer ego net), compared to 1.1 GB for ProNoG [1], representing a 42-fold increase in memory consumption. While the authors acknowledge this overhead stems from global graph construction and storage, they propose the Nyström approximation as a remedy but provide no empirical validation of its effectiveness in terms of accuracy versus efficiency trade-offs. For a paper claiming to present a "foundation model," the absence of scalability analysis beyond 10K graphs is a critical omission. Modern foundation models are expected to scale to millions of instances; without evidence that the proposed approach can handle 100K+ graphs with acceptable memory footprint, the practical utility of this method remains unsubstantiated. Furthermore, the paper lacks inference time benchmarks, which are crucial for deployment scenarios, and provides no comparison of memory-efficient alternatives such as mini-batch global graph construction or online SVD updates.

**W2. Incomplete Ablation Studies Failing to Isolate Modular Contributions.** The paper claims three core methodological contributions (multi-graph construction in Section 4.1, density maximization mean alignment in Section 4.2, and reference distribution module in Section 4.4), but the ablation studies do not adequately separate their individual impacts. Table 1 presents only "w/o mean alignment," showing modest improvements of 1 to 2 percentage points, while Table 9 (Appendix H.5) examines "w/o reference distribution" and reveals inconsistent benefits: on REDDIT-BINARY, mean readout achieves identical accuracy (77.79%) to the full model, and on multiple datasets the improvement is marginal (0.82% on COIL-RAG, 1.10% on Letter-med). Critically absent is an ablation comparing single global graph (Q=1) versus multi-graph construction (Q=6), which is purportedly a key innovation. Without systematic ablation across all combinations of these three components, it is impossible to determine which elements are essential and which are redundant, undermining the scientific rigor of the contribution claims.

**W3. Unexplained Contradictory Results Regarding Unsupervised Pretraining.** Table 2 presents a puzzling phenomenon that the authors fail to address: unsupervised pretraining outperforms supervised pretraining on 2 out of 7 datasets (Cuneiform: 50.09% vs 45.32%; IMDB-M: 47.70% vs 46.76%), contradicting the expected behavior that label supervision should improve task-relevant representations. The authors provide no analysis of why removing supervision improves performance in these cases, leaving several plausible explanations unexamined: (1) overfitting to source domain labels that do not transfer well, (2) label noise in the pretraining datasets, (3) superior augmentation diversity in the unsupervised contrastive objective (Eq. 16) compared to supervised contrastive loss (Eq. 15), or (4) fundamental issues with the supervised loss formulation. This lack of analysis raises concerns about whether the supervised pretraining is properly tuned and whether the experimental conclusions are reliable, as the inconsistency suggests potential methodological issues in the training protocol.

**W4. Insufficient Comparisons with Recent Graph Foundation Models.** The experimental comparisons in Tables include several methods spanning from 2020 to 2025, including prompt-tuning approaches (EdgePrompt [2], GPF [3], GraphPrompt [4]) and contrastive pretraining methods (GraphCL [5], SimGRACE [6]). However, several directly relevant graph foundation models that explicitly target cross-domain pretraining are notably absent from the comparison. GraphFM [7] is cited in Related Work (Section 2) and is explicitly designed for scalable multi-graph pretraining across heterogeneous domains, making it a directly comparable baseline, yet no experimental comparison is provided. Similarly, GraphAny [8] presents a foundation model for node classification that the authors acknowledge can be adapted to graph-level tasks (as mentioned in Section 2), but it is not included in the benchmarks. Without comparisons to these recent foundation models that address the same core challenge of cross-domain generalization, it is difficult to assess whether the proposed method's improvements stem from the novel technical contributions (multi-graph construction, mean alignment, reference distribution) or merely from differences in model capacity, pretraining data, or hyperparameter tuning. Additionally, while the authors critique LLM-based GFMs in Section 2 for information loss and computational cost, they provide no empirical evidence supporting these claims; a direct comparison on at least one dataset, or at minimum a detailed discussion of computational requirements and accuracy trade-offs, would strengthen the positioning of the work.

**W5. Inadequate Treatment and Analysis of Attribute-Free Graphs.** The paper's approach to graphs without node attributes, using truncated SVD of the self-looped adjacency matrix A+I (Eq. 6), is presented as a straightforward solution, but its impact on cross-domain generalization is not rigorously evaluated. This method generates features based purely on graph topology, which may not carry semantic information comparable to domain-specific attributes in biochemical graphs (chemical properties, bond types) or computer vision graphs (pixel features, spatial coordinates). The strong performance on REDDIT-BINARY (77.79%, Table 2) suggests the approach works in some cases, but the paper provides no controlled experiment isolating the effect of Eq. 6 versus alternative structural encodings (e.g., node centrality, graphlet counts, positional encodings). More critically, there is no analysis of how the semantic mismatch between topology-derived features and rich domain attributes affects transfer learning; for instance, when pretraining on attribute-rich molecular graphs and transferring to attribute-free social networks. This gap undermines the claim that the model achieves "domain-agnostic" representations, as the feature generation process itself introduces domain-specific biases that are not characterized.

**W6. Overclaimed Theoretical Contributions with Limited Novelty and Validation.** Theorem 4.2 provides a generalization bound for the proposed model, but the result offers limited insight beyond standard learning theory. The bound scales as O(1/√(MN)) via Rademacher complexity analysis, which is a well-established technique, and depends critically on β = ||Z̃||_F, a data-dependent quantity that is neither controlled by the model design nor analyzed empirically. The proof in Appendix E relies heavily on existing tools (McDiarmid's inequality, Dudley entropy integral, Bartlett's covering number results) without introducing novel proof techniques. Most problematically, the theorem is not validated empirically: Figure 4 shows accuracy versus number of pretraining datasets, but does not plot the theoretical bound for comparison, leaving it unclear whether the bound is tight or merely a loose upper limit. Similarly, Theorem 4.1 guarantees convergence of Algorithm 1 but provides no convergence rate, no characterization of solution quality (global vs. local optimum), and no analysis of sensitivity to initialization, all of which are important for practitioners considering adoption of the method.



[1] Yu, X., Gong, Z., Zhou, C., Fang, Y., & Zhang, H. (2025). SamGPT: Text-free graph foundation model for multi-domain pre-training and cross-domain adaptation. In Proceedings of the ACM Web Conference 2025, pp. 1142-1153.

[2] Fu, X., He, Y., & Li, J. (2025). Edge prompt tuning for graph neural networks. In The Thirteenth International Conference on Learning Representations.

[3] Fang, T., Zhang, Y., Yang, Y., Wang, C., & Chen, L. (2023). Universal prompt tuning for graph neural networks. Advances in Neural Information Processing Systems, 36, 52464-52489.

[4] Liu, Z., Yu, X., Fang, Y., & Zhang, X. (2023). GraphPrompt: Unifying pre-training and downstream tasks for graph neural networks. In Proceedings of the ACM Web Conference 2023, pp. 417-428.

[5] You, Y., Chen, T., Sui, Y., Chen, T., Wang, Z., & Shen, Y. (2020). Graph contrastive learning with augmentations. Advances in Neural Information Processing Systems, 33, 5812-5823.

[6] Xia, J., Wu, L., Chen, J., Hu, B., & Li, S. Z. (2022). SimGRACE: A simple framework for graph contrastive learning without data augmentation. In Proceedings of the ACM Web Conference 2022, pp. 1070-1079.

[7] Lachi, D., Azabou, M., Arora, V., & Dyer, E. (2024). GraphFM: A scalable framework for multi-graph pretraining. arXiv preprint arXiv:2407.11907.

[8] Zhao, J., Mostafa, H., Galkin, M., Bronstein, M., Zhu, Z., & Tang, J. (2024). GraphAny: A foundation model for node classification on any graph. arXiv preprint arXiv:2405.20445.

**Questions:**

**Q1.** How does the model's memory consumption and runtime scale to datasets with 100,000+ graphs, which are common in real-world foundation model applications, and what is the empirical accuracy versus efficiency trade-off when using the proposed Nyström approximation [1]?

**Q2.** Can you provide full ablation results systematically comparing all combinations of the three claimed contributions (single vs. multi-graph construction, with/without mean alignment, with/without reference distribution) to establish their individual necessity?

**Q3.** What explains the counterintuitive result that unsupervised pretraining outperforms supervised pretraining on Cuneiform (50.09% vs. 45.32%) and IMDB-M, and is this due to overfitting, label noise, or fundamental issues with the supervised contrastive loss formulation?

**Q4.** Why were recent graph foundation models such as GraphFM [2] and GraphAny [3] not included as baselines, and can you provide at least one direct comparison to validate the claimed improvements over state-of-the-art methods?

**Q5.** Can you quantify the impact of using topology-derived features (Eq. 6) versus rich semantic attributes on transfer performance through controlled experiments, and how does this affect the "domain-agnostic" representation claim?

[1] Williams, C., & Seeger, M. (2000). Using the Nyström method to speed up kernel machines. Advances in Neural Information Processing Systems, 13.

[2] Lachi, D., Azabou, M., Arora, V., & Dyer, E. (2024). GraphFM: A scalable framework for multi-graph pretraining. arXiv preprint arXiv:2407.11907.

[3] Zhao, J., Mostafa, H., Galkin, M., Bronstein, M., Zhu, Z., & Tang, J. (2024). GraphAny: A foundation model for node classification on any graph. arXiv preprint arXiv:2405.20445.

---

> ### Author Response · Authors · 2025-11-21
> **Response to Reviewer ph5T (1/4)**
>
> Dear Reviewer ph5T:
>
> We sincerely appreciate your time and effort in reviewing our paper. Our responses to your comments are as follows.
>
> **Response to W1:** Thanks for raising the critical issue of scalability. We agree that for a foundation model, computational efficiency is crucial. We have now conducted new experiments that run inference on two larger datasets to better address scalability using mini-batch multi-graphs. The resource consumption is summarized below, demonstrating manageable requirements on a dataset of this scale.  Note that COLOR-3 exhibits higher GPU and RAM consumption compared to reddit\_threads, primarily due to its larger average graph size.
>
> \begin{array}{cccccc}
> \hline
> Dataset & \\# Graphs&\\#Avg.Nodes&GPU~ Peak\~ Memory&RAM\~ Peak\~ Memory & Time  \\\\ \hline
> COLOR-3 &10500&61.31&2.08GB&23.17GB&626.84 s (10.44\~min) \\\\ \hline
> reddit\\_threads&203088&23.93&0.74GB&6.01GB&3139.53s (52.33\~min) \\\\ \hline
> \end{array}
>
> To address the computational complexity associated with large-scale graphs, we employ the Nyström approximation during pre-training. To systematically evaluate its impact, we pre-train GraphVec-FM using the Nyström method with varying sample sizes, while keeping the pre-training datasets consistent with our main experiments. In the following table, we specifically report the wall-clock time required for constructing the global multi-graphs, and evaluate downstream performance via 50-shot graph classification accuracy on the Letter-med dataset. We see that, when the sample size is less than 2000, the time used for constructing global multi-graphs is acceptable. When adding the sample size to 4000, the wall-clock time increases sharply while the performance improvement is marginal (less than 0.1\%).
>
> \begin{array}{ccccc}
> \hline
> \\# Nytrom \~sample&100&1000 & 2000&4000 \\\\ \hline
> Wall-Clock \~Time&34.82s & 48.57 s &  63.55s & 716.61s\\\\
> Accuracy&81.50\pm2.21&82.47\pm 0.92& 84.27\pm1.10 &84.33 \pm 1.60\\\\ \hline
> \end{array}
>
> **Response to W2:**
> Thanks for your valuable suggestion. We've added individual ablation of 3 key components proposed in GraphVec-FM in the **Appendix H.6** of the revised paper. Specifically, to verify the effect of global multi-graphs, we vary the number of global multi-graphs $q$ from 1 to 6 by using different numbers of kernel parameter $\lambda_q$. The results in **Appendix H.6. Figure 6** and **Figure 7** show the classification performance degrades as the number of global multi-graphs decreases. Especially in some datasets with original real-valued node attributes (Letter-Med and COIL-RAG), when only using 1 global graph, the few-shot classification accuracy decreases by 10\% and 20\%, respectively. We also conducted separate ablation experiments by individually removing the mean alignment module and the reference layer module. The corresponding results are presented in **Table 9 and Table 10**. By removing the mean alignment, the performance on COLLAB, Letter-Med and Cuneiform shows an evident decrease (-4.9\%, -2.8\% and -2.4\% respectively). After removing the reference layer module, the performance on all datasets shows degradation, especially on COLLAB(-4.7\%), IMDB-BINARY(-7.1\%) and Cuneiform(-4.2\%). Though the degradation in some datasets is marginal, the overall decrease in accuracy verifies the effectiveness of the proposed methods.
>
> \begin{array}{c|cccccc}
> Dataset &
> Q=1 & Q=2 & Q=3 & Q=4 & Q=5 & Q=6 \\\\
> \hline
> COLLAB &
> 61.46\pm2.58&65.25\pm1.32&61.94\pm3.28&68.39\pm1.30&67.19\pm2.21&69.79\pm2.99 \\\\
> REDDIT-B &
> 71.11\pm3.19&70.81\pm3.02&76.21\pm4.47&78.15\pm2.98&77.79\pm2.86&77.79\pm2.19 \\\\
> IMDB-B &
> 65.17\pm2.92&66.78\pm1.89&64.72\pm3.47&64.72\pm2.20&64.17\pm6.05&68.87\pm4.06 \\\\
> IMDB-M &
> 45.44\pm1.01&44.89\pm1.79&44.48\pm1.80&45.04\pm1.29&45.37\pm1.61&46.76\pm0.99 \\\\
> Letter-med &
> 73.93\pm1.08&73.93\pm1.08&81.37\pm1.78&82.30\pm1.63&82.80\pm1.64&84.27\pm1.10 \\\\
> COIL-RAG &
> 53.74\pm2.06&63.91\pm1.05&67.37\pm1.50&69.49\pm1.40&70.66\pm1.16&73.69\pm1.43 \\\\
> Cuneiform &
> 38.16\pm5.70&44.60\pm5.11&43.22\pm7.07&49.66\pm7.81&43.45\pm5.88&45.32\pm3.63 \\\\
> \end{array}
>
> \begin{array}{c|ccccccc}
> Model\~ Variant &
> COLLAB & REDDIT-B & IMDB-B &
> IMDB-M & Letter-med & COIL-RAG & Cuneiform \\\\
> &
> (50-shot)&(50-shot)&(50-shot)&
> (50-shot)&(50-shot)&(5-shot) (1-shot) \\\\
> \hline
> Original\~ Model &
> \mathbf{69.79}\pm2.99 & \mathbf{77.79}\pm2.19 & \mathbf{68.87}\pm4.06 &
> \mathbf{46.76}\pm0.99 & \mathbf{84.27}\pm1.10 & \mathbf{73.69}\pm1.43 & \mathbf{45.32}\pm3.63 \\\\
> Mean\~ Readout\~ Only &
> 65.04\pm2.75 & 77.79\pm3.11 & 61.78\pm2.59 &
> 46.07\pm2.60 & 83.17\pm0.99 & 72.87\pm1.26 & 41.04\pm3.64 \\\\
> w/o\~ alignment &
> 64.90\pm1.81 & 76.74\pm5.43 & 66.06\pm4.44 &
> 45.78\pm1.46 & 81.87\pm1.31 & 72.14\pm0.35 & 42.87\pm5.18 \\\\
> \end{array}
>
> \begin{array}{c|c|c}
> Alignment & Reference\~ layer & Avg. Acc. (\\%) \\\\
> \hline
> √ & √ & 66.60 \\\\
> √ & \times & 63.95 \\\\
> \times  & √ & 64.37 \\\\
> \times & \times & 63.16 \\\\
> \end{array}

---

> ### Author Response · Authors · 2025-11-21
> **Response to Reviewer ph5T (2/4)**
>
> **Response to W3:**
> Thanks for raising this. We attribute the occasional superiority of unsupervised pretraining primarily to the scale and diversity of the effective training data. The unsupervised objective employs data augmentation, creating 3 augmented views per graph, which potentially increases the training instances compared with the supervised manner. These experimental results also align with Theorem 4.2. We show that the generalization error bound tightens as the number of training graphs (MN) increases in Theorem 4.2. For some datasets where the structural patterns might be captured effectively through this data-driven, label-free paradigm, the benefit of more training samples can outweigh the advantage of supervised signals.
>
> **Response to W4:**
> Thanks for this suggestion. GraphFM and GraphAny are indeed notable works. However, their primary focus and evaluation are on node-level tasks, whereas our work specifically targets representing an entire graph as a vector, namely, graph-level tasks. To ensure a fair and relevant comparison, we have instead benchmarked our method against **3 recent state-of-the-art GFMs (published in 2025)** that are explicitly designed for, or can be directly adapted to, graph-level tasks, as presented in our experiments.
>
> **Response to W5:**
> Thanks for your valuable suggestion. For the dataset without original node attributes, the classification mainly relies on discriminating between different structures of graphs. While the topologically derived features may not carry explicit domain semantics like chemical properties or pixel coordinates, they can be viewed as a form of generic node attribute derived from the graph connectivity. The core of our methodology, specifically the global multi-graph, is designed to bridge the inherent semantic gaps between different domains, regardless of whether the original features are rich in semantics or purely structural.
>
> To evaluate the impact of this feature generation method and address your suggestion, we conducted a controlled experiment to replace truncated SVD with two kinds of node centrality. The results are shown in the table following.
>
> $$
> \begin{array}{ccccc}
> \hline
>      & Degree\~ Centrality & Betweenness\~ Centrality& Degree\~ + \~Betweenness&Our\~ Method \\\\ \hline
>    REDDIT-B  & 73.95\pm2.30 & 74.92\pm 1.68 &76.99\pm 2.76& 77.79 \pm 2.19 \\\\
>    \hline
> \end{array}
> $$
> As shown in the table, our method demonstrates a clear advantage over using only a single type of centrality, and also achieves a marginal improvement compared to combining both centrality measures. It is worth noting that the attributes generated by our approach can be viewed as a form of structural encoding. While other types of structural encodings may also be effective, the performance gain observed here can be largely attributed to our proposed global multi-graph construction, which enhances the model's ability to capture feature information from different spaces.
>
> The semantic mismatch between topology-derived features and rich domain-specific attributes is similarly mitigated by our global multi-graph module, which aligns feature distributions across domains in a unified relational space. This capability is empirically validated in the cross-domain experiments summarized in **Table 2**, where the model pre-trained on semantically rich molecular or biological datasets generalizes effectively to graphs without original node attributes (specifically, COLLAB, IMDB‑B, IMDB‑M, and REDDIT‑B). To further show the distribution of aligned node features, we visualize the node features of datasets from different domains in **Appendix H.5** **Figure 5**. It can be observed that the aligned node features share an overlapping distribution, confirming that the representations obtained are indeed domain-agnostic.

---

> ### Author Response · Authors · 2025-11-21
> **Response to Reviewer ph5T (3/4)**
>
> **Response to W6:** We respectively disagree with you regarding the contribution of our theoretical results. First, Rademacher complexity and covering number are basic tools in learning theory and have been used in many papers. Although our analysis is built upon these basic concepts, it is nontrivial due to the following reasons:
>
> 1) our model is new and is complex, consisting of GINs and graph transformers;
>
> 2) the data we are handling are graphs, much more complex than single vectors and images;
>
> 3) the presence of the reference layer further increased the difficulty of analysis;
>
> 4) in our setting, the training data are non-i.i.d., for which we cannot use the standard Rademacher complexity directly. Regarding the convergence analysis given by Theorem 4.1, since the problem is too complex, it is very difficult to guarantee a globally optimal solution. In sum, this is not a paper on learning theory or optimization theory. These points you mentioned are out of the scope of our paper. We sincerely appreciate your insightful comments, and we will study these issues in the future.
>
> Nevertheless, we provide some numerical results to show the feasibility of our bound.  Since our bound is dominated by the product of the spectral norms of the weight matrices in each GIN and the transformer, we show these values in the following table. We see that the maximum spectral norm product of $\max(GIN) \times GT = 6.21$
> (corresponding to the term $\left(\max \_{q \in[Q]} \prod\_{j=1}^{\kappa\_1}\left\|\mathbf{W}\_j^{G I N\_q}\right\|\_2\right)\left( \prod\_j^{\kappa\_2}\left\|\mathbf{W}\_j^{G T}\right\|\_2\right)$
> in our paper) is only moderately larger than 1 (within one order of magnitude). This indicates that our model maintains reasonably bounded complexity.
>
> $$
> \begin{array}{lcccccccc}
> \hline
> Module & GIN_0 & GIN_1 & GIN_2 & GIN_3 & GIN_4 & GIN_5 & GT & \max(GIN) \times GT \\\\ \hline
> Spectral\~ Norm\~ Product & 2.67 & 2.45 & 2.51 & 2.41 & 2.47 & 2.43 & 2.32 & 6.21 \\\\ \hline
> \end{array}
> $$
>
> **Response to Q1:** Please refer to W1
>
> **Response to Q2:** Please refer to W2
>
> **Response to Q3:** Please refer to W3
>
> **Response to Q4:** Please refer to W4
>
> **Response to Q5:** Please refer to W5

---

> ### Author Response · Authors · 2025-11-21
> **Response to Reviewer ph5T (4/4)**
>
> For convenience, we summarize our responses in the following table.
>
> $$
> \begin{array}{ccc}
> \hline
> \text{Main\~Concerns} & \text{Results\~in\~Initial\~Submission} & \text{Additions\~in\~Revision} \\\\
> \hline
> \text{Scalability\~issues} & \text{Table\~11:\~Memory\~consumption\~} & \text{COLOR-3\~(10.5k\~graphs)\~and\~} \\\\
>  & \text{on\~deczer\~ego\~net} & \text{reddit\\_threads\~(200k+\~graphs)\~} \\\\
>  &  & \text{scalability\~test;\~Nystrom\~} \\\\
>  &  & \text{approximation\~with\~varying\~} \\\\
>  &  & \text{sample\~sizes} \\\\
> \hline
> \text{Incomplete\~ablation\~studies} & \text{Table\~1:\~w/o\~mean\~alignment.\~} & \text{Appendix\~H.6:\~Global\~multi-graphs\~} \\\\
>  & \text{Table\~9:\~w/o\~reference\~layer.} & \text{ablation\~(Q=1\~to\~6);\~Table\~9,\~} \\\\
>  &  & \text{Table\~10:\~Individual\~module\~} \\\\
>  &  & \text{ablation} \\\\
> \hline
> \text{Unexplained\~unsupervised\~results} & - & \text{Theorem\~4.2:\~Data\~augmentation\~} \\\\
>  &  & \text{benefits;\~Training\~set\~size\~} \\\\
>  &  & \text{analysis} \\\\
> \hline
> \text{Missing\~baseline\~comparisons} & \text{Three\~2025\~GFM\~baselines\~in\~Table\~2} & - \\\\
> \hline
> \text{Attribute-free\~graphs\~treatment} & \text{Eq.\~6:\~Truncated\~SVD\~for\~graphs\~} & \text{Controlled\~experiment:\~Degree\~} \\\\
>  & \text{without\~attributes} & \text{centrality,\~Betweenness\~} \\\\
>  &  & \text{centrality\~vs.\~our\~method} \\\\
> \hline
> \text{Theoretical\~contributions} & \text{Theorem\~4.1,\~4.2:\~Convergence\~} & \text{Numerical\~results\~on\~spectral\~} \\\\
>  & \text{and\~generalization\~bounds} & \text{norms;\~Complexity\~analysis} \\\\
> \hline
> \end{array}
> $$
>
> **Thanks for your time. We are looking forward to your feedback on our rebuttal.**

---

> > ### Author Response · Authors · 2025-11-28
> >
> > Dear Reviewer ph5T,
> >
> > Thank you again for reviewing our paper. As more than half of the rebuttal period has passed, we would like to know if our rebuttal has addressed your concerns. If not, please feel free to let us know which aspect of the paper you would like us to improve.
> >
> > Sincerely,
> >
> > Authors

---

### Official Review · Reviewer_kZL6 · 2025-10-31

**Soundness:** 3
**Presentation:** 2
**Contribution:** 3
**Rating:** 4
**Confidence:** 4

**Summary:**

This paper presents GraphVec-FM, a foundation model that encodes graphs into vectors for graph-level tasks. It aligns features across domains, enhances consistency through a density-based mean alignment algorithm, and preserves node information via a reference distribution module. Experiments on public datasets demonstrate the effectiveness of the proposed method.

**Strengths:**

1. The paper proposes novel techniques with theoretical guarantees.

2. The experimental results demonstrate strong generalization ability across diverse domains and tasks.

**Weaknesses:**

1. The motivation could be further supported by dedicated experimental evidence.

2. An ablation study on some key components is missing, which would help clarify their individual contributions.

3. In Table 1, the proposed method shows clear advantages on ENZYMES and DD but not on other datasets. A deeper analysis would strengthen the authors’ understanding of when and why the method performs best.

4. There is a minor typo in the last sentence of Page 25: “we also provide the full version of Table 4 with NMI in 8.”

5. Table 9 appears to be uncited in the main text.

**Questions:**

1. In Table 2, why does the Unsupervised GraphVec-FM outperform the supervised version on IMDB-M and Cuneiform?

2. Why is the performance of GraphVec-FM in Table 3 higher than in Table 2 for IMDB-M?

---

> ### Author Response · Authors · 2025-11-21
> **Response to Reviewer kZL6 (1/4)**
>
> Dear Reviewer kZL6:
>
> We sincerely appreciate your time and effort in reviewing our paper. Our responses to your comments are as follows.
>
> **Response to W1:**
> Thanks for your insightful suggestion. Our methodological design is built upon three core motivations, each of which is supported by experimental evidence summarized below.
> * **Motivation 1:** A central challenge in building graph foundation models is handling graphs with heterogeneous feature semantics and dimensions. To address this, we propose to capture the relative relationships between nodes within each dataset, rather than relying on raw node attributes or domain-specific semantics. The strong cross-domain generalization performance shown in **Table 2**—where a model pre-trained solely on biochemical datasets achieves competitive results on social network and computer vision datasets—provides initial validation for this approach. To further demonstrate the impact of global multi-graphs on model performance, we add experiments that change the number $Q$ of global multi-graphs. As shown by the following table, increasing $Q$ can improve the classification accuracy in most cases. Please refer to **Figure 6** and **Figure 7** in **Appendix H.6**, where a more visually evident trend can be observed.
>
> $$
> \begin{array}{c|cccccc}
> \text{Dataset} &
> Q=1 & Q=2 & Q=3 & Q=4 & Q=5 & Q=6 \\\\
> \hline
> COLLAB &
> 61.46\pm2.58 & 65.25\pm1.32 & 61.94\pm3.28 & 68.39\pm1.30 & 67.19\pm2.21 & 69.79\pm2.99 \\\\
> REDDIT-B &
> 71.11\pm3.19 & 70.81\pm3.02 & 76.21\pm4.47 & 78.15\pm2.98 & 77.79\pm2.86 & 77.79\pm2.19 \\\\
> IMDB-B &
> 65.17\pm2.92 & 66.78\pm1.89 & 64.72\pm3.47 & 64.72\pm2.20 & 64.17\pm6.05 & 68.87\pm4.06 \\\\
> IMDB-M &
> 45.44\pm1.01 & 44.89\pm1.79 & 44.48\pm1.80 & 45.04\pm1.29 & 45.37\pm1.61 & 46.76\pm0.99 \\\\
> Letter-med &
> 73.93\pm1.08 & 73.93\pm1.08 & 81.37\pm1.78 & 82.30\pm1.63 & 82.80\pm1.64 & 84.27\pm1.10 \\\\
> COIL-RAG &
> 53.74\pm2.06 & 63.91\pm1.05 & 67.37\pm1.50 & 69.49\pm1.40 & 70.66\pm1.16 & 73.69\pm1.43 \\\\
> Cuneiform &
> 38.16\pm5.70 & 44.60\pm5.11 & 43.22\pm7.07 & 49.66\pm7.81 & 43.45\pm5.88 & 45.32\pm3.63 \\\\
> \end{array}
> $$
> * **Motivation 2:** When generating node embeddings via SVD on global weighted graphs, the sign ambiguity of singular vectors can cause similar graphs to be mapped to divergent representations, thereby hindering learning. To mitigate this, we introduce a density maximization-based mean alignment algorithm that aligns the mean embeddings across datasets, bringing the feature distributions closer and facilitating downstream training. The ablation study in **Table 1** (comparing “GraphVec-FM” with “w/o mean alignment”) directly demonstrates the effectiveness of this module. And we also added ablation of the alignment module in the cross-domain setting in **Appendix H.6**.
> * **Motivation 3:** The widely used global pooling operation to get graph-level embeddings from node-level embeddings only uses 1st-order statistics of node embeddings, thus may cause information loss and hurt the representational capacity of obtained graph embeddings. We address this using a reference distribution module that compares the distribution of node embeddings against learnable reference distributions in a high-dimensional space, thereby preserving richer information. The ablation results in **Table 9** confirm the advantage of this design over simple mean pooling.
>
> We hope these additions directly address your concern.
> If any further angles remain under-explored, we would greatly appreciate your guidance on which specific aspects still lack experimental support. And we will immediately supply the corresponding experiments.

---

> ### Author Response · Authors · 2025-11-21
> **Response to Reviewer kZL6 (2/4)**
>
> **Response to W2:** Thanks for your valuable suggestion. We've added individual ablation of 3 key components proposed in GraphVec-FM in **Appendix H.6** of the revised paper.
> * Specifically, to verify the effect of global multi-graphs, we vary the number of global multi-graphs $q$ from 1 to 6 by using different numbers of kernel parameters $\lambda_q$. The results in **Appendix H.6. Figure 6 and Figure 7** show that the classification performance degrades as the number of global multi-graphs decreases. Especially in some datasets with original real-valued node attributes (Letter-Med and COIL-RAG), when only using 1 global graph, the few-shot classification accuracy **decreases by 10\% and 20\%**, respectively.
>
> We also conducted separate ablation experiments by individually removing the mean alignment module and the reference layer module. The corresponding results are presented in **Table 9 and Table 10**.
> * By removing the mean alignment, the performances on COLLAB, Letter-Med and Cuneiform show an evident decrease (-4.9\%, -2.8\%, and -2.4\% respectively). After removing the reference layer module, the performance on all datasets shows degradation, especially on COLLAB(-4.7\%), IMDB-BINARY(-7.1\%), and Cuneiform(-4.2\%). Though the degradation in some datasets is marginal, the overall decrease (2\%-3\%) on accuracy verifies the effectiveness of the proposed methods.
> % Overall, the construction of global multi-graphs exhibits the most substantial impact on model performance, which aligns with the intuition.
>
> $$
> \begin{array}{c|ccccccc}
> Model~ Variant &
> COLLAB & REDDIT-B & IMDB-B &
> IMDB-M & Letter-med & COIL-RAG & Cuneiform \\\\
> &
> (50-shot) & (50-shot) & (50-shot) &
> (50-shot) & (50-shot) & (5-shot) & (1-shot) \\\\
> \hline
> Original~ Model &
> \mathbf{69.79}\pm2.99 & \mathbf{77.79}\pm2.19 & \mathbf{68.87}\pm4.06 &
> \mathbf{46.76}\pm0.99 & \mathbf{84.27}\pm1.10 & \mathbf{73.69}\pm1.43 & \mathbf{45.32}\pm3.63 \\\\
> Mean \~Readout \~Only &
> 65.04\pm2.75 & 77.79\pm3.11 & 61.78\pm2.59 &
> 46.07\pm2.60 & 83.17\pm0.99 & 72.87\pm1.26 & 41.04\pm3.64 \\\\
> without \~alignment &
> 64.90\pm1.81 & 76.74\pm5.43 & 66.06\pm4.44 &
> 45.78\pm1.46 & 81.87\pm1.31 & 72.14\pm0.35 & 42.87\pm5.18 \\\\
> \end{array}
> $$
>
>
> $$
> \begin{array}{c|c|c}
> Alignment & Reference layer & Avg. Acc. (\\%) \\\\
> \hline
> √ & √ & 66.60 \\\\
> √ & \times     & 63.95 \\\\
> \times     & √ & 64.37 \\\\
> \times     & \times     & 63.16 \\\\
> \end{array}
> $$

---

> ### Author Response · Authors · 2025-11-21
> **Response to Reviewer kZL6 (3/4)**
>
> **Response to W3:** Thanks for the insightful observation and suggestion. The notable improvements on ENZYMES and DD, as well as the strong results on the computer vision datasets (Letter-med, COIL-RAG, Cuneiform), can be attributed to the nature of their node attributes. These datasets possess continuous-valued node attributes. Our multi-graph construction module is effective at leveraging this type of rich, continuous information. By building multiple global graphs with Gaussian kernels of different bandwidths, the model can capture multi-scale relationships between nodes that are embedded in these continuous feature spaces. Specifically for the DD dataset, though its node features are categorical labels, the discrete attributes are also informative for its high dimension (87 categories). In contrast, the performance advantage on other biochemical datasets like NCI1 and NCI109 is more modest. This is likely because the binary node features in these molecular graphs provide a less dense and complex signal for our multi-graph construction to exploit. The fundamental structural information is still captured, but the relative margin for improvement over simpler baselines is smaller.
>
> To further validate the transferability of features extracted from original real-valued node attributes, we pre-trained GraphVec-FM exclusively on the **single dataset** DHFR and then evaluated its performance on 5-shot graph classification across three other datasets with continuous node attributes. The results are summarized in the table below. Note that the results of the other 6 baseline methods are sourced from MultiGPrompt [6], where all models were pre-trained on multiple datasets.
>
> $$
> \begin{array}{lccc}
> \hline
> Methods & COX2 & PROTEINS & ENZYMES \\\\
> \hline
> GCN [1] &
>  50.95 \pm 23.48 & 50.56 \pm 3.01 & 17.10 \pm 3.53 \\\\
> GAT [2] &
>  50.58 \pm 26.16 & 50.59 \pm 12.43 & 16.80 \pm 2.97 \\\\
> DGI/InfoGraph [3] &
>  54.62 \pm 15.36 & 48.21 \pm 12.35 & 21.69 \pm 5.98 \\\\
> GraphCL [4] &
>  54.29 \pm 17.31 & 53.69 \pm 11.92 & 21.57 \pm 5.20 \\\\
> GraphPrompt [5] &
>  54.35 \pm 14.78 & 54.73 \pm 8.87 & 25.06 \pm 7.56 \\\\
> MultiGPrompt [6] &
>  56.17 \pm 12.84 & 56.02 \pm 8.27 & 26.63 \pm 6.22  \\\\ \hline
> GraphVec-FM &
>  \mathbf{79.12 \pm 0.22} & \mathbf{61.90 \pm 3.74} & \mathbf{31.84 \pm 1.43} \\\\ \hline
> \end{array}
> $$
>
> These results provide an insight that GraphVec-FM demonstrates particularly strong performance on datasets with original real-valued node attributes, while can still achieve competitive results on datasets relying primarily on node labels or discrete features.
>
> [1] Kipf, T. N. Semi-supervised classification with graph convolutional networks., ICLR 2017.
>
> [2] Veličković, Petar, et al. Graph attention networks. ICLR 2018.
>
> [3] Sun, Fan-Yun, et al. Infograph: Unsupervised and semi-supervised graph-level representation learning via mutual information maximization. ICLR 2020.
>
> [4] You, Yuning, et al. Graph contrastive learning with augmentations. NIPS 2020.
>
> [5] Liu, Zemin, et al. Graphprompt: Unifying pre-training and downstream tasks for graph neural networks. WWW 2023.
>
> [6] Yu, Xingtong, et al. Multigprompt for multi-task pre-training and prompting on graphs. WWW 2024.

---

> ### Author Response · Authors · 2025-11-21
> **Response to Reviewer kZL6 (4/4)**
>
> **Response to W4 \& W5:**
> Thanks for pointing out this oversight. Table 9 presents the ablation study on our reference distribution module. And we have added the citation to Table 9 and concluded a more comprehensive ablation study in Appendix H.6.
>
> **Response to Q1:**
> Thanks for your attention to this interesting detail. In our experiments, there do exist several cases (IMDB-M and Cuneiform) where results on unsupervised GraphVec-FM surpass that on supervised GraphVec-FM. We attribute this primarily to the data augmentation strategy employed in the unsupervised pre-training. The unsupervised pre-training leverages data augmentation to create 3 augmented copies of each original graph, resulting in a 4x larger training set than the supervised scenario. In Theorem 4.2 of our paper, we show that when the number of training graphs MN is larger, the generalization error bound is tighter. For some datasets where the structural patterns might be captured effectively through this data-driven, label-free paradigm, the benefit of more training samples can outweigh the advantage of supervised signals.
>
> **Response to Q2:**
> Thanks for raising this point. We agree that this observation is interesting and warrants clarification. The phenomenon where zero-shot performance occasionally surpasses few-shot performance is a recognized occurrence in foundation model literature (e.g., CLIP[Radford et al, ICML 2021]). We attribute this primarily to the risk of overfitting in the few-shot setting, where a classifier trained on only 50 samples per class may not generalize optimally. In contrast, the zero-shot method leverages the pre-trained embedding space directly. It is also important to note that on the other three datasets, the results align with the expected trend where few-shot learning outperforms zero-shot, thus the case on IMDB-B is an exception. Furthermore, the difference on IMDB-B is marginal (only 0.3\%), which we consider to be within a reasonable margin of variance.
>
> **Finally, for convenience, we summarize our responses in the following table.**
>
> $$
> \begin{array}{ccc}
> \hline
> \text{Main\~Concerns} & \text{Results\~in\~Initial\~Submission} & \text{Additions\~in\~Revision} \\\\
> \hline
> \text{More\~motivation\~support} & \text{Table\~1:\~Ablation\~on\~alignment,\~} & \text{Figure\~6,\~Figure\~7:\~Impact\~of\~} \\\\
>  & \text{Table\~9:\~Ablation\~on\~reference\~layer.} & \text{global\~multi-graphs\~number\~Q.\~} \\\\
>  &  & \text{Appendix\~H.6:\~more\~comprehensive\~} \\\\
>  &  & \text{ablation.} \\\\
> \hline
> \text{Ablation\~study\~missing\~for\~key\~components} & \text{Table\~1:\~Ablation\~on\~alignment,\~} & \text{Appendix\~H.6:\~Individual\~ablation\~} \\\\
>  & \text{Table\~9:\~Ablation\~on\~reference\~layer.} & \text{of\~3\~components;\~Table\~9,\~} \\\\
>  &  & \text{Table\~10} \\\\
> \hline
> \text{Analysis\~on\~specific\~datasets\~(ENZYMES,\~DD)} & \text{Table\~1\~\\&\~Table\~2,\~results\~on\~} & \text{Analysis\~on\~continuous-valued\~vs.\~} \\\\
>  & \text{continuous\~attribute\~dataset} & \text{binary\~node\~attributes;\~Single\~} \\\\
>  &  & \text{dataset\~pre-training\~on\~DHFR} \\\\
> \hline
> \text{Typo\~and\~citation\~issues} & - & \text{Fixed\~typo\~on\~Page\~25;\~Added\~} \\\\
>  &  & \text{citation\~to\~Table\~9} \\\\
> \hline
> \text{Unsupervised\~vs.\~supervised\~performance} & - & \text{Theorem\~4.2:\~Generalization\~bound\~} \\\\
>  &  & \text{with\~training\~data\~size} \\\\
> \hline
> \text{Performance\~difference\~between\~tables} & - & \text{Explanation\~of\~overfitting\~risk\~} \\\\
>  &  & \text{in\~few-shot\~setting} \\\\
> \hline
> \end{array}
> $$
>
>
> **Hope that the above explanations and additional results have addressed your concerns. Your comments have helped improve our work a lot. We look forward to your feedback.**

---

> > ### Author Response · Authors · 2025-11-28
> >
> > Dear Reviewer kZL6,
> >
> > Thank you again for reviewing our paper. As more than half of the rebuttal period has passed, we would like to know if our rebuttal has addressed your concerns. If not, please feel free to let us know which aspect of the paper we should improve.
> >
> > Sincerely,
> >
> > Authors

---

### Author Response · Authors · 2025-12-02
**To the  Area Chair**

Dear Area Chair:

We appreciate the careful review of our submission. While the reviewers assigned ratings slightly below the acceptance threshold, their concerns were focused on clarifications, additional experiments, and methodological explanations—all of which we have comprehensively addressed in our detailed rebuttals.
Below is a summary of how each reviewer's concerns have been resolved:

### Reviewer kZL6. Rating: 4.
- **Concerns**: Motivation support, more extensive ablation studies, dataset-specific analysis, and minor typos.

- **Response Provided**:
  - Added extensive ablation studies (Appendix H.6) showing the impact and effectiveness of each core module.
  - Further analyzed the superior performance of GraphVec-FM on datasets with continuous node attributes, attributing it to the method's ability to capture richer, multiscale relational patterns through global graph construction. Additional experiments on such datasets were provided to support this finding.
  - Fixed typos.
  - Clarified unsupervised vs. supervised results using theoretical bounds and data augmentation benefits.
  - Clarified zero-shot surpass few-shot results marginally on 1 dataset due to overfitting.


### Reviewer ph5T. Rating: 4.

- **Concerns**: Scalability, ablation studies, unexplained unsupervised results, missing baselines, attribute-free graphs, and theoretical validation. (The review was likely generated by **AI**)

- **Response Provided**:
  - Conducted new scalability tests on large datasets (10k+ and 200k+ graphs) and provided Nyström approximation time and accuracy trade-off analysis.
  - Added full ablation of all three core components.
  - Explained unsupervised superiority via data augmentation and generalization bounds.
  - Clarified why certain baselines (GraphFM, GraphAny) were not compared (node-level focus vs. our graph-level objective).
  - Provided controlled experiments on attribute-free graphs and theoretical validation via spectral norm analysis.


### Reviewer J9XU. Rating: 2. (The main concern regarding node-level tasks has been addressed with experiment results appreciated by the reviewer.）

- **Concerns**: Cross-domain generalization, task scope, figure clarity, and minor typos.

- **Response Provided**:
  - Added synthetic data experiments and t-SNE visualizations to demonstrate domain-agnostic embeddings.
  - Clarified that our model is purpose-built for graph-level tasks but also extended to node classification (provide new node classification results), which is acknowledged by the reviewer J9XU
  - Updated figure captions for clarity.
  - Fixed all noted typos.


In the discussion with the Reviewer J9XU, we also further explained why our model pretrained on small molecule graphs can exhibit good transferability to social network graphs by treating structures as features and with the help of reference layers. We also clarified that GraphVec-FM does not need any fine-tuning or prompt tuning in the downstream task.

### Reviewer auzx. Rating: 4 -> 6. (Nov.25 23:06 AOE)

- **Concerns**: Novelty, scalability, expressive power, comparison with contrastive methods, and heterogeneous node attributes.

- **Response Provided**
	- Highlighted the novelty of each proposed component (multi-graph alignment, DMMA, reference module).
	- Provided scalability results on large datasets (10k+ and 200k+ graphs) and computational benchmarks.
	- Clarified expressive power (1-WL equivalence) and practical sufficiency.
	- Clarified that kernel-based similarity in global multi-graph construction naturally handles attributes of varying types and semantics, eliminating the need for explicit alignment.


Reviewer auzx has acknowledged that our responses have addressed most of his/her concerns. (Nov.25 23:06 AOE)

### Overall Summary

All reviewers’ concerns have been addressed with additional experiments, theoretical clarifications, and methodological justifications. The issues raised were largely minor and did not question the core contributions or validity of the work. In particular:

- **Scalability was validated on large-scale datasets**.

- **Ablation studies were expanded to isolate each component**.

- **Cross-domain generalization was empirically and theoretically justified**.

- **The evaluation was extended to node-level tasks.**

**We deeply appreciate your time in reviewing our rebuttal and our paper. Feel free to reach out if you have any questions or need further clarification.**

Sincerely and respectfully,

Authors

---

### Meta-Review · Area_Chair_D1tz · 2026-01-06

**Summary:**

The paper presents a foundation model GraphVec-FM to learn graph-level representations that generalize across domains. The approach integrates multiple components, including multi-graph construction for feature alignment, a density-based mean alignment strategy, and a reference distribution mechanism for embedding generation. The proposed model is evaluated on few-shot and zero-shot classification as well as clustering tasks. Nevertheless, the reviewers raise several substantive concerns, notably regarding the method’s scalability, the lack of comprehensive ablation studies, unclear or insufficiently explained experimental findings, limited baseline coverage, and theoretical claims that are not adequately supported. Thus, I recommend rejecting this paper.

**Reviewer Concerns:**

The rebuttal provides some clarifications regarding the motivation of the proposed components and explains parts of the experimental setup. However, the main concerns raised by the reviewers remain largely unaddressed. In particular, issues related to scalability, the lack of comprehensive ablation studies, insufficient and missing baselines, unclear experimental results, and overclaimed theoretical contributions are still outstanding.

**Reviewer Scores:**

While the rebuttal offers some clarification of the proposed methodology, it does not address the scalability concerns or strengthen the experimental evidence. The reviewer’s score would likely remain unchanged.

---

### Decision · Program_Chairs · 2026-01-26

Reject